

# Spin-liquid behaviour and the interplay between Pokrovsky-Talapov and Ising criticality in the distorted, triangular-lattice, dipolar Ising antiferromagnet

**Andrew Smerald[1,2]\* and Frédéric Mila[1]**

**1** Institute of Physics, Ecole Polytechnique Fédérale de Lausanne (EPFL),
CH-1015 Lausanne, Switzerland
**2** Max Planck Insitut für Festkörperforschung, Heisenbergstraße 1,
D-70569 Stuttgart, Germany

\* andrew.smerald@gmail.com

## Abstract

We study the triangular-lattice Ising model with dipolar interactions, inspired by its realisation in artificial arrays of nanomagnets. We show that a classical spin-liquid forms at intermediate temperatures, and that its behaviour can be tuned by temperature and/or a small lattice distortion between a string Luttinger liquid and a domain-wall-network state. At low temperature there is a transition into a magnetically ordered phase, which can be first-order or continous with a crossover in the critical behaviour between Pokrovsky-Talapov and 2D-Ising universality. When the Pokrovsky-Talapov criticality dominates, the transition is essentially of the Kasteleyn type.



# 1 Introduction

Spin liquids, which can be found in both quantum and classical systems, are often defined by the absence of symmetry breaking at low temperature. This raises the question of what is happening at low temperature, and the variety of possible behaviour is comparable to that of the vast array of symmetry-broken phases [1].

One of the oldest and best-understood examples of a classical spin liquid is the Ising model on the triangular lattice with nearest-neighbour interactions. It has been known for many years that this fails to order even at zero temperature [2,3], instead forming a critical state with long-range, algebraic spin correlations [4,5]. The key feature of the spin liquid is that there is robust local ordering associated with the requirement that every triangle has only two equivalent spins, but there are exponentially many configurations that respect this local order, resulting in long-range disorder [2].

This behaviour is not confined to $T = 0$, but holds to a good approximation throughout the region $0 \leq T \lesssim J_1$, where $J_1$ is the nearest-neighbour coupling constant. For $T > 0$ the correlations between spins are exponentially rather than algebraically decaying, but the correlation length remains large within the low-temperature region. The whole region $0 \leq T \leq J_1$ can therefore be considered as a spin-liquid, and weakly-correlated, paramagnetic behaviour only occurs for $T > J_1$.

The reason that the nearest-neighbour, triangular-lattice, Ising antiferromagnet (TLIAF) is so well understood is that it can be mapped onto a 1D model of free spinless fermions [2,4,6]. This almost magical transformation converts a strongly-interacting spin problem into a non-interacting fermion problem, thus making possible the calculation of virtually all quantities of interest directly in the thermodynamic limit. The key to this "magic" is that the constraints imposed by the strong interactions between spins map directly to the Pauli exclusion principle, and can therefore be dealt with trivially in the fermionic picture.

The situation becomes more difficult, and more interesting, when additional interactions couple spins beyond nearest neighbour. Further-neighbour interactions tend to stabilise an ordered phase at temperatures below a characteristic, further-neighbour energy scale, $J_{\mathsf{fn}}$ [7–11], but this leaves open the possibility of spin-liquid behaviour in the temperature window $J_{\mathsf{fn}} \lesssim T \lesssim J_1$.

The difficulty in analysing further-neighbour models is due to the fact that they cannot be mapped onto free-fermion models, and therefore lack simple analytical solutions. Nevertheless, the fermion picture can still provide useful insights, and in particular one can classify different regions of the spin liquid as being weakly or strongly coupled in a fermionic sense. Weak coupling can be expected for $J_{\mathsf{fn}} \ll T < J_1$ where the free-fermion model is only weakly perturbed, and therefore one expects to find the 2D classical equivalent of a Luttinger liquid. On the other hand, for $T \sim J_{\mathsf{fn}}$ the fermionic model is strongly coupled, making simple predictions more difficult.

There is an essentially infinite number of ways to include further-neighbour interactions, and rather than trying to study all possible combinations of couplings, we concentrate on dipolar coupling between spins, where the interactions fall off with the cube of the separation. Nevertheless, we suggest that the results we obtain are very likely to be qualitatively correct for any system in which the coupling constants are monotonically decreasing with distance (if this condition is not respected there are other possibilities [10,11]). The Hamiltonian we consider is therefore given by,

$$\mathcal{H}_{\mathsf{dip}} = \sum_{(i,j)} J_{ij}\sigma_i\sigma_j, \quad J_{ij} = \frac{1}{|\mathbf{r}_i - \mathbf{r}_j|^3}, \tag{1}$$

where $\sigma_i = \pm 1$ denotes an Ising spin at site $i$, the sum over $(i, j)$ includes all possible pairs of

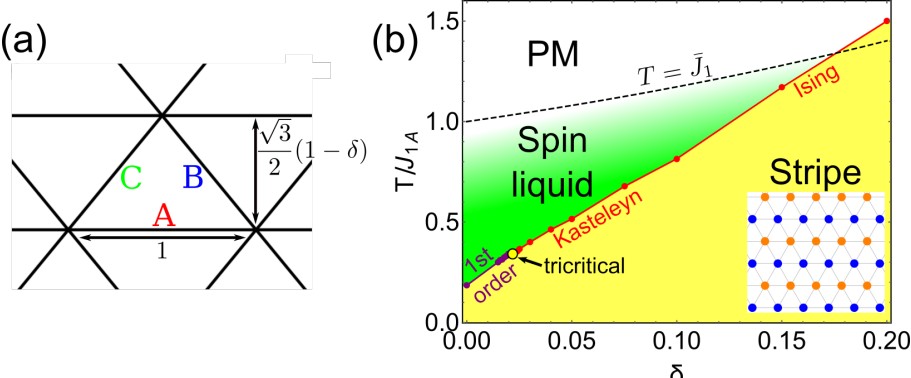

Figure 1: The phase diagram of the dipolar, triangular-lattice, Ising antiferromagnet, $\mathcal{H}_{\text{dip}}$ [Eq. 1], with distortion parameter $\delta$. (a) Triangular-lattice bonds are labelled A, B and C and the distortion is such that the length of A bonds remains invariant while the height of each triangle is reduced by a factor $1 - \delta$, thus shrinking B and C bonds equally. (b) Phase diagram showing the weakly-correlated paramagnet (white), spin liquid (green) and stripe ordered phase (yellow), as calculated by Monte Carlo simulation. The transition between the stripe and spin-liquid phases changes from first to second order at a tricritical point, and the critical behaviour close to the second-order transition can be of Kasteleyn/Pokroksky-Talapov type or 2D Ising. The spin liquid crosses over to an uncorrelated paramagnet at $T \sim \bar{J}_1 = (J_{1A} + J_{1B} + J_{1C})/3$.

spins with $i \neq j$ and $\mathbf{r}_i$ is the position of the $i$th spin.

Our choice to concentrate on dipolar interactions is not made at random, but is motivated by experiment. In particular, artificial spin systems consisting of arrays of out-of-plane nano-magnets arranged on a triangular lattice are starting to be fabricated, and these realise the dipolar TLIAF to a good approximation [12, 13]. While artificial systems have been studied for a number of years [14–23], recent advances have made it possible to make the nano-dots small enough that they remain thermally active at experimentally viable temperatures [24, 25], motivating our study of the equilibrium properties of $\mathcal{H}_{\text{dip}}$ [Eq. 1].

While the primary experimental motivation comes from artificial spin systems, it is worth pointing out that there are many other realisations of TLIAFs with further-neighbour couplings. Examples include crystals of trapped ions [26], the disordered lattice structure of the spin-orbital liquid candidate material $Ba_3Sb_2CuO_9$ [27] and frustrated Coulomb liquids [28].

The only tuneable parameter in $\mathcal{H}_{\text{dip}}$ [Eq. 1] is the temperature, and this already leads to subtle behaviour. Nevertheless, we also find it interesting to consider a second tuneable parameter, namely a small lattice disortion associated with squeezing the lattice, and this is parametrised by $\delta$ as shown in Fig. 1 (we only consider $\delta > 0$).

Such a distortion is both experimentally accessible, since it is possible to build it into the fabrication procedure of artificial spin systems, and convenient, since it can be used to tune the collective transition temperature relative to the single-nano-dot blocking temperature [13].

At least as important is that there is a good theoretical motivation to study the effect of distorting the lattice. For isotropic systems it has been shown that by carefully choosing the relative strengths of the further-neighbour interactions, it is possible to stabilise an intermediate nematic phase that breaks the 3-fold rotational symmetry of the triangular lattice, but not the Ising symmetry [10,11]. This nematic phase can be characterised by a set of fluctuating strings that wind the system and form a disordered grill-like superstructure, and the density of the strings can be controlled by temperature. At low temperatures the transition into an ordered phase takes place via a Kasteleyn transition, and shows Pokrovsky-Talapov critical behaviour, while at higher temperatures the nematic transitions into a paramagnet via a less-interesting first-order transition (see phase diagram in Ref. [11]). The problem with realising such physics in isotropic systems is that one requires a non-monotonically decreasing interaction strength,

with, for example, $J_5 > J_4$, and this is difficult to find in nature.

By adding anisotropy of the type parameterised by $\delta$, the symmetry distinction between the nematic and the paramagnet is lost, and therefore there is never a high-temperature phase transition between a paramagnet and a nematic, whatever the form of the further-neighbour interactions. However, the anisotropy drives the appearance of the most interesting features of the nematic, even for monotonically decreasing further-neighbour interactions, in particular the stabilisation at low temperature of a state with a tuneable density of fluctuating strings that shows Pokrovsky-Talapov critical behaviour approaching a Kasteleyn transition into a fluctuationless, low-temperature ordered phase. Since in such a situation the isotropic model is expected to have a direct first-order transition from the disordered to the ordered state, the addition of anisotropy also opens up the possibility that there is an unusual tricritical point, as a first-order transition turns into a Kasteleyn transition.

As a foretaste of the results to come, we show in Fig. 1 a simplified phase diagram of $\mathcal{H}_{\text{dip}}$ [Eq. 1] as a function of $T$ and $\delta$. One can see that there is a spin-liquid region sandwiched between an ordered stripe phase and a weakly correlated paramagnet. The focus of this article will be on the nature of the spin liquid, as well as on the transition from the spin liquid into the ordered phase, and it can be seen that, as expected, this changes from a first-order to a second-order, essentially Kasteleyn, transition via a tricritical point. The boundary between the spin-liquid and the paramagnet is a crossover and not a phase transition, and a naive guess puts this crossover at $T \approx \bar{J}_1 = (J_{1A} + J_{1B} + J_{1C})/3$, where, $J_{1A}$, $J_{1B}$ and $J_{1C}$ refer to nearest-neighbour interactions along A, B and C bonds (see Fig. 1 for bond labelling). We provide better ways of determining the boundary between the spin-liquid and paramagnetic regions below, but find that they essentially agree with the simple estimate given by $\bar{J}_1$.

Our results for the dipolar TLIAF are presented in the main text of the article, since this is the most experimentally relevant form of the interactions. The extensive appendicies discuss related, but simpler models, in which the couplings are short range. This allows important features of the TLIAF to be isolated and studied in more detail than is possible for the dipolar model, since there is both more freedom to separate competing energy scales, and the simpler models are more amenable to analytic calculations and larger scale Monte Carlo simulations.

## 2 Methods

We employ two complementary methods to study the dipolar TLIAF, Monte Carlo simulation and mapping onto a model of strings/fermions.

### 2.1 Monte Carlo simulations

Monte Carlo is the standard way to simulate 2D Ising systems, but in the case of the dipolar TLIAF proves difficult to equilibrate. To overcome this problem we use a combination of update methods, including parallel tempering, single-spin-flip updates and worm updates.

Equilibration difficulties are most acute close to the transition temperature, and are related to a vanishingly small density of defect triangles (triangles with three equivalent spins). In consequence local-update algorithms (e.g. single-spin-flip) have freezing problems.

Our solution is to employ a worm algorithm [29–31] in which loops are constructed on the dual honeycomb lattice, taking into account the local interactions [11, 32–34] (see Appendix B for a discussion of dimer configurations on the honeycomb lattice). The sets of Ising spins within these loops are then flipped with high probability, allowing the system to tunnel between very different configurations. For systems with local interactions (e.g. up to 5th neighbour) the loops of the worm algorithm can be constructed such that detailed balance is

automatically obeyed, and therefore the algorithm is rejection free [11]. For dipolar interactions the construction of rejection-free updates is prohibitively time consuming, and instead the algorithm uses effective values of the local coupling constants, $J_1^{\mathrm{worm}}$, $J_2^{\mathrm{worm}}$ and $J_3^{\mathrm{worm}}$ to guide the loop creation. If these are well chosen, then accepting a flip of all the spins within the loop has a reasonable probability. In practice we found that these parameters have to be carefully tuned so as to target the configurations expected just above the phase transition.

For the isotropic, dipolar TLIAF just above the phase transition, an acceptance probability of about 0.035 was found for the worm updates, which dropped to about 0.004 on crossing the transition, before continuing to decrease. By running a dense set of temperatures, parallel tempering steps were accepted with a probability of at least 0.8 across the transition, providing a considerable aid to equilibration.

Increasing the transition temperature, for example by adding a lattice distortion, simplifies the simulations by increasing the density of defect triangles in the neighbourhood of the transition. Once this density is high enough, it is possible to use a simple single spin-flip algorithm in combination with parallel tempering.

Simulations are run on hexagonal clusters that preserve all the symmetries of the triangular lattice and have periodic boundary conditions. The linear size of the clusters is $L$ and the total number of triangular-lattice sites is $N = 3L^2$. For the dipolar TLIAF we typically use clusters sizes of $L = 24$, $L = 36$ and $L = 48$. While in 2D the dipolar energy is convergent, it was found to be useful to use Ewald summation of the interactions to take into account their slow decay [35].

When performing Monte Carlo simulations a number of physical quantities are sampled. This includes the energy, $E$, and heat capacity, $C$, as well as the stripe order parameter and its associated susceptibility,

$$m_{\mathrm{stripe}} = \frac{1}{N}\sqrt{\sum_\alpha \left(\sum_i \tau_i^\alpha \sigma_i\right)^2}, \qquad \chi_{\mathrm{stripe}} = \frac{N}{T}\left(\langle m_{\mathrm{stripe}}^2\rangle - \langle m_{\mathrm{stripe}}\rangle^2\right), \qquad (2)$$

where $\alpha \in \{A, B, C\}$, $i$ labels triangular lattice sites and $\tau_i^\alpha$ is the spin at the $i$th site for perfect stripe order parallel to the $\alpha$ bond direction. It is also useful to track the triangular average of the winding number, $\mathbf{W} = (W_1, W_2)$ (see below for the definition of the winding number, and also Appendix B), given by,

$$W_{\mathrm{tri}} = \frac{1}{L}\sqrt{W_1^2 - W_1 W_2 + W_2^2}, \qquad \chi_{\mathrm{W}} = \frac{L}{T}\left(\langle W_{\mathrm{tri}}^2\rangle - \langle W_{\mathrm{tri}}\rangle^2\right), \qquad (3)$$

which is designed such that $W_{\mathrm{tri}} = 1$ for $\mathbf{W} = (L, L)$, $\mathbf{W} = (-L, 0)$ and $\mathbf{W} = (0, -L)$, while $W_{\mathrm{tri}} = 0$ for $\mathbf{W} = (0, 0)$. Alternatively one can track the Monte Carlo average of the density of strings, $n_{\mathrm{string}}$ (see Section B.2 for the definition of strings).

Correlations can be understood by measurement of the spin structure factor, defined in the usual way in real and reciprocal space as,

$$S(\mathbf{r}) = \langle \sigma_i \sigma_j\rangle, \qquad S(\mathbf{q}) = \frac{1}{N}\sum_{\mathbf{r}} e^{i\mathbf{q}\cdot\mathbf{r}} S(\mathbf{r}), \qquad (4)$$

where $\mathbf{r} = \mathbf{r}_i - \mathbf{r}_j$ and $\mathbf{q}$ is in the Brillouin zone of the triangular lattice.

## 2.2 String/fermion mapping

In order to gain intuitive insights into the TLIAF that complement the Monte Carlo simulations, it is useful to consider some of the mappings that can be made.

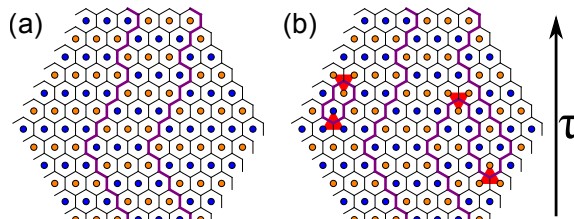

**Figure 2:** Mapping between Ising configurations on the triangular lattice and string configurations on the dual honeycomb lattice. (a) In the absence of defect triangles, strings (purple) wind the system. (b) Defect triangles (red) act as sources and sinks of pairs of strings, allowing strings to turn back on themselves and form short closed loops. By choosing one of the spatial directions as imaginary time, $\tau$, the strings can be interpreted as the worldlines of fermions, with defect triangles corresponding to pair creation or annihilation.

One option is to make a mapping onto a height model, which describes the configurations of the TLIAF in terms of a coarse-grained height field, and this is particularly powerful way to capture the long-wavelength features of the nearest-neighbour TLIAF [36].

For the questions we are interested in here, we find it more intuitive to make a mapping onto strings [37,38], which can then be interpreted as the worldlines of fermions. The mapping onto strings requires the choice of one of the 3 principal lattice directions as being special, and while this is natural for the anisotropic TLIAF the decision is arbitrary for the isotropic model. The strings live on the dual honeycomb lattice, whose bonds bisect those of the triangular lattice (see Fig. 2 and also Appendix B for the link to dimer mappings). Along the special direction each honeycomb bond bisecting an antiferromagnetically-aligned, triangular-lattice bond is assigned to be a segment of a string, while in the other two directions honeycomb bonds bisecting ferromagnetically-aligned, triangular-lattice bonds are assigned as string segments. The string-free configuration is thus seen to correspond to an Ising stripe configuration, with stripes of aligned Ising spins parallel to the special direction.

Using the above definition, each honeycomb-lattice site can be touched by either 0 or 2 string segments, and this ensures the continuity of the strings. For a system with periodic boundary conditions the strings form closed loops, and no two strings can touch, let alone cross, one another (see Fig. 2). If a reference line is chosen that winds the system, the number of strings crossing it has to be even, and therefore string parity is conserved. If no defect triangles are present, then there is no way for a string to turn back on itself, and strings both wind the system and have a fixed length, and in this sense the strings are taut. Defect triangles act as sources or sinks of pairs of strings, allowing them to turn back on themselves and thus either form local loops or long floppy strings that wind the system.

In the absence of defect triangles the string degrees of freedom provide a way of labelling the Ising configurations according to a pair of winding numbers. Two reference lines can be chosen that wrap around a periodic cluster of linear size $L$, and the number of strings crossing each reference line is related to the associated winding number according to $W_i = [L - \text{no. of strings crossing ref. line } i]$ (see Appendix B for the link to dimer representations of the Ising variables). In order to transition between Ising configurations with different winding numbers, it is necessary to either create a pair of defect triangles and transport one of them around the system before they recombine, or to make a non-local change of the Ising configuration.

The properties of the strings allow them to be interpreted as the worldlines of spinless fermions, as is common for 2D statistical mechanics problems [39–44]. The strings "travel" in the direction perpendicular to the special lattice direction, and this is interpreted as the imaginary time direction (see Fig. 2). At a minimum, the quantum Hamiltonian has to include a chemical potential measuring the energy cost of creating a string/fermion, a hopping term

that allows the strings to move in the direction perpendicular to that of imaginary time and pair creation and annihilation terms that take into account the effect of defect triangles. For the case of the nearest-neighbour triangular lattice these three terms are sufficient, and there is an exact mapping onto,

$$\mathcal{H}_{1D} = \sum_i \left[ -\mu c_i^\dagger c_i + t \left( c_i^\dagger c_{i+1} + c_{i+1}^\dagger c_i \right) + \Delta \left( c_i^\dagger c_{i+1}^\dagger + c_{i+1} c_i \right) \right], \qquad (5)$$

where the parameters of the 1D quantum model can be expressed in terms of those of the 2D classical model (see Appendix C, Appendix D and Appendix G for details). The simplicity of the fermionic model is due to the fact that the string-string interactions in the nearest-neighbour Ising model are purely entropic – they arise from the non-crossing constraint – and this maps onto the fermionic Pauli exclusion principle.

Once further-neighbour Ising interactions are included, it is necessary to take into account energetic string-string interactions, and these map onto fermionic interactions. However, a phenomenological free-fermion model can still be applicable when the string/fermion density is low, and can provide quantitative insights into the critical behaviour of the Ising model. Futhermore, qualitative insights into the Ising system can be gained from considering the form of the fermion-fermion interactions.

## 3  Monte Carlo simulation of the dipolar TLIAF

Here we determine the main physical features of the dipolar TLIAF using Monte Carlo simulation. A more detailed discussion of their physical origin is postponed until Section 5.

The ground state of the dipolar TLIAF is 6-fold degenerate and consists of alternating stripes of equivalent Ising spins running parallel to A, B or C bonds (see Fig. 5). The 3-fold degeneracy associated with the choice of stripe direction is multiplied by a 2-fold Ising degeneracy associated with a global spin flip, giving the overall 6-fold degeneracy.

At low temperature there is a stripe-ordered phase, which is dominated by the ground state configurations. Local fluctuations are highly suppressed because they involve the creation of pairs of defect triangles, and the associated energy cost is large. In principle it is also possible to create strings that wind the system, but these are forbidden in the thermodynamic limit as they cost a finite free energy per unit length [10, 11].

On further increasing the temperature, there is a transition from the stripe phase into a disordered phase. A previous study determined the transition temperature to be $T/J_1 \approx 0.18$, but was not able to determine the nature of the transition or achieve equilibration across the transition [45]. Our simulations, which do achieve equilibration, show that the transition is first order, and this is clear from histogram analysis of the energy close to the transition temperature, as shown in Fig. 3. The transition temperature can be determined from the peak in the heat capacity, $C$, the order parameter susceptibility $\chi_{\text{stripe}}$ [Eq. 2] or the winding number susceptibility $\chi_W$ [Eq. 3]. The positions of the peaks in these different quantities coincide for a given system size, $L$, and we show results for $\chi_W$ in Fig. 3. The position of the peak shows a weak $L$ dependence, and using the standard scaling of a first-order transition temperature with $1/N$, we determine a transition temperature of $T_1/J_1 = 0.1845 \pm 0.0010$.

The first-order nature of the transition is also clear from simulations of the heat capacity, which are shown in Fig. 4. Integrating $C/T$ from infinity shows that the disordered state just above the phase transition has an entropy per site of $S/N \approx 0.22$. While this is less than the Wannier entropy $S_{\text{wan}}/N = 0.323\ldots$ associated with the ground state of the nearest-neighbour TLIAF [2], it is still considerable. The low-temperature stripe phase is essentially fluctuationless and has $S/N = 0$, showing that there is a significant entropy release in the first-order transition.

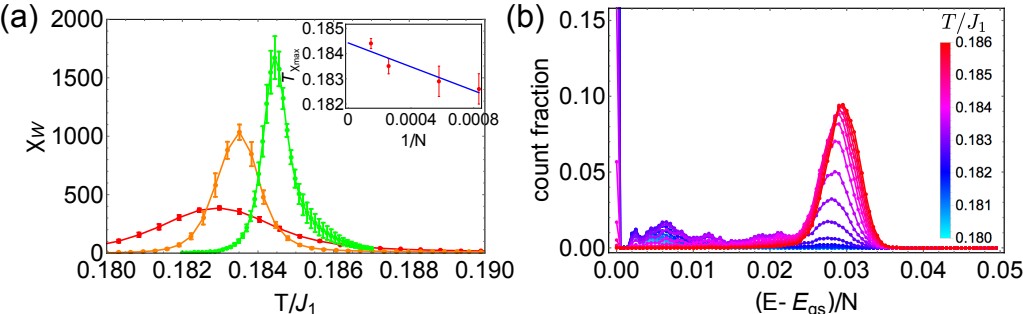

Figure 3: Monte Carlo simulations probing the phase transition in $\mathcal{H}_{\text{dip}}$ [Eq. 1]. (a) The winding number susceptibility, $\chi_{\text{W}}$ [Eq. 3], is shown in a narrow temperature window surrounding the transition for hexagonal clusters of size $L = 24$ (red), $L = 36$ (orange) and $L = 48$ (green). (Inset) The maximum of $\chi_{\text{W}}$ scales approximately with $1/N$, as is standard for a 1st order phase transition, leading to our estimate that in the thermodynamic limit the transition temperature is $T_1/J_1 = 0.1845 \pm 0.0010$. (b) Energy-histogram analysis of an $L = 36$ cluster for temperatures close to the $\chi_{\text{W}}$-maximum of $T_1/J_1 \approx 0.183$. Energies are measured in units of $J_1$ and energy bins have width $0.0005J_1$. A sharp low-energy peak, associated with an almost fluctuationless low-temperature phase, is separated from a Gaussian peak associated with the high-temperature phase by an energy gap of approximately $0.03J_1$ per site. The separation of the two peaks is evidence of a first-order transition.

The nature of the correlations can be accessed via the spin structure factor, and a representative set of examples are shown in Fig. 5. In the stripe-ordered phase there are Bragg peaks associated with the three different stripe directions, and these occur at $\mathbf{q}_{\text{stripe}} = (0, 2\pi/\sqrt{3})$ and symmetry related wavevectors. At temperatures just above the transition, the structure factor develops weight on the whole of the Brillouin zone boundary, but remains peaked at $\mathbf{q}_{\text{stripe}}$, despite the significant entropy change. Small further increases in temperature result in the growth of sharp structure-factor peaks at $\mathbf{q}_{\text{tri}} = (2\pi/3, 2\pi/\sqrt{3})$, and the disappearance of peaks at $\mathbf{q}_{\text{stripe}}$. Further increasing the temperature results in the structure-factor weight becoming more diffuse, and the peaks at $\mathbf{q}_{\text{tri}}$ become less sharp.

The crossover from a highly-correlated paramagnet with sharp structure-factor peaks to a weakly-correlated paramagnet with a diffuse structure factor is governed by the presence or absence of defect triangles. The temperature evolution of the density of defect triangles, $n_{\text{def}}$, is shown in Fig. 4, where $n_{\text{def}}$ is defined as the total number of triangular plaquettes with three equivalent spins divided by the total number of plaquettes, $2N$. Just above the transition temperature the density is very low, and at $T/J_1 = 0.2$, one finds $n_{\text{def}} \approx 10^{-4}$. On the other hand, in the uncorrelated, infinite-temperature limit the defect-triangle density saturates at $n_{\text{def}} = 0.25$, since triangles can take 8 possible equally probable configurations, 2 of which have three spins aligned. For simplicity we take the crossover from strong to weak correlation to be at $n_{\text{def}} = 0.025$ (i.e. 10% of the saturation value) and this occurs at $T = 0.75J_1$, which matches well to the broad peak in the heat capacity (see Fig. 4).

It is possible to estimate the typical energy of an isolated defect triangle by making fits to $n_{\text{def}}$ in the temperature range $T \lesssim J_1$. This shows activated behaviour, and a crude estimate of the functional form is derived in Appendix A. The best fit, shown in Fig. 4, corresponds to a defect-triangle energy of $E_{\text{def}} = 1.60J_1$. This shows that one effect of the further-neighbour interactions is to slightly decrease the typical energy of a defect triangle relative to the nearest-neighbour TLIAF, where $E_{\text{def}} = 2J_1$.

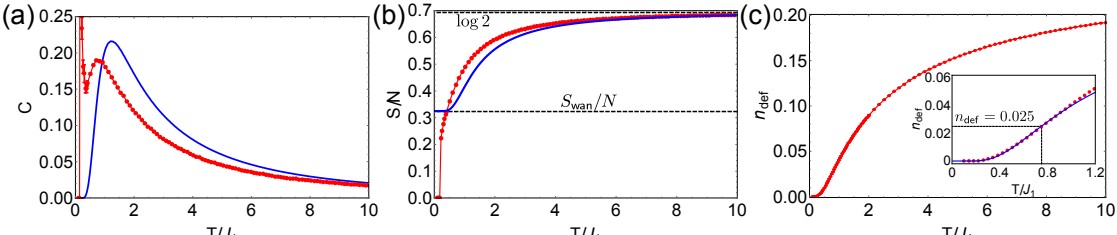

Figure 4: The heat capacity, entropy per site and defect triangle density for $\mathcal{H}_{\text{dip}}$ [Eq. 1] with $\delta = 0$. Error bars are smaller than the point size unless explicitly shown. (a) The heat capacity (red) shows a broad maximum centred on $T = 0.8J_1$, which corresponds to the freezing out of defect triangles, and a sharp peak at $T = 0.185J_1$ due to a first-order phase transition. This can be compared to the case of the nearest-neighbour TLIAF (blue), which shows a similar broad maximum centred on $T = 1.2J_1$, but no low-temperature phase transition. (b) The entropy per site (red) is calculated by integrating the heat capacity from infinity. The entropy passes though $S_{\text{wan}}/N$, at $T = 0.5J_1$, but does not show an extended plateau, unlike the nearest-neighbour TLIAF (blue). At the phase transition there is an entropy jump of $\Delta S/N \approx 0.22$. (c) The defect triangle density, $n_{\text{def}}$. (Inset) At low temperatures $n_{\text{def}}$ follows an activated behaviour, and the blue line is the best fit to Eq. 29 with $A = 0.24$ and $E_{\text{def}} = 1.60$ (see Appendix A).

# 4 Monte Carlo simulation of the anisotropic, dipolar TLIAF

Next we turn to the distorted triangular lattice, and show that even quite small distortions can lead to significant changes in the physical behaviour compared to the isotropic lattice.

The distortion, which is parameterised by $\delta$, leaves the length of A bonds invariant, while reducing the length of B and C bonds and therefore breaks the 6-fold ground-state degeneracy of the isotropic lattice down to a 2-fold degeneracy. Stripes form parallel to A bonds, and the 2-fold, ground-state degeneracy is simply due to an Ising degree of freedom, associated with a global flip of all the spins.

As in the isotropic case, the transition from the stripe phase to the disordered phase can be located using the peaks in the heat capacity, $C$, the order parameter susceptibility $\chi_{\text{stripe}}$ [Eq. 2] or the winding number susceptibility $\chi_W$ [Eq. 3], and the resulting phase diagram is shown in Fig. 6. Histogram analysis of both $n_{\text{string}}$ and $E$ show that the transition changes from first order at low $\delta$ (see Fig. 3) to second order at high $\delta$ (see Fig. 6), and the change occurs at $\delta_{\text{tri}} \approx 0.02$. However, this type of analysis is not a very precise gauge of $\delta_{\text{tri}}$, due to both finite-size effects and the fact that the first-order nature of the transition becomes weaker approaching $\delta_{\text{tri}}$. In Section 5 we use finite-size scaling analysis to determine how the critical exponents depend on $\delta$, and thus demonstrate that the change from first to second order occurs via a tricritical point located at $\delta = \delta_{\text{tri}} = 0.022$ and $T = T_{\text{tri}} = 0.343$.

The spin-liquid region, in which strong local correlation co-exists with long-range disorder, is found to extend until approximately $\delta \approx 0.15$, with the associated temperature window decreasing with increasing $\delta$. This is shown in Fig. 6, where we continue to use a defect-triangle density, $n_{\text{def}} = 0.025$ (10% of the saturation value) to signify the upper extent of the spin liquid.

The structure factor shows signs of a second order transition for $\delta > \delta_{\text{tri}}$ and can also be used to characterise the disordered region. For $\delta_{\text{tri}} \leq \delta \lesssim 0.1$ satellite peaks appear at the transition either side of $\mathbf{q}_{\text{stripe}} = (0, 2\pi/\sqrt{3}(1 - \delta))$, and gradually shift towards $\mathbf{q}_{\text{tri}} = (2\pi/3, 2\pi/\sqrt{3}(1 - \delta))$ as the temperature is increased. We will argue below that these follow the string density, and the structure factor is peaked at $\mathbf{q} = \mathbf{q}_{\text{string}} = (\pi n_{\text{string}}(T, \delta), 2\pi/\sqrt{3}(1 - \delta))$. In the spin-liquid region these peaks are sharp, and this is associated with a long spin-spin correlation length. In the weakly-correlated region, the structure factor becomes diffuse, signalling the breakdown of strong correlations and a

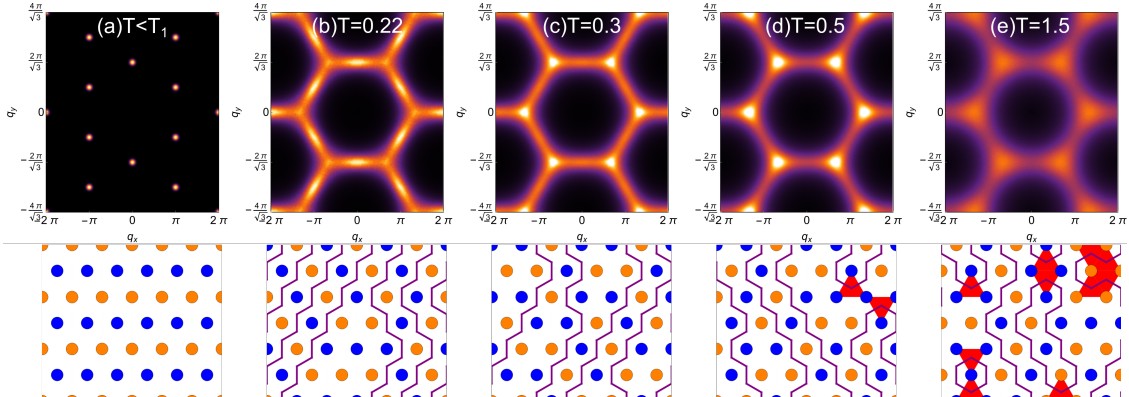

Figure 5: The structure factor of the dipolar TLIAF, together with typical configurations. (a) Below the first order transition there are Bragg peaks at $\mathbf{q}_{\text{stripe}} = (0, 2\pi/\sqrt{3})$ and symmetry-related wavevectors, associated with the stripe ordering. (b) Above the transition weight develops around the BZ boundary, but is peaked at $\mathbf{q}_{\text{stripe}}$ and $\mathbf{q}_{\text{tri}} = (2\pi/3, 2\pi/\sqrt{3})$, and this is associated with a domain-wall network configuration. (c) Further increasing the temperature results in the peaks at $\mathbf{q}_{\text{tri}}$ dominating over those at $\mathbf{q}_{\text{stripe}}$, and this is associated with a switch from attractive to repulsive string-string interactions. (d) At still higher temperatures the peaks at $\mathbf{q}_{\text{tri}}$ remain sharp, and the system can be described as a string-Luttinger liquid. (e) Once the temperature becomes comparable with $J_1$ the peaks at $\mathbf{q}_{\text{tri}}$ lose their sharpness, and this is associated with the proliferation of defect triangles and the loss of significant correlation.

short spin-spin correlation length. For $\delta \gtrsim 0.15$ the spin-liquid region is totally suppressed and, above the transition, the structure factor has peaks at $\mathbf{q}_{\text{stripe}}$. In this region the critical behaviour shows the characteristics of a usual second-order Ising transition into a symmetry-broken, stripe phase, with a structure factor peak developing in the disordered region at the ordering vector.

# 5 Discussion and analysis

In order to gain physical insight into the Monte Carlo simulation results presented in Sections 3 and 4, it is useful to analyse them in terms of the string model introduced in Section B.2. We first discuss the nature of the phase transitions and then move on to the nature of the correlations within the classical spin-liquid region.

## 5.1 The nature of the phase transitions

Depending on the value of the anisotropy parameter, $\delta$, the phase transition has different character, with a first-order transition at $\delta < \delta_{\text{tri}}$ turning into a second order transition at $\delta > \delta_{\text{tri}}$ via a tricritical point at $\delta = \delta_{\text{tri}}$ (see Fig. 6).

The nature of the second-order transition for $\delta > \delta_{\text{tri}}$ is complicated by the fact that it shows a combination of Pokrovsky-Talapov and Ising criticality, with the details depending on $\delta$ and $T$. Here we show that the criticality is 2D Ising over some potentially narrow temperature window close to the transition, and then crosses over to Pokrovsky-Talapov outside this region (see Appendix D for a discussion of similar behaviour in a simpler setting). The width of the Ising temperature window is exponentially suppressed as $\delta$ decreases and for small $\delta$ (e.g. $\delta = 0.05$) the transition is to all intents and purposes in the Pokrovsky-Talapov universality class (i.e. is of Kasteleyn type).

First we consider the extreme case where defect triangles are completely absent and the transition is strictly in the Pokrovsky-Talapov universality class (see also Appendix C). While

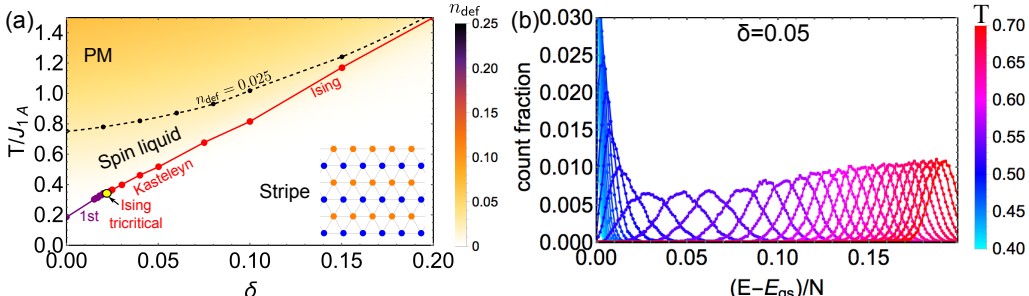

**Figure 6:** The phase diagram of the dipolar TLIAF as a function of $T$ and $\delta$. (a) The phase transition out of the stripe phase changes from first to second order with increasing anisotropy, $\delta$, via a tricritical point at $\delta_{\text{tri}} = 0.022$. Strong correlations are associated with a small defect triangle density, $n_{\text{def}}$, and this is shown by the colour scheme. The crossover from the strong-correlated (spin-liquid) regime to the weakly-correlated paramagnet occurs at approximately $n_{\text{def}} = 0.025$ (i.e. 10% of the saturation value). (b) Energy-histogram analysis, showing a second-order transition for $\delta = 0.05$. Unlike in the case of a first-order transition (see Fig. 3), the energy histogram evolves smoothly across the transition, showing no sign of phase coexistence.

this is never rigorously true in the dipolar TLIAF, it is a good approximation at low $T$. The partition function of the 2D classical model can be mapped onto that of a 1D quantum model with the Hamiltonian,

$$\mathcal{H}_{1D} = \sum_q \omega_q c_q^\dagger c_q, \quad \omega_q = a + bq^2 + \dots, \quad a = a_0(T_c - T), \tag{6}$$

where $T_c$ is the critical temperature of the dipolar TLIAF, and $a_0$ and $b$ are phenomenological parameters. In the simpler case of the nearest-neighbour TLIAF, the exact 1D quantum Hamiltonian is given in Eq. 5, and matching this to Eq. 6 requires $\mu + 2t \to a_0(T - T_c)$, $t \to b$ and $\Delta \to 0$. For the dipolar TLIAF such a microscopic matching of parameters is not possible, but the phenomenological dispersion given in Eq. 6 is valid as long as the probability of creating a defect triangle is very low. Furthermore, truncation of the dispersion beyond the $q^2$ term remains a good approximation as long as the string density is low.

For the fermion model, the $T$ that appears in Eq. 6 does not have the meaning of temperature, and is simply a tuneable parameter that controls the transition from an insulating phase with no fermions ($T < T_c$) to a metallic phase with gapless excitations ($T > T_c$) at a fermi wavevector $q_f = \sqrt{-a/b}$. Once this is mapped back to the dipolar TLIAF, $T$ regains the meaning of temperature, the insulating fermionic phase corresponds to the string vacuum (the stripe phase) and the metallic fermionic phase to the spin-liquid phase where there is a finite density of fluctuating strings that wind around the system.

The density of strings in the 2D classical model is equal to the density of fermions in the 1D quantum model, and can be calculated as,

$$n_{\text{string}} = \frac{1}{\pi} \int_0^{q_f} dq = \frac{q_f}{\pi} \quad \Rightarrow \quad n_{\text{string}} = \begin{cases} 0 & T < T_c \\ \frac{1}{\pi}\sqrt{\frac{-a}{b}} & T > T_c \end{cases}. \tag{7}$$

In the fluctuating phase the string density can be expressed as $n_{\text{string}} \propto (T - T_c)^\beta$, with the critical exponent $\beta = 1/2$. This is characteristic of Pokrovsky-Talapov-type critical behaviour [39, 40] associated with a Kasteleyn transition [46].

To make a connection with previous work [40], it is useful to determine the free energy of the 2D classical model, which is just given by the energy of the 1D quantum model, resulting in,

$$F_{\text{PT}} = \frac{1}{\pi} \int_0^{q_f} \omega_q dq = a_0(T_c - T) n_{\text{string}} + \frac{b\pi^2}{3} n_{\text{string}}^3 + \dots, \tag{8}$$

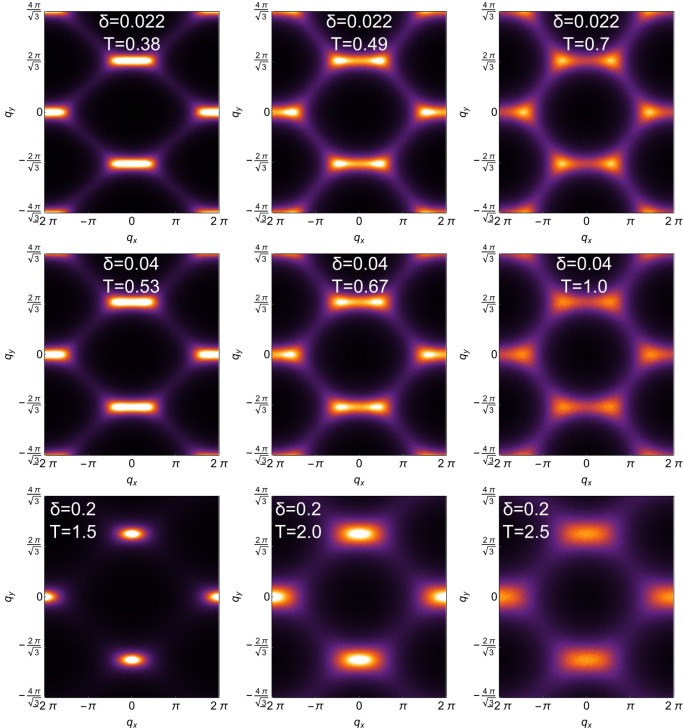

**Figure 7:** The structure factor $S(\mathbf{q})$ [Eq. 4] in the anisotropic, dipolar, triangular-lattice, Ising antiferromagnet. Simulations are run for hexagonal clusters with $L = 48$. (Top row) At $\delta = \delta_{\text{tri}} = 0.022$ there is a tricritical point at $T = T_{\text{tri}} = 0.343$, and above this tricritical point there is a broad maxima in $S(\mathbf{q})$, showing that there are significant fluctuations in the string density, $n_{\text{string}}$. Increasing the temperature leads to a sharper maximum at $\mathbf{q}_{\text{string}}$, showing a reduction in the variance of $n_{\text{string}}$. (Middle row) At $\delta = 0.04$ there are peaks in $S(\mathbf{q})$ at $\mathbf{q}_{\text{string}}$ that coexist close to the phase transition with an additional peak at $\mathbf{q}_{\text{stripe}}$. At higher temperatures the peaks at $\mathbf{q}_{\text{stripe}}$ are suppressed, leaving the peaks at $\mathbf{q}_{\text{string}}$ more visible. (Bottom row) At large values of $\delta$ (here $\delta = 0.2$) the spin liquid region is absent, and $S(\mathbf{q})$ is peaked at $\mathbf{q}_{\text{stripe}}$, displaying the usual behaviour associated with a second-order transition.

where $n_{\text{string}} = q_f / \pi$. In Ref. [40] it was shown that the cubic term describes the string-string repulsion.

For the dipolar TLIAF, the above analysis should apply to the second-order phase transition at anisotropy values $\delta \gtrsim \delta_{\text{tri}}$, where the transition temperature is low enough that there are very few defect triangles in the system. In order to test this, we perform simulations of $\mathcal{H}_{\text{dip}}$ [Eq. 1] for $\delta = 0.05$ at a range of system sizes. While finite-size effects make it hard to directly measure the exponent $\beta$ in simulations, it is possible to write down a scaling hypothesis for $n_{\text{string}}$ and use this to determine $\beta$. We consider,

$$n_{\text{string}}(T, L) = (T - T_c)^{\beta} g_{\text{PT}} \left( \frac{L}{\zeta_{\parallel}} \right), \tag{9}$$

where $g_{\text{PT}}$ is an unknown scaling function and $\zeta_{\parallel}$ is the correlation length in the direction parallel to the domain walls, defined as the lengthscale at which asymptotic values of the structure factor becomes valid [47]. In the critical region it is expected that $\zeta_{\parallel} \propto (T - T_K)^{-\nu_{\parallel}}$ with $\nu_{\parallel} = 1$, and this can be compared to the correlation length perpendicular to the domain walls, which is given by $\zeta_{\perp} \propto (T - T_K)^{-\nu_{\perp}}$ with $\nu_{\perp} = 1/2$ [40, 47, 48]. Since we typically consider hexagonal shaped clusters, finite-size effects will be dominated by $\zeta_{\parallel}$, since this diverges faster than $\zeta_{\perp}$.

In order to quantitatively test the goodness of the data collapse according to the scaling hypothesis [Eq. 9], we use the measure proposed in [49]. The best collapse was found for

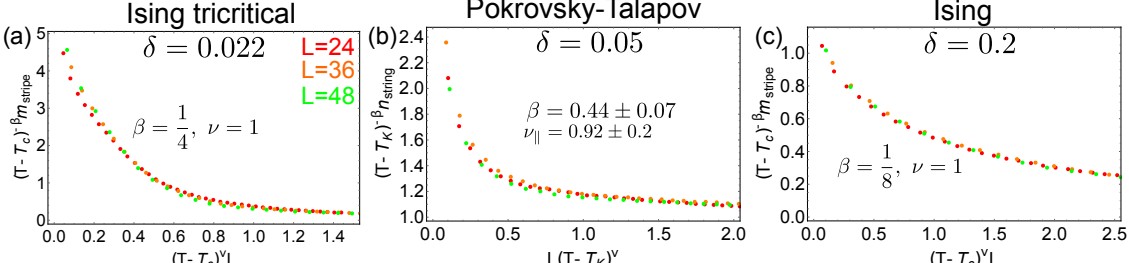

**Figure 8:** Data collapse close to the critical point for $\mathcal{H}_{dip}$ [Eq. 1]. Monte Carlo simulation results are shown for $m_{stripe}$ [Eq. 2] and $n_{string}$ [Eq. 45] on hexagonal clusters of size $L = 24$ (red), $L = 36$ (orange) and $L = 48$ (green). Error bars are in all cases smaller than the point size. (a) In the tricritical region the best data collapse is found for $m_{stripe}$ at $\delta = 0.022$ using the scaling hypothesis given in Eq. 14 with Ising tricritical exponents $\beta = 1/4$ and $\nu = 1$. (b) Data collapse at $\delta = 0.05$ using the scaling hypothesis given in Eq. 9 for Pokrovsky-Talapov critical behaviour and for exponents that minimise the "goodness of collapse" measure proposed in [49]. This gives $\beta = 0.44 \pm 0.07$ and $\nu_{\parallel} = 0.92 \pm 0.2$, which are consistent with the expected Pokrovsky-Talapov exponents $\beta = 1/2$ and $\nu_{\parallel} = 1$. (c) Data collapse at $\delta = 0.2$ using the scaling hypothesis given in Eq. 14 and the Ising critical exponents $\beta = 1/8$ and $\nu = 1$.

$\beta = 0.44 \pm 0.07$ and $\nu_{\parallel} = 0.92 \pm 0.2$ (see Fig. 8), which is consistent with the expected Pokrovsky-Talapov exponents of $\beta = 1/2$ and $\nu_{\parallel} = 1$. Thus we conclude that at low values of $\delta$ the second-order transition shows critical behaviour associated with a Kasteleyn-type transition, which is driven by the appearance of non-local strings that wind the system.

In reality $\mathcal{H}_{dip}$ [Eq. 1] supports a small density of defect triangles at any finite temperature, and even a tiny density of defect triangles drives the transition to be in the Ising universality class (see also Appendix D). However the temperature window over which Ising criticality applies is exponentially suppressed at small $\delta$. In order to better understand the nature of the suppression and the crossover between Ising and Pokrovsky-Talapov criticality, we consider the phenomenological 1D quantum Hamiltonian,

$$\mathcal{H}_{1D} = \sum_{q>0} \left[ A_q \left( c_q^\dagger c_q + c_{-q}^\dagger c_{-q} \right) + B_q \left( c_q^\dagger c_{-q}^\dagger + c_{-q} c_q \right) \right],$$
$$A_q = a_0(T_c - T) + bq^2 + \mathcal{O}(q^4), \quad B_q = 4q z_{def} + \mathcal{O}(q^3), \tag{10}$$

where $z_{def} = e^{-\frac{E_{def}}{T}}$ and $E_{def}$ is a measure of the energy cost of a defect triangle. Diagonalisation via a Bogoliubov transformation, results in,

$$\mathcal{H}_{1D} = \sum_q \omega_q a_q^\dagger a_q + \frac{1}{2} \sum_q \left( A_q - \omega_q \right), \quad \omega_q = \sqrt{A_q^2 + B_q^2}, \tag{11}$$

where $a_q$ and $a_q^\dagger$ are fermionic operators.

In terms of fermions the parameter $T$ controls a transition from a gapped, insulating phase at $T < T_c$ to a gapped, p-wave-superconducting phase at $T > T_c$ via a gapless point at $T = T_c$. In terms of the 2D classical model this maps onto the transition from the stripe phase at $T < T_c$ to a phase with fluctuating strings at $T > T_c$.

The Ising/Pokrovsky-Talapov nature of the criticality is encoded in the location of the minimum of $\omega_q$ [Eq. 11]. Ising criticality is associated with a minimum at $q = 0$, and this occurs for $T_c < T < T_{Is}$, where, $T_{Is} = T_c + 8z_{def}^2/(a_0 b)$. The 2D Ising nature of the criticality in this temperature window is clear from considering the correlation length, which goes as $\xi_{Is} \sim |T - T_c|^{-1}$ [43]. For $T > T_{Is}$ the dispersion minimum moves away from $q = 0$ to $q_{min} = (-a/b - 8z_{def}^2/b^2)^{1/2}$ and the system enters the crossover region between Ising and Pokrovsky-Talapov universality. Pure Pokrovsky-Talapov critical behaviour is recovered in the

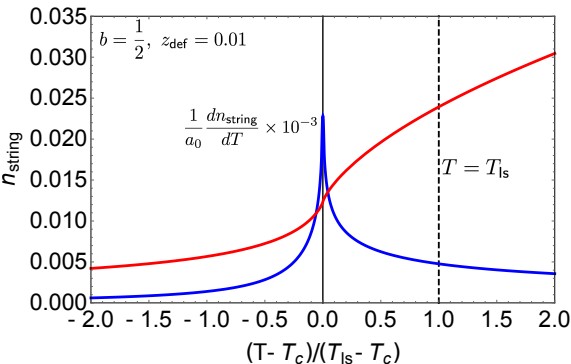

**Figure 9:** Behaviour of the string density, $n_{\text{string}}$, in the vicinity of the critical point, calculated from $\mathcal{H}_{\text{1D}}$[Eq. 10]. For $z_{\text{def}} \neq 0$ the string density remains finite at the critical point, but $dn_{\text{string}}/dT$ shows a logarithmic divergence (blue).

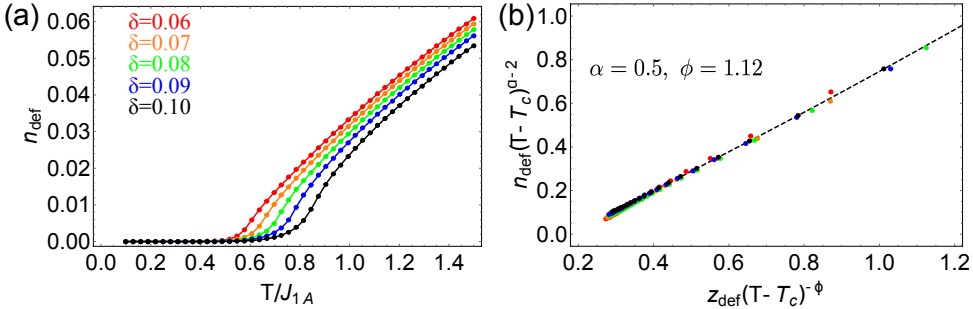

**Figure 10:** The $\delta$ dependence of the defect triangle density, $n_{\text{def}}$, and its crossover scaling. Monte Carlo simulations are run for $L = 24$ for a number of $\delta$ values and error bars are smaller than the point sizes. (a) $n_{\text{def}}$ as a function of T for variable $\delta$. (b) Collapse using the crossover scaling function of Eq. 13 gives $\alpha = 0.5$ and $\phi = 1.12$.

limit $T - T_{\text{c}} \gg T_{\text{Is}} - T_{\text{c}}$, where $q_{\text{min}} = (-a/b)^{1/2}$ is recovered, and therefore $n_{\text{string}} \propto (T - T_{\text{c}})^{1/2}$. In the case $T \ll E_{\text{def}}$, the temperature width of the Ising window is exponentially suppressed due to the $z_{\text{def}}^2$ factor, and the system shows Pokrovsky-Talapov characteristics over all accessible temperatures.

In the critical region the string density and its derivative are given by,

$$n_{\text{string}} = \frac{1}{2\pi} \int_0^\pi dq \left( 1 - \frac{A_q}{\omega_q} \right), \qquad \frac{1}{a_0} \frac{dn_{\text{string}}}{dT} = \frac{1}{2\pi} \int_0^\pi dq \frac{B_q^2}{\omega_q^3}. \qquad (12)$$

In the region $T_{\text{c}} < T < T_{\text{Is}}$ it is possible to extract analytic expressions for these quantities in terms of elliptic integrals. However, these expressions are not so enlightening, and we instead show a numerical evaluation in Fig. 9. For $z_{\text{def}} \neq 0$, the string density is finite both above and below the transition and takes the value $n_{\text{string}} = 2z_{\text{def}}/(\pi b)$ at $T = T_{\text{c}}$. As such $n_{\text{string}}$ is not technically a good order parameter, but it still provides a useful indicator of the transition temperature since $dn_{\text{string}}/dT$ shows a logarithmic divergence according to $dn_{\text{string}}/dT \propto \log|T - T_{\text{c}}|$ (see Fig. 9).

At intermediate values of $\delta$ (e.g. $\delta \approx 0.1$), the dipolar TLIAF should show a crossover between Ising and Pokrovsky-Talapov criticality. As is standard in such situations, an exponent $\phi$ can be used to parametrise the crossover [50, 51]. This appears in the scaling form of physical quantities, and we consider the defect-triangle density, which is expected to scale as,

$$n_{\text{def}}(T, z_{\text{def}}) = |T - T_{\text{c}}|^{2-\alpha} g_\phi \left( \frac{z_{\text{def}}}{|T - T_{\text{c}}|^\phi} \right), \qquad (13)$$

where $\alpha$ is associated with the Pokrovsky-Talapov critical behaviour and therefore expected to take the value $\alpha = 1/2$ [48]. As shown in Fig. 10, scaling collapse of Monte Carlo simulation data in the range $0.06 \leq \delta \leq 0.1$ works well for $\alpha = 0.5$ and $\phi = 1.12$. This compares well to a similar scaling analysis of the nearest-neighbour model, where $n_{\text{def}}$ can be calculated directly in the thermodynamic limit, and we find $\alpha = 0.5$ and $\phi = 1$ [see Appendix D].

At large values of $\delta$ there is a high density of defect triangles at the transition, and one expects Ising criticality to apply over a wide temperature window. That this is indeed the case can be shown by analysing simulation data at $\delta = 0.2$. The standard scaling hypothesis for the Ising order parameter, $m_{\text{stripe}}$ [Eq. 2], is,

$$m_{\text{stripe}}(T, L) = (T - T_{\text{K}})^{\beta} g_{\text{Is}}\left(\frac{L}{\xi_{\text{Is}}}\right), \tag{14}$$

and it is expected that data collapse occurs for $\beta = 1/8$ and $\xi_{\text{Is}} \sim |T - T_{\text{c}}|^{-1}$. It can be seen in Fig. 8 this results in good collapse of the simulation data.

At $\delta = \delta_{\text{tri}} = 0.022$ there is a tricritical point, and the critical behaviour is different from that of the second-order transition. Since the transition temperature at the tricritical point is low, one would naively expect that the associated low density of defect triangles would result in Pokrovsky-Talapov tricritical behaviour (see Appendix E for a discussion of Pokrovsky-Talapov tricriticality). Pokrovsky-Talapov tricriticality can be described by the 1D quantum dispersion,

$$\omega_q = a(T_{\text{K}} - T) + bq^2 + cq^4 + \dots, \tag{15}$$

where $a > 0$, $c > 0$ and $b(\delta - \delta_{\text{tri}})$ is an odd function of $\delta - \delta_{\text{tri}}$ that changes sign when $\delta = \delta_{\text{tri}}$. It follows that exactly at the tricritical point (b=0),

$$q_{\text{f}} = \left(\frac{a(T - T_{\text{K}})}{c}\right)^{\frac{1}{4}}, \quad n_{\text{string}} = \begin{cases} 0 & T < T_{\text{c}} \\ \frac{1}{\pi}\left(\frac{a(T - T_{\text{K}})}{c}\right)^{\frac{1}{4}} & T > T_{\text{c}} \end{cases}, \tag{16}$$

resulting in a critical exponent of $\beta = 1/4$. In terms of the free energy of the 2D classical model, the tricritical point occurs when the cubic term disappears, resulting in,

$$F_{\text{tri}} = a(T_{\text{K}} - T)n_{\text{string}} + \frac{c\pi^4}{5}n_{\text{string}}^5 + \dots \tag{17}$$

Since the cubic term controls the string-string interaction, changing its sign is equivalent to going from a repulsive interaction associated with a second-order transition to an attractive interaction associated with a first-order transition.

For the tricritical point to be effectively in the Pokrovsky-Talapov-tricritical universality class, it is necessary that the Ising temperature window is negligible. The problem with this is that the expression we previously calculated, $T_{\text{Is}} = T_{\text{c}} + 8z_{\text{def}}^2/(a_0 b)$, diverges as $b \to 0$. Including the fourth order term in $\omega_q$ [Eq. 15] results in $T_{\text{Is}} = T_{\text{c}} + 6z_{\text{def}}^{4/3}/(ac^{1/3})$ at $b = 0$. The exponential suppression of the Ising temperature region with $E_{\text{def}}/T$ is thus less pronounced than at the critical point, and, depending on the value of $c$, this could in principle lead to a wide temperature window of Ising tricriticality.

Monte Carlo simulations allow us to test whether Pokrovsky-Talalapov or Ising tricriticality dominates, and come down in favour of a significant Ising-tricritical window. This can be seen in Fig. 8, where $m_{\text{stripe}}$ [Eq. 2] shows convincing data collapse for 2D Ising tricritical exponents. At higher temperatures the system presumably crosses over to Pokrovsky-Talapov tricriticality, but the Ising temperature window is wide enough that this is difficult to ascertain.

For $\delta < \delta_{\text{tri}}$ the transition is first order. In this situation the string-string interaction is attractive, and the string density jumps at the transition. Expansion of the free energy in

terms of the string density is therefore only possible close to the tricritical point, where the jump in the string density is relatively small.

Close to the first-order transition the effective fermion degrees of freedom are long lived, since decay of fermions is associated with the, essentially negligible, presence of defect triangles in the 2D classical model. As a result, even though the fermions are strongly interacting, the interactions just renormalise the free-fermion terms in the Hamiltonian, but don't generate new terms. Thus one can still think in terms of an effective free-fermion dispersion, $\omega_q$. However, the deeper one goes into the first-order region the larger the jump in the string density and the more (even) powers of $q$ have to be retained in the expansion of $\omega_q$. This is because in order to generate a jump in $n_{\text{string}}$ a finite region of $q$ values must have a flat dispersion with $\omega_q = 0$ at the transition, and the larger this region, the more powers of $q$ are required to capture it effectively (strictly all powers of $q$ are required for a region of $\omega_q = 0$, but close to the tricritical point this effect can still be essentially captured by a finite expansion). The predictive power of the phenomenological theory thus reduces away from the tricritical point due to the rapid increase in the number of coefficients.

Taking all the results of this section together, one can see that with relatively few parameters, it is possible to understand the full gamut of critical behaviour in the dipolar TLIAF. The important parameters are the reduced temperature, $(T - T_c)/T_c$, which changes sign at a second-order phase transition, the distortion-dependent parameter $b$, which changes sign at the tricritical point, and the ratio $E_{\text{def}}/T_c$, which determines the temperature window of Ising criticality at the transition.

## 5.2  Correlations between spins

Next we turn to the spin correlations, the nature of which can be used to attain a more detailed understanding of the phase diagram. These can be probed via the spin structure factor, $S(\mathbf{q})$ or $S(\mathbf{r})$ [Eq. 4].

The correlations are very simple in the stripe-ordered phase, which has Bragg peaks in $S(\mathbf{q})$ at $\mathbf{q} = \mathbf{q}_{\text{stripe}} = (0, 2\pi/\sqrt{3}(1-\delta))$, and symmetry-related wavevectors (see Fig. 5 and Fig. 7). Since at low temperatures the stripe phase is essentially fluctuationless, virtually all the spectral weight is contained in the Bragg peak, and in real space there is no decay of the spin correlations with separation. For larger values of $\delta$ the stripe-ordered phase survives to higher temperatures, and for $T \sim J_1$ local fluctuations around the stripe ground state associated with pair creation of defect triangles become significant, resulting in some diffuse scattering surrounding the Bragg peak.

More interesting is the disordered phase, which shows three qualitatively different regimes of spin correlations. At high $T$ there is a weakly-correlated paramagnet in which the spin correlations are short ranged, at lower $T$ and for $\delta < 0.15$ there is a strongly-correlated regime with longer-range correlations that we name a string Luttinger liquid and for $T < T_{\text{tri}}$ and $\delta < \delta_{\text{tri}}$ there is a different strongly-correlated regime that we call a domain-wall network (see Fig. 11).

It is instructive to discuss each of these regimes in more detail, and first we turn to the string Luttinger liquid (see also Appendix E). In this regime the density of defect triangles is low, and there is a repulsive interaction between the strings. Since the strings repel one another they form a (disordered) grill-like superstructure where the average spacing between the strings depends on the string density, $n_{\text{string}}(T, \delta)$ (see Fig. 12). As a result of this superstructure, the structure factor is peaked at $\mathbf{q} = \mathbf{q}_{\text{string}}(T, \delta) = (\pi n_{\text{string}}(T, \delta), 2\pi/\sqrt{3}(1-\delta))$ and related wavevectors (see Fig. 7). However, since the strings are fluctuating, the peaks are not Bragg peaks, and in real space spin correlations decay to zero for large enough separations.

In the absence of defect triangles spin correlations in real space decay algebraically, while in the presence of defect triangles the decay is exponential at large enough distances. We make

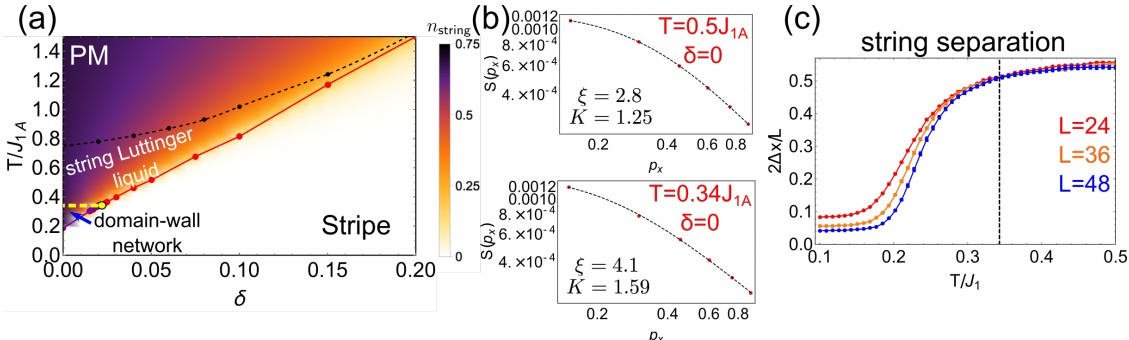

**Figure 11:** Correlations in the dipolar TLIAF. (a) Phase diagram showing the string density, $n_{string}$ and the three qualitatively different regimes of the disordered phase: a weakly-correlated paramagnet (PM), a strongly-correlated string Luttinger liquid and a strongly-correlated domain-wall network. (b) Within the string-Luttinger-liquid regime the correlation length, $\xi$, and Luttinger parameter, $K$, (see Eq. 18) can be determined at a given $T$ and $\delta$ by fitting $S(\mathbf{q})$ using Eq. 19. (c) The crossover from the string-Luttinger-liquid to the domain-wall-network regime is due to a change from repulsive to attractive string-string interactions. The temperature of this crossover can be approximately determined from simulating the average string separation, $\Delta x$ (see Appendix E.4 for a definition of $\Delta x$), in a system constrained to have exactly two strings.

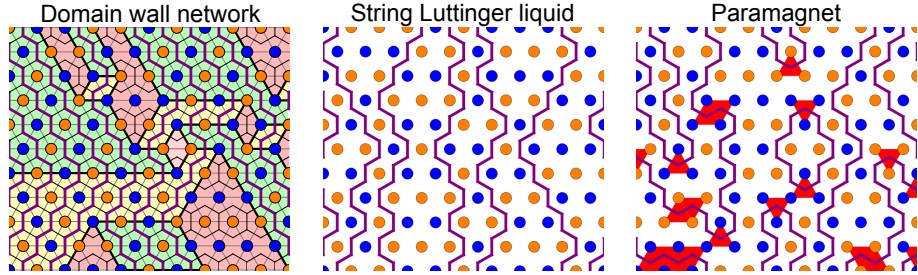

**Figure 12:** Typical string configurations in the domain-wall-network, string-Luttinger-liquid and paramagnetic regions. In the domain-wall-network region strings typically wind the system and attractive string-string interactions cause them to bind together. In the string-Luttinger-liquid region the strings also tend to wind the system, but repulsive interactions result in a grill-like superstructure with strings avoiding one another as far as possible. In the paramagnet there are many defect triangles that act as sources and sinks of pairs of strings, resulting in the strings being floppy and forming short closed loops.

the ansatz that the real-space, spin-correlation function takes the asymptotic form,

$$S(\mathbf{r}) \propto \frac{\cos[\mathbf{q}_{string} \cdot \mathbf{r}] \, e^{-\frac{r_x}{\xi_\perp}} e^{-\frac{r_y}{\xi_\parallel}}}{|\mathbf{r}|^{\frac{K}{2}}}, \tag{18}$$

where $\xi_\perp$ and $\xi_\parallel$ are correlation lengths in the directions perpendicular and parallel to the strings (in the isotropic case $\xi_\perp = \xi_\parallel$). The exponential nature of the decay only becomes apparent when the spin separation is comparable to the correlation length $\xi_\perp$ or $\xi_\parallel$. For low defect-triangle densities, this correlation length is typically large (it becomes infinite when the defect triangle density goes to zero) and for $r \lesssim \xi$ the spin correlations are essentially algebraic.

The parameter $K$ detemines the speed of the algebraic part of the decay, and is nothing but the Luttinger parameter familiar from 1D fermionic systems [52]. That this should appear is unsurprising given the mapping between strings and spinless fermions (see Section B.2) and due to the fact that a repulsive interaction between strings maps onto a weakly attractive interaction between fermions, one expects $K > 1$ (for comparison non-interacting fermions have $K = 1$).

In practice simulations show that the reciprocal-space structure factor in the string-Luttinger-liquid region is dominated by peaks at $\mathbf{q} = \mathbf{q}_{\text{string}}$, but there is also spectral weight on the line of $\mathbf{q}$ values joining $\mathbf{q}_{\text{string}}$ and $-\mathbf{q}_{\text{string}}$ (see Fig. 7). In consequence it is better to fit the structure factor in reciprocal space than in real space, and Fourier transforming the asymptotic form given in Eq. 18 in the vicinity of $\mathbf{q} = \mathbf{q}_{\text{string}}$ and for $\xi_\perp = \xi_\parallel = \xi$ gives [53],

$$S(\mathbf{p}) \propto \frac{1}{p^{2-\frac{K}{2}}} g(p\xi), \tag{19}$$

where $\mathbf{p} = \mathbf{q} - \mathbf{q}_{\text{string}}$, $p = |\mathbf{p}|$ and,

$$\begin{aligned}
g(p\xi) &= 2\pi \int_0^\infty dx\ x^{1-\frac{K}{2}} e^{-\frac{x}{p\xi}} J_0(x) \\
&= 2\pi (p\xi)^{2-\frac{K}{2}}\, \Gamma\left(2 - \frac{K}{2}\right) {}_2F_1\left(\frac{6-K}{4}, 1 - \frac{K}{4}; 1; -(p\xi)^2\right),
\end{aligned} \tag{20}$$

where $J_0(x)$ is the Bessel function of the first kind, $\Gamma(x)$ is the Euler Gamma function and ${}_2F_1(a, b; c; z)$ is the hypergeometric function. The result of fitting this to simulations for $\delta = 0$ and appropriate $T$ is shown in Fig. 11, and it can be seen that the Luttinger parameter does indeed take values $K > 1$, and the correlation length can be many multiples of the lattice spacing. A more precise determination of $K$ and $\xi$ would require simulations on larger clusters (for a numerical determination of $K$ in a simpler model see Appendix E).

It is clear from the simulations of $S(\mathbf{q})$ shown in Fig. 5 that the string-Luttinger liquid regime does not survive all the way down to the phase transition when $\delta < \delta_{\text{tri}}$. Rather than being peaked at $\mathbf{q} = \mathbf{q}_{\text{string}}$, the low-temperature structure factor in the disordered region has spectral weight spread around the BZ boundary, and in particular weight starts to develop at $\mathbf{q} = \mathbf{q}_{\text{stripe}}$. As was shown in Ref. [11] this type of stucture factor is associated with the formation of sizeable domains of stripe order, with neighbouring domains having stripes along different principal axes (see Fig. 12).

The formation of large stripe-ordered domains is suggestive that within this regime the strings attract one another (see also Appendix E). It makes sense that the crossover from the string-Luttinger-liquid regime (high $T$, repulsive string-string interactions) to the domain-wall-network regime (low $T$, attractive string-string interactions) should occur at approximately $T = T_{\text{tri}}$, and this is consistent with the $S(\mathbf{q})$ measurements (see Fig. 5). A simple way to test wether this is the case is to perform simulations in a highly restricted manifold of Ising configurations containing two strings, each of which winds the system. The average separation of the strings does indeed show a significant drop starting at $T \approx T_{\text{tri}}$, indicating a shift from repulsive to attractive interactions (see Fig. 11). We find that the temperature at which this change occurs is essentially independent of $\delta$ in the relevant region ($0 < \delta < \delta_{\text{tri}}$), and therefore the crossover between the string-Luttinger-liquid and domain-wall-network regimes is approximately flat, as shown in Fig. 11.

In terms of fermions, the domain-wall network state can be thought of as being a fluctuating, phase-separated state, with a loose analogy to the clustering of holes in superconductors [54, 55].

At high temperatures the system forms a weakly-correlated paramagnetic regime. The correlations can still be described by Eq. 18, but the correlation length is comparable to the lattice spacing, and so the correlations are exponentially decaying at all length scales. While we have defined the crossover from the Luttinger liquid regime to the weakly-correlated regime in terms of the density of defect triangles reaching 10% of its saturation value (see Fig. 6) this is roughly equivalent to defining a crossover in terms of the correlation length reducing to about 2 lattice spacings.

For $\delta \gtrsim 0.15$ there is a direct transition from the stripe-ordered phase to a standard paramagnet, and as such the structure factor shows the usual features of a second-order transition, with spectral weight building up at the ordering vector as the transition is approached from above, and a diverging correlation length at the transition that results in the formation of a Bragg peak.

### 5.3 Triangular lattice antiferromagnets with general couplings

Here we take a step back and discuss the general features of TLIAF models with monotonically decreasing further-neighbour interactions. We have argued that a good way to understand such models is in terms of the string degrees of freedom, which can be thought of either in their 2D classical incarnation or as the worldlines of spinless fermions in 1D. As such we would like to determine which energy scales present in a given microscopic model dictate the behaviour of the strings and therefore the form of the phase diagram and the physical observables.

In general TLIAF models have many competing couplings, as is clearly true in the dipolar case. Our claim is that these can in most cases be distilled into four important energy scales (it is worth noting that other energy scales can become important if the further-neighbour interactions are not monotonically decreasing interactions [10, 11]).

The first and most important energy scale is the isotropic part of the nearest-neighbour interaction; that is the part of the nearest-neighbour interaction that does not vary with the anisotropy (i.e. $J_{1A}$). This approximately sets the energy cost of creating defect triangles, and therefore "interesting", strongly-correlated physics only occurs in the region $T < J_{1A}$.

Next are two energy scales that combine the isotropic parts of the further-neighbour couplings. The first of these, $J_{fn}$, is a measure of the internal energy of a string and also sets the string-string interaction energy scale. As an example, for the TLIAF with $J_1$, $J_2$ and $J_3$ couplings it is given by $J_{fn} = J_2 - 2J_3$ [10]. This shows that even if the further-neighbour couplings are comparable with $J_1$, their combined effect can still be small due to frustration. The second energy scale is $J_c$, and this is related to the energy cost associated with a string changing direction, and in the case of the $J_1$-$J_2$-$J_3$ model this is given by $J_c = J_2$. One thing that is important to note is that we always consider $J_{fn}, J_c > 0$, and if this is not the case different physics can be expected [11].

The final energy scale we consider is a measure of the anisotropy and is labelled $J_{an}$. For the dipolar TLIAF it clearly depends on $\delta$, and a rough estimate is given by the difference in the nearest-neighbour interaction strengths, resulting in,

$$J_{an}(\delta) \approx J_{1B} - J_{1A} = \frac{9\delta}{4}J_{1A} + \mathcal{O}(\delta^2). \tag{21}$$

The energy scales $J_{1A}$, $J_{fn}$, $J_c$ and $J_{an}$ have been constructed with the string degrees of freedom in mind, and we now make the link more explicit. We concentrate in particular on $J_{1A} \gg J_{fn}, J_{an}$, which is the requirement for the existence of spin-liquid behaviour.

A particularly important quantity is the internal free energy per unit length of an isolated string, which depends on $J_{fn}$, $J_c$ and $J_{an}$, and is approximately given by [10, 27],

$$f_{string}(T) \approx 2J_{an} + 4J_{fn} - T \log\left[1 + e^{-\frac{2J_c}{T}}\right]. \tag{22}$$

If string-string interactions are ignored, strings will be present in the system above a temperature $T_{string}$, and this is approximately given by,

$$T_{string} \approx \frac{2J_{an} + 4J_{fn} + J_c}{\log 2}. \tag{23}$$

While a number of approximations have been made in order to arrive at this simple expression, except in the extreme case of $J_c \gg J_{fn}$, it matches well to Monte Carlo simulations of simple models [11].

In reality the strings are not isolated, and the transition temperature and the nature of the correlations in the spin liquid depend on the string-string interactions. These have two main contributions, the first of which is an entropically-driven repulsion associated with the no-crossing constraint, and in the fermion language this maps onto the Pauli exclusion principle. The second is an energetically-driven attraction due to the further-neighbour interactions, which is approximately measured by $J_{fn}$, and in the fermion language it is only this second contribution that counts as an interaction.

If the attractive interaction dominates in the vicinity of $T = T_{string}$ then an array of strings can lower their energy by binding together, and this binding energy results in a first-order transition with $T_1 < T_{string}$. As a result the string density jumps at the transition from $n_{string} \approx 0$ to a finite value. At temperatures just above the transition the string-string interactions remain attractive and the strings loosely bind together, forming a domain-wall-network state. The domain-wall-network state also relies on a positive $J_c$ which penalises changes in direction of the strings. The larger the value of $J_c$ and $J_{fn}$ relative to $T$, the larger the domain size will be. The dominance of attractive interactions in the string picture corresponds to the strong-coupling regime of the fermionic model.

When $T \gg J_{fn}$ the entropically-driven repulsion between strings dominates over the energetically-driven attraction. If $T_{string} \gg J_{fn}$ then the strings repel one another in the critical region, resulting in a second-order transition at $T = T_{string}$. As long as $J_{1A} \gg T_{string}$ then this transition is essentially of the Kasteleyn type, since it is driven by the sudden appearance of strings that mostly wind the system. This type of phase transition is quite different from the more usual Ising transition which is driven by the proliferation of local defects.

Above the second-order transition the string density, $n_{string}$, increases with increasing temperature, and, while the strings fluctuate, they on average form an equally-spaced, grill-like structure due to their mutual repulsion. In the fermionic language this corresponds to weak coupling and a 2D classical equivalent of a Luttinger liquid forms.

When the attractive and repulsive interactions balance, the phase transition is tricritical, and this occurs when $J_{an} \approx J_{fn}$. Just above the transition the string or fermion dispersions are soft, resulting in large fluctuations in the string/fermion density.

The crossover between the spin liquid and paramagnet occurs at $T \approx J_{1A}$ and at this temperature defect triangles become common. As a result strings form short closed or longer floppy loops that typically don't wind the system. If $T_{string} \approx J_{1A}$, then the transition is in the Ising universality class and is driven by the proliferation and growth of local defects, resulting in a direct transition from the ordered phase to the weakly-correlated paramagnet. In the dipolar TLIAF this occurs for $\delta \gtrsim 0.15$.

For the dipolar TLIAF it is possible to approximately determine the appropriate energy scales as $J_{fn} \approx 0.02 J_{1A}$, $J_{an} \approx 9\delta J_{1A}/4$ and $J_c \approx 0.08 J_{1A}$. Here $J_c$ is determined as half the energy cost of an isolated corner, while $J_{fn}$ is determined so as to be consistent both with $T_{string}$ [Eq. 23] and with Monte-Carlo, worm-update simulations, which are found to work best with approximately this value of $J_2^{worm} - 2J_3^{worm}$ (see Section 2.1). Despite the slowly decreasing nature of the dipolar interaction with distance, it can be seen that frustration leads to a value of $J_{fn}$ that is considerably smaller than $J_{1A}$, resulting in a significant window in which the spins are strongly correlated.

An obvious question raised by this analysis is how to further reduce the value of $J_{fn}$ and $J_c$ relative to $J_{1A}$, since this would increase the size of the spin-liquid region and give a cleaner realisation of the Kasteleyn transition. One possibility would be to find systems with local interactions such that $J_1 \gg J_2, J_3 \ldots$, but we are not currently aware of any such systems. A more

realistic option is to change the nature of the long-range interaction such that $J_{ij} \propto |\mathbf{r}_i - \mathbf{r}_j|^{-a}$, where $a = 3$ corresponds to the dipolar case. The possibility of changing $a$ has been realised experimentally using trapped ions that naturally form a triangular lattice, and $a$ was found to be tuneable in the range $0 < a < 3$ [26]. Estimating the relationship between $J_{\mathrm{fn}}$, $J_{\mathrm{c}}$ and $a$ is complicated, due to the competition between the further-neighbour interactions, but it seems most likely that suppression of $J_{\mathrm{fn}}$ would require the further-neighbour interactions to fall off faster than in the dipolar case, and therefore $a > 3$.

Another possibility is to add a small transverse magnetic field. This would tend to act in opposition to the further-neighbour interactions, since quantum fluctuations favour nearest-neighbour-flippable configurations of Ising spins, while the stripe configuration is maximally unflippable. Therefore a transverse field would be likely to reduce the critical temperature by suppressing $J_{\mathrm{fn}}$.

## 6  Conclusion

We have shown that the dipolar TLIAF shows a variety of behaviours, with stripe-ordered, spin-liquid and paramagnetic phases. Furthermore, the nature of the spin-liquid region can be tuned by temperature between a "strongly-coupled" domain-wall network and a "weakly-coupled" string Luttinger liquid, where the strength of the coupling refers to a mapping to a 1D fermionic model. The addition of a small anisotropy allows the nature of the spin liquid to be further tuned, and this in turn changes the critical behaviour from first order to Kasteleyn-like, via a tricritical point with mixed tricritical-Ising and tricritical-Pokrovsky-Talapov characteristics.

We end with the hope that the physics we have described will soon be explored experimentally in artificial spin systems. In such a setting the physics of the isotropic dipolar TLIAF may be even richer, since it is likely that the dynamics will be too local to reliably find the stripe-ordered phase at low temperature, and instead the domain-wall network state will likely freeze to form a glassy state.

## Acknowledgements

We benefited from very useful discussions with Sergey Korshunov at the beginning of this work. We thank Naemi Leo, Oles Sendetskyi and Laura Heyderman for discussions about artificial magnetic systems. We thank Marie Ioannou for discussions concerning the calculation of correlation functions in the Grassmann path integral approach.

**Funding information**   We thank the Swiss National Science Foundation and its SINERGIA network "Mott physics beyond the Heisenberg model" for financial support.The calculations have been performed using the facilities of the Scientific IT and Application Support Center of EPFL.

## A  Defect triangles in the dipolar TLIAF

In this appendix we construct a crude model for the density of defect triangles, $n_{\mathrm{def}}$, in the low-temperature paramagnetic state of the dipolar TLIAF. The aim is to justify the simple functional form of $n_{\mathrm{def}}$ used to fit the Monte Carlo simulations in Fig. 4.

Defect triangles are constrained to occur in pairs, and can be considered to appear on top

of microstates of the constrained manifold (configurations without defect triangles). We make the crude assumption that the energy cost of these defect triangles is only weakly dependent on position and has an average value $E_{\text{def}}$. In this approximation, the total energy due to the defect triangles is given by,

$$E = N_{\text{def}} E_{\text{def}}, \tag{24}$$

where $N_{\text{def}}$ is the number of defect triangles and interactions between defect triangles have been ignored.

The number of ways $N_{\text{def}}$ defect triangles can be placed in the system with $N_{\text{plq}}$ triangular plaquettes is simply given by the binomial coefficient, and therefore the associated partition function is,

$$
\begin{aligned}
\mathcal{Z}_{\text{def}} &= \sum_{N_{\text{def}}=0,2,4\ldots}^{N_{\text{plq}}} \frac{N_{\text{plq}}!}{N_{\text{def}}!(N_{\text{plq}}-N_{\text{def}})!} e^{-\beta E_{\text{def}} N_{\text{def}}} \\
&= \frac{1}{2} \sum_{N_{\text{def}}=0}^{N_{\text{plq}}} \left[1+(-1)^{N_{\text{def}}}\right] \frac{N_{\text{plq}}!}{N_{\text{def}}!(N_{\text{plq}}-N_{\text{def}})!} e^{-\beta E_{\text{def}} N_{\text{def}}} \\
&= \frac{1}{2}\left[\left(1+e^{-\beta E_{\text{def}}}\right)^{N_{\text{plq}}} + \left(1-e^{-\beta E_{\text{def}}}\right)^{N_{\text{plq}}}\right],
\end{aligned} \tag{25}
$$

where $\beta = 1/T$. The average number of defect triangles is given by,

$$\langle N_{\text{def}} \rangle = -\frac{1}{\beta} \frac{\partial \log \mathcal{Z}_{\text{def}}}{\partial E_{\text{def}}} = N_{\text{plq}} \frac{\left(1+e^{-\beta E_{\text{def}}}\right)^{N_{\text{plq}}-1} - \left(1-e^{-\beta E_{\text{def}}}\right)^{N_{\text{plq}}-1}}{\left(1+e^{-\beta E_{\text{def}}}\right)^{N_{\text{plq}}} + \left(1-e^{-\beta E_{\text{def}}}\right)^{N_{\text{plq}}}} e^{-\beta E_{\text{def}}}. \tag{26}$$

In the limit $T \ll E_{\text{def}}/\log N_{\text{plq}}$ the defect triangle density is given by,

$$n_{\text{def}} \propto N_{\text{plq}} e^{-2\beta E_{\text{def}}}, \tag{27}$$

in agreement with an exact calculation for the nearest-neighbour TLIAF. In the opposite limit of $T \gg E_{\text{def}}/\log N_{\text{plq}}$ then for $T \ll E_{\text{def}}$ one finds,

$$n_{\text{def}} \propto e^{-\beta E_{\text{def}}}. \tag{28}$$

While the above analysis is clearly highly simplified with respect to the true situation in the dipolar TLIAF, it suggests that at low temperature and on finite-size systems one should expect the density of defect triangles to obey the relationship,

$$n_{\text{def}} \approx A \frac{\left(1+e^{-\beta E_{\text{def}}}\right)^{N_{\text{plq}}-1} - \left(1-e^{-\beta E_{\text{def}}}\right)^{N_{\text{plq}}-1}}{\left(1+e^{-\beta E_{\text{def}}}\right)^{N_{\text{plq}}} + \left(1-e^{-\beta E_{\text{def}}}\right)^{N_{\text{plq}}}} e^{-\beta E_{\text{def}}}, \tag{29}$$

with $A$ and $E_{\text{def}}$ fitting parameters. The result of fitting this to Monte Carlo simulations is shown in Fig. 4, and we find $E_{\text{def}} = 1.60 J_1$ in the isotropic, dipolar TLIAF. This can be compared with the nearest-neighbour TLIAF, where the energy cost per defect triangle is $2J_1$.

## B  Mappings and winding numbers

There are a number of possible mappings from Ising configurations of the TLIAF to dimer and string representations. Here we review the mappings used in this article and the links between them. In order to do this it is useful to define two different manifolds of Ising configurations, the unconstrained manifold that contains all possible configurations and the constrained manifold that contains only those configurations that are ground states of the nearest-neigbour TLIAF. The constrained manifold is clearly smaller than the unconstrained one, but is itself extensive [2].

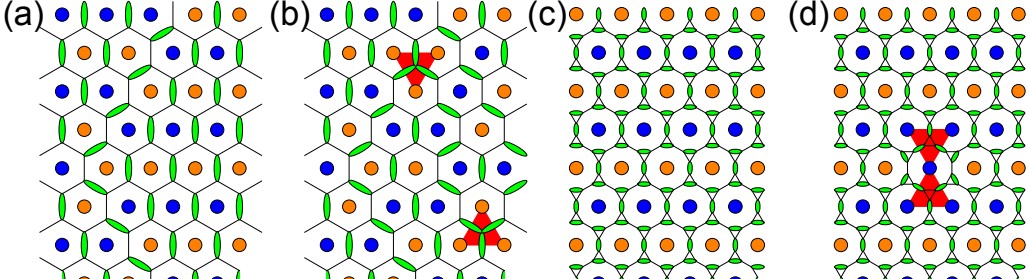

**Figure 13:** Mapping between Ising configurations on the triangular lattice and dimer configurations on the honeycomb and extended honeycomb lattices. (a) There is a correspondence between bonds of the triangular and honeycomb lattices, and this is used to define a honeycomb-lattice dimer model. If the spins are aligned on the triangular-lattice bond, the associated honeycomb bond is covered by a dimer. If the spins are opposite, then the honeycomb bond is empty. In the ground state of the nearest-neighbour TLIAF all honeycomb vertices are covered by one dimer (i.e. the model is hardcore). (b) The presence of defect triangles (coloured red) results in honeycomb sites that are covered with three dimers. (c) and (d) In order to obtain a dimer model that is hardcore for all Ising configurations the honeycomb lattice can be extended by the Fisher construction [56].

## B.1 Mapping to dimer coverings of the dual lattice

One useful mapping is from Ising configurations on the triangular lattice to dimer configurations on the dual honeycomb lattice [46]. We use this when constructing Monte Carlo worm updates [11].

The dual honeycomb lattice is constructed such that its bonds cut exactly one bond of the original triangular lattice (and vice versa), as shown in Fig. 13. If the triangular-lattice bond has two equivalent spins, then the honeycomb-lattice bond is covered by a dimer, while if the spins are inequivalent the honeycomb-lattice bond is left empty. The mapping between spin and dimer configurations is $2 \rightarrow 1$, since the dimer configuration is unaffected by a global flip of all the Ising spins.

Configurations within the constrained manifold (i.e. ground states of the nearest-neighbour TLIAF model) have one ferromagnetic bond per triangle, and therefore the number of dimers is fixed and equal to the number of triangular lattice sites, $N$. It follows that sites on the honeycomb lattice respect the usual dimer model constraint of being covered by exactly one dimer, as shown in Fig. 13(a).

In the unconstrained manifold, for each pair of defect triangles there are two additional dimers, and therefore the number of dimers is not fixed. The honeycomb-lattice site at the centre of a defect triangle is covered by three dimers, and therefore does not respect the usual dimer model constraint (see Fig. 13(b)).

For the unconstrained manifold of Ising configurations an alternative dimer mapping is possible, which is constructed such that the number of dimers is fixed and each site obeys the usual dimer-model constraint of being covered by exactly one dimer [56]. This involves extending the honeycomb lattice such that every original site is replaced by three new sites arranged in a triangle (see Fig. 13). Dimers are then placed on the original honeycomb lattice bonds in the same way as before, leaving a unique way of dimer covering the remaining sites of the extended honeycomb lattice such that every site is covered exactly once.

## B.2 Mapping to string configurations on the dual lattice

The main mapping used throughout the article is onto string configurations on the dual honeycomb lattice [37, 38]. Here we show how this is related to the dimer mapping described above. This proceeds by comparing a given dimer configuration to a reference configuration in which all the dimers are parallel (see Fig. 14). Any honeycomb-lattice bonds on which

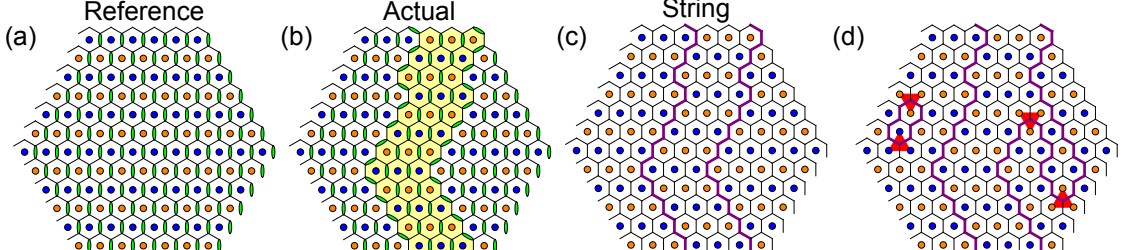

**Figure 14:** Mapping between Ising configurations on the triangular lattice and string configurations on the honeycomb lattice. (a) Reference Ising configuration. (b) A configuration of interest. Spins that differ from the reference configuration are highlighted in yellow. (c) The configuration of interest can be specified (up to a global spin flip) by a set of strings (purple), which measure the difference in dimer covering between the actual and reference configurations. These strings are always non-crossing, and for configurations within the constrained manifold are directed in the sense that they never turn back on themselves, and therefore wind the system. (d) Defect triangles (red) act as sources and sinks of pairs of strings, and allow strings to turn back on themselves.

there is a discrepancy between the actual dimer configuration and the reference configuration is assigned to be part of a string.

The chosen reference configuration consists of alternating horizontal stripes of aligned Ising spins, and this corresponds to all vertical bonds of the honeycomb lattice being covered by dimers (see Fig. 14(b)). This choice of reference configuration results in a number of useful properties of the strings, the most important of which is that strings never touch or cross. For periodic boundary conditions there is the additional constraint that the number of strings crossing an arbitrary reference line that winds the system has to be even, meaning that the string parity is conserved. If the Ising configurations are restricted to be in the constrained manifold the strings are directed, in the sense that they cannot turn back on themselves, and therefore have to wind the system, as shown in Fig. 14(c). In the unconstrained manifold defect triangles act as sources and sinks of pairs of strings, resulting in (non-winding) closed loops of strings as well as strings that turn back on themselves, as shown in Fig. 14(d).

### B.3 Winding number sectors

In the presence of periodic boundary conditions, Ising configurations within the constrained manifold can be labelled by a pair of winding numbers.

One way to define the winding numbers, $\mathbf{W} = (W_1, W_2)$, is to consider a pair of reference lines, as shown in Fig. 15. For each dimer crossing the horizontal part of the reference line the winding number is augmented by $+1$, and for each dimer crossing the angled part of the reference line it is augmented by $-1$. For hexagonal clusters of linear size $L$ with $N = 3L^2$ triangular-lattice sites, the allowed winding number sectors form a triangle with vertices at $\mathbf{W} = (L, L)$, $\mathbf{W} = (0, -L)$ and $\mathbf{W} = (-L, 0)$. Within this triangle, all even values of $W_1$ and $W_2$ are allowed.

In the string picture, the winding number is simply given by,

$$W_1 = L - \text{no. strings crossing ref line 1}$$
$$W_2 = L - \text{no. strings crossing ref line 2}, \tag{30}$$

and it follows that the density of strings in the constrained manifold can be written as,

$$n_{\text{string}} = \frac{2}{3} - \frac{W_1 + W_2}{3L}. \tag{31}$$

The string vacuum is therefore equivalent to the winding number sector $\mathbf{W} = (L, L)$ and the sector $\mathbf{W} = (0, 0)$ corresponds to a density $n_{\text{string}} = 2/3$.

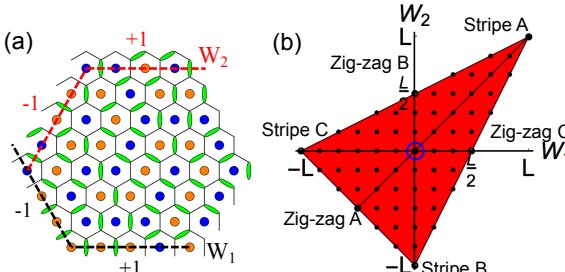

**Figure 15:** Winding numbers sectors of the triangular-lattice Ising antiferromagnet (reproduced from Ref. [11]). (a) A pair of reference lines is defined on hexagonal clusters with periodic boundary conditions, and the dimer crossing of these lines gives the winding number $\mathbf{W} = (W_1, W_2)$. Dimers crossing the horizontal lines increase $W_i$ by 1, while those crossing angled lines decrease $W_i$ by 1. This is simply related to the number of strings crossing the reference lines by Eq. 30. (b) The allowed winding number sectors for an $L = 12$ cluster are shown by black dots. $W_1$ and $W_2$ are both even, and lie within a triangle with vertices at $\mathbf{W} = (L, L)$, $\mathbf{W} = (-L, 0)$ and $\mathbf{W} = (0, -L)$.

The winding numbers split the constrained manifold into topological sectors, in the sense that it is not possible to move between configurations with different winding numbers by making a series of local spin flips. Instead it is necessary to flip clusters of spins that wind the system.

In the unconstrained manifold (defect triangles allowed) $\mathbf{W}$ remains a useful quantity, but is no longer strictly a winding number, since the creation of a pair of defect triangles on the reference line is a local move that alters $\mathbf{W}$. Nevertheless, it remains a useful concept when the defect-triangle density, $n_{\text{def}}$, is low.

## C $\quad J_{1A}$-$J_{1B}$ model with a constrained manifold

In order to isolate and study some of the important features of general TLIAF's, we consider a number of simple models, in which the interactions are local and can be varied at will. The subject of this appendix is the simplest of these models, the TLIAF with anisotropic nearest-neighbour interactions and a constraint forbidding defect triangles. The purpose of studying such a model is to understand the Pokrovsky-Talapov critical behaviour [39, 40] and the correlations within the spin-liquid phase in a simple setting. In terms of the anisotropic, dipolar TLIAF studied in the main text, the ideas will be particularly relevant to the phase transition in the region $\delta_{\text{tri}} < \delta \lesssim 0.1$ and to the correlations in the string-Luttinger liquid phase for $T \ll J_{1A}$.

The solution of this model is already well known due to the fact it can be mapped to free fermions, and was studied by Wannier in the case of isotropic interactions [2], and can be transformed to the Kasteleyn model for anisotropic interactions [46]. The Hamiltonian is given by,

$$\mathcal{H}_{\text{ABB}} = J_{1A} \sum_{\langle ij \rangle_A} \sigma_i \sigma_j + J_{1B} \sum_{\langle ij \rangle_B} \sigma_i \sigma_j + J_{1B} \sum_{\langle ij \rangle_C} \sigma_i \sigma_j, \tag{32}$$

where $\langle ij \rangle_\alpha$ denotes nearest-neighbour bonds in the $\alpha$ direction (see Fig. 1 for the definition of bond directions). An alternative parametrisation can be achieved by writing,

$$J_{1B} = J_{1A} + \delta J, \tag{33}$$

and we consider the case $\delta J > 0$ (equivalently $J_{1A} < J_{1B}$). We also impose the constraint that defect triangles are forbidden, which corresponds to taking the limit $J_{1A}/\delta J \to \infty$.

## C.1  Dimer mapping

$\mathcal{H}_{\text{ABB}}$ [Eq. 32] can be mapped onto the Kasteleyn model of dimer coverings of the honeycomb lattice, which has an exact solution [46]. The mapping from Ising spins on the triangular lattice to dimers on the dual honeycomb lattice is described in Appendix B.1, and the energy of a dimer configuration is given by,

$$E_{\text{ABB}} = -\frac{1}{3}(J_{1A} + 2J_{1B})N_{\text{bond}} + 2J_{1A}N_{\text{dim}}^{A} + 2J_{1B}(N_{\text{dim}}^{B} + N_{\text{dim}}^{C}), \tag{34}$$

where $N_{\text{bond}} = 3N$ is the total number of bonds (this is the same for the triangular and dual honeycomb lattices) and $N_{\text{dim}}^{\alpha}$ is the number of dimers covering $\alpha$-type bonds. Since defect triangles are forbidden, the total number of dimers is fixed as $N_{\text{dim}}^{A} + N_{\text{dim}}^{B} + N_{\text{dim}}^{C} = N_{\text{bond}}/3$. In the ground state $N_{\text{dim}}^{A} = N_{\text{bond}}/3$ and $N_{\text{dim}}^{B} = N_{\text{dim}}^{C} = 0$, and therefore the energy of a given configuration relative to the ground state energy is,

$$\Delta E_{\text{ABB}} = 2\,\delta J(N_{\text{dim}}^{B} + N_{\text{dim}}^{C}). \tag{35}$$

It follows that the partition function can be written, up to a configuration independent prefactor, as,

$$\mathcal{Z}_{\text{ABB}} \propto \mathcal{Z}_{\text{hon}} = \sum_{\text{dimer cov}} z^{N_{\text{dim}}^{B} + N_{\text{dim}}^{C}}, \tag{36}$$

where the sum is over all dimer coverings of the honeycomb lattice. It can be seen from Eq. 35 that the weight associated with dimer covering a B or C bond is given by,

$$z = e^{-\frac{2\delta J}{T}}. \tag{37}$$

## C.2  Evaluation of the partition function

It has been known how to evaluate partition functions of the type $\mathcal{Z}_{\text{hon}}$ [Eq. 36] for many years [6, 46]. Here we will briefly sketch the solution, since it will prove a useful basis from which to consider more complicated models.

The starting point is to introduce a real, anticommuting Grassmann variable at each site of the honeycomb lattice [6, 57, 58]. These variables obey the usual rules: $a_i a_j = -a_j a_i$, $\int da_i = 0$ and $\int da_i a_i = 1$. Since the honeycomb lattice has a 2-site basis, it is useful to label Grassmann variables as $a$ and $b$ on the two sublattices, and the partition function is therefore given by,

$$\mathcal{Z}_{\text{hon}} = \int \prod_i da_i db_i\, e^{\mathcal{S}_2[a,b]} = \det K, \quad \mathcal{S}_2[a,b] = \sum_{ij} a_i K_{ij} b_j, \tag{38}$$

where $i$ labels unit cells and $\mathcal{S}_2[a,b]$ is the Kastelyn action. Here $K$ is a signed adjacency matrix, known as the Kasteleyn matrix [46], and contains the weights $z$ [Eq. 37]. The reason for the appearance of $\det K$ rather than the more usual Pfaffian is that the matrix connects sites on different sublattices, but not those on the same sublattice.

To simplify the geometry the honeycomb lattice is distorted into the brick lattice, as shown in Fig. 16. Bonds are assigned a direction in accordance with the Kasteleyn theorem, which states that transition cycles should have an odd number of arrows in each sense [46]. The bond weights are assigned according to Eq. 36, with weight 1 on A bonds and weight $z$ on B and C bonds. It follows that the action can be written as,

$$\mathcal{S}_2[a,b] = \sum_i \left( z a_i b_{i+\hat{e}_y} + z b_{i-\hat{e}_x} a_i + b_i a_i \right), \tag{39}$$

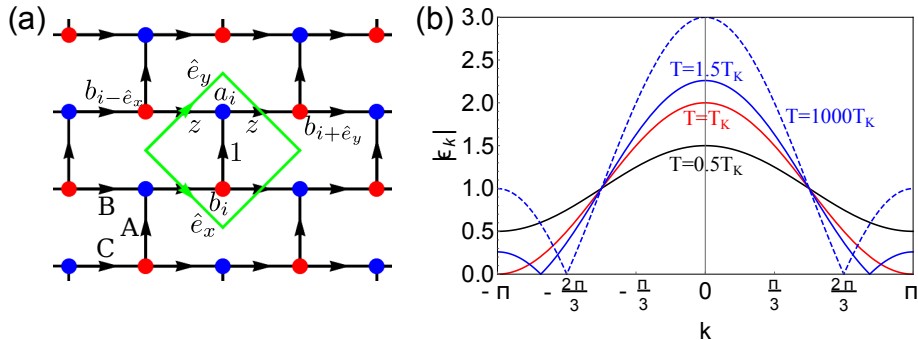

Figure 16: The brick lattice and Kasteleyn-matrix spectrum, $|\epsilon_{\mathbf{k}}|$ [Eq. 42], for $\mathcal{H}_{\text{ABB}}$ [Eq. 32] in the constrained manifold. (a) Bond directions (black arrows) are chosen so as to respect Kasteleyn's theorem [46], and bond weights are chosen to be $z$ on B and C bonds and 1 on A bonds, in accordance with $\mathcal{Z}_{\text{hon}}$ [Eq. 36]. The $i$th unit cell (green) contains two sites with associated Grassmann variables $a_i$ (blue) and $b_i$ (red). The translation vectors of the unit cell are $\hat{e}_x$ and $\hat{e}_y$, and these are taken to be unit length. (b) The spectrum $|\epsilon_{\mathbf{k}}|$ is shown along the path $\mathbf{k} = (k, k + \pi)$. For $T < T_K$ (black) the spectrum is gapped at all $\mathbf{k}$, and this corresponds to the stripe-ordered phase. At $T = T_K$ (red) the gap closes at $\mathbf{k} = (\pi, 0)$ and a Kasteleyn transition occurs. For $T > T_K$ (blue) the gapless point migrates from $\mathbf{k} = (\pi, 0)$ to $\mathbf{k} = (2\pi/3, -\pi/3)$ with increasing temperature, and the location of this gap is related to the density of strings, $n_{\text{string}}$ [Eq. 45].

where the coordinate system is defined by the unit vectors $\hat{e}_x$ and $\hat{e}_y$, as shown in Fig. 16. The action is simply diagonalised by taking the Fourier Transform,

$$a_i = \frac{1}{\sqrt{N}} \sum_{\mathbf{k}} a_{\mathbf{k}} e^{i\mathbf{k}\cdot\mathbf{r}_i} e^{-i\frac{k_x - k_y}{2}}, \quad b_i = \frac{1}{\sqrt{N}} \sum_{\mathbf{k}} b_{\mathbf{k}} e^{i\mathbf{k}\cdot\mathbf{r}_i}, \tag{40}$$

to give,

$$\mathcal{S}_2[a, b] = \sum_{\mathbf{k}} \epsilon_{\mathbf{k}} a_{\mathbf{k}} b_{-\mathbf{k}}, \qquad \epsilon_{\mathbf{k}} = -2iz \sin\left[\frac{k_x + k_y}{2}\right] - e^{-i\frac{k_x - k_y}{2}}. \tag{41}$$

Finally the partition function can be evaluated as,

$$\mathcal{Z}_{\text{hon}} = \prod_{\mathbf{k}} \epsilon_{\mathbf{k}} = \prod_{\mathbf{k}} \sqrt{\epsilon_{\mathbf{k}} \epsilon_{-\mathbf{k}}} = \prod_{\mathbf{k}} |\epsilon_{\mathbf{k}}|,$$
$$|\epsilon_{\mathbf{k}}| = \sqrt{1 + 2z(\cos k_x - \cos k_y) + 2z^2(1 - \cos[k_x + k_y])}, \tag{42}$$

where $\epsilon_{\mathbf{k}}^* = \epsilon_{-\mathbf{k}}$ has been used.

## C.3  Physical properties

In order to understand better the physical properties of $\mathcal{H}_{\text{ABB}}$ [Eq. 32] it is useful to notice that the free energy, which is given by,

$$\mathcal{F}_{\text{hon}} = -T \log \mathcal{Z}_{\text{hon}} = -T \sum_{\mathbf{k}} \log|\epsilon_{\mathbf{k}}|, \tag{43}$$

is typically dominated by the minimal values of $|\epsilon_{\mathbf{k}}|$. The physical characteristcs of the system are therefore determined predominantly by the "low-energy" part of $|\epsilon_{\mathbf{k}}|$.

At low temperature, the spectrum of $|\epsilon_{\mathbf{k}}|$ is gapped at all $\mathbf{k}$, as shown in Fig. 16, and this corresponds to the stripe-ordered state. The gap closes at $\mathbf{k} = (\pi, 0)$ at the temperature,

$$2z(T_K) = 1, \quad T_K = \frac{2\delta J}{\log 2}, \tag{44}$$

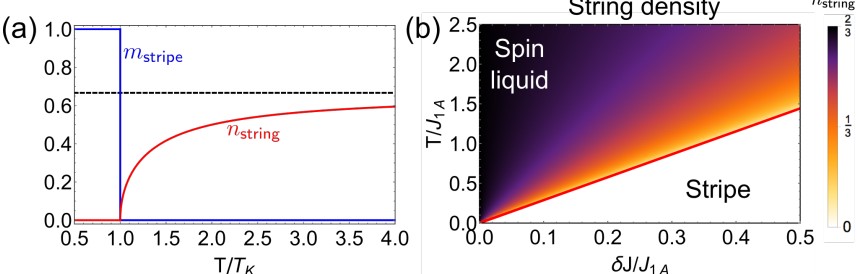

Figure 17: The string density, $n_{\text{string}}$ [red, Eq. 49], and the phase diagram for $\mathcal{H}_{\text{ABB}}$ [Eq. 32] in the constrained manifold. (a) Comparison between $n_{\text{string}}$ [red, Eq. 49] and the stripe order parameter, $m_{\text{stripe}}$ [blue, Eq. 2]. $m_{\text{stripe}}$ shows a step-like behaviour at the critical point, $T = T_K$, while $n_{\text{string}}$ has Pokrovsky-Talapov critical behaviour, with $n_{\text{string}} \propto (T - T_K)^\beta$ and $\beta = 1/2$ for small $T - T_K > 0$. In the limit $T \to \infty$ it saturates at $n_{\text{string}} = 2/3$. (b) The phase diagram showing stripe and spin-liquid phases separated by a second-order transition at $T = T_K$ [Eq. 44], and overlaid with the string density, $n_{\text{string}}$ [Eq. 45].

and this corresponds to the temperature at which the free energy of strings goes to zero.

At $T = T_K$ strings condense into the system, and there is a Kasteleyn transition out of the stripe-ordered phase and into the spin liquid. This is second order due to the non-crossing constraint of the strings, which results in an entropically-driven string-string repulsion. The transition is in the Pokrovsky-Talapov universality class [39, 40].

For all $T > T_K$ the spectrum of $|\epsilon_{\mathbf{k}}|$ is gapless. The position of the gapless point moves from $\mathbf{k} = (\pi, 0)$ at $T = T_K$ to $\mathbf{k} = (2\pi/3, -\pi/3)$ at $T \to \infty$ and the position of this point is simply related to the string density, which smoothly increases with increasing temperature. We show below that the gaplessness of the spectrum is associated with algebraic decay of the spin-spin correlations [4, 5].

In order to detect the transition between the stripe-ordered phase and the paramagnet, one possibility is to measure the local stripe order parameter $m_{\text{stripe}}$ [Eq. 2]. However, this is somewhat unsatisfactory, since $m_{\text{stripe}} = 1$ in the ordered phase, and there is a discontinuous jump to $m_{\text{stripe}} = 0$ at $T = T_K$ (see Fig. 17). Thus $m_{\text{stripe}}$ does not show critical behaviour, and this is due to the fact that the transition is not driven by the proliferation of local defects, but by strings that wind the system.

A more useful physical quantity is the density of strings, $n_{\text{string}}$, and this does show critical behaviour. However, unlike a conventional order parameter, $n_{\text{string}} = 0$ in the ordered phase, and only takes a finite value for $T > T_K$. It can most simply be calculated in terms of dimer densities, according to,

$$n_{\text{string}} = \frac{1}{2} + \frac{\langle N^B_{\text{dim}} + N^C_{\text{dim}} - N^A_{\text{dim}} \rangle}{2N}, \tag{45}$$

where the normalisation is such that $0 \leq n_{\text{string}} \leq 1$. In the case of the constrained manifold, the total number of dimers is fixed (see Appendix C.1) and this leads to the simplified expression [37],

$$n_{\text{string}} = \frac{\langle N^B_{\text{dim}} + N^C_{\text{dim}} \rangle}{N} = \frac{z}{N} \frac{\partial \log \mathcal{Z}_{\text{hon}}}{\partial z} = \frac{1}{N} \sum_{\mathbf{k}} \frac{z}{|\epsilon_{\mathbf{k}}|} \frac{\partial |\epsilon_{\mathbf{k}}|}{\partial z}, \tag{46}$$

where the second equality follows from the expression for $\mathcal{Z}_{\text{hon}}$ [Eq. 36].

When working in the constrained manifold, the string density can be calculated in a simple closed form. Substituting the expression for $|\epsilon_{\mathbf{k}}|$ [Eq. 42] into $n_{\text{string}}$ [Eq. 46], making the change of variables $p_x = (k_x + k_y - \pi)/2$ and $p_y = (k_x - k_y - \pi)/2$, and taking the thermodynamic

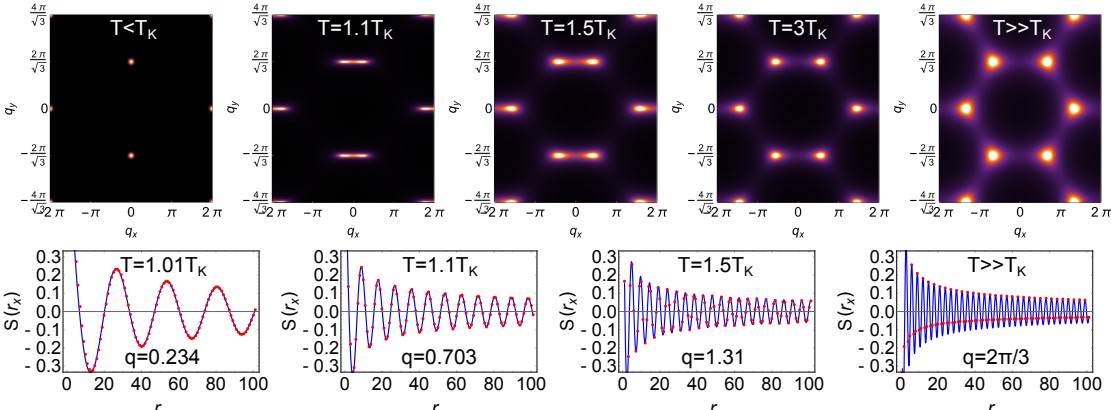

Figure 18: The structure factor of $\mathcal{H}_{ABB}$ [Eq. 32]. (Top) The reciprocal-space structure factor $S(\mathbf{q})$ at various temperatures, calculated by Monte Carlo simulation for a hexagonal cluster with $L = 72$. For $T < T_K$ there are Bragg peaks at $\mathbf{q}_{stripe} = (0, 2\pi/\sqrt{3})$, and these have been artificially broadened for clarity. For $T > T_K$ there are divergencies at the wavevectors $\mathbf{q}_{string}(T) = (\pm\pi n_{string}, 2\pi/\sqrt{3})$ that go as $S(\mathbf{q}_{string} + \delta\mathbf{q}) \propto |\delta\mathbf{q}|^{-3/2}$. For $T \gg T_K$ the physics of the isotropic TLIAF is recovered, with $n_{string} = 2/3$ and $\mathbf{q}_{string}(T) = (\pm 2\pi/3, 2\pi/\sqrt{3})$. (Bottom) The real-space structure factor, $S(r_x)$ [Eq. 4], in the direction perpendicular to the strings (i.e. parallel to A bonds), calculated in the thermodynamic limit using the Grassmann path integral approach (see Appendix F). Red dots show calculations and the blue line shows the best fit to $S(r_x) = A\cos(\pi n_{string} r_x)/\sqrt{r_x}$, where $A$ is the only free parameter.

limit results in,

$$
\begin{aligned}
n_{string} &= \frac{1}{\pi^2} \int_0^\pi dp_x \int_0^\pi dp_y \frac{4z^2 \cos^2 p_x - 2z \cos p_x \cos p_y}{1 - 4z \cos p_x \cos p_y + 4z^2 \cos^2 p_x} \\
&= \frac{1}{\pi^2} \int_0^\pi dp_x \frac{u}{2} \frac{\partial}{\partial u} \int_0^\pi dp_y \log[1 - 2u \cos p_y + u^2],
\end{aligned} \tag{47}
$$

where $u = 2z \cos p_x$. The integral over $p_y$ is tablulated and given by [2],

$$
\int_0^\pi dp_y \log[1 - 2u \cos p_y + u^2] = 2\pi[1 - D(u)] \log u, \qquad D(u) = \begin{cases} 1 & |u| < 1 \\ 0 & |u| > 1 \end{cases}. \tag{48}
$$

As a result one finds,

$$
n_{string} = \begin{cases} \frac{2}{\pi} \arccos\left[\frac{1}{2z}\right] & z > \frac{1}{2} \\ 0 & z < \frac{1}{2} \end{cases}, \tag{49}
$$

and this is plotted in Fig. 17. It can be seen that for $T \to \infty$ ($z \to 1$) the string density saturates at $n_{string} = 2/3$, while for $T \approx T_K$ it shows Pokrovsky-Talapov critical behaviour with $n_{string} \propto (T - T_K)^\beta$ and $\beta = 1/2$.

## C.4    Correlations

In order to better understand the correlations it is useful to study the spin-spin structure factor [defined in Eq. 4]. When doing this care should be taken not to confuse $\mathbf{q}$, which denotes a reciprocal vector in the Brillouin zone of the triangular lattice, with $\mathbf{k}$, which lives in the Brillouin zone of the brick lattice.

The structure factor can be calculated in the thermodynamic limit using the Grassmann path integral approach, following the general method proposed in Ref. [59]. For $\delta J = 0$ this reproduces the results of Ref. [4, 5]. A detailed summary of the calculation is given in

Appendix F, both for pairs of spins separated by an arbitrary number of A bonds (i.e. in the direction perpendicular to the strings, denoted $r_x$) and for separations orthogonal to A bonds (i.e. in the direction parallel to the strings, denoted $r_y$). In both cases the structure factor can be written as the determinant of a Toeplitz matrix, whose dimension is proportional to the separation between the spins. Exact expressions can be written for the matrix elements, but we find it necessary to calculate the determinant numerically.

In the case of isotropic interactions the structure factor takes the asymptotic form [4, 5],

$$S(\mathbf{r}) \propto \frac{\cos \mathbf{q} \cdot \mathbf{r}}{\sqrt{|\mathbf{r}|}}, \tag{50}$$

where $\mathbf{q} = (\pm 2\pi/3, 2\pi/\sqrt{3})$, as can be seen in Fig. 18. The algebraic decay of correlations shows that the $T = 0$ nearest-neighbour TLIAF is critical, and is on the verge of forming 3-sublattice order. The combination of long-range disorder and local correlation means that the system forms a classical spin liquid.

For $\delta J \neq 0$ and $T > T_K$ the structure factor retains the long-distance functional form given in Eq. 50, but the wavevector becomes temperature dependent and is given by $\mathbf{q} = \pm \mathbf{q}_{\text{string}}(T) = (\pm \pi n_{\text{string}}(T), 2\pi/\sqrt{3})$. This is clearly physically sensible, since the strings separate regions in which Ising spins have opposite sign, and the oscillation of the correlation function in the direction perpendicular to the strings should therefore have a period given by the average string separation. Some typical examples are shown in Fig. 18. Also shown is $S(\mathbf{q})$, which has pairs of algebraically diverging peaks at $\mathbf{q} = \pm \mathbf{q}_{\text{string}}(T)$. In the vicinity of these peaks $S(\mathbf{q}_{\text{string}} + \delta \mathbf{q}) \propto |\delta \mathbf{q}|^{-3/2}$ in agreement with the algebraic decay of $S(\mathbf{r})$ [Eq. 50]. It can be seen that the critical nature of the correlations is not broken by a non-zero $\delta J$ as long as $T > T_K$, and this is due to the constraint forbidding defect triangles.

Having stated that the structure factor has the functional form given in Eq. 50 in the long distance limit, it is useful to be more precise over what counts as long distance. This has been considered in the closely related field of adsorption of a gas onto a substrate, where there exist domain walls with similar properties to the strings of the TLIAF [47]. A correlation length can be defined beyond which the long-distance algebraic correlation function given in Eq. 50 applies. In the direction perpendicular to the strings it is intuitively obvious that this is given by the average string-string separation, and therefore $\zeta_\perp \sim 1/n_{\text{string}}$. In the direction parallel to the strings it can be argued that $\zeta_\parallel \sim 1/n_{\text{string}}^2$ [47]. In the case of $\delta J = 0$ (or $T \gg \delta J$), where $n_{\text{string}} = 2/3$, the correlation lengths, $\zeta_\perp$ and $\zeta_\parallel$, are not much longer than a single lattice spacing, and the long distance asymptotic form of the structure factor is recovered for spins separated by only a few lattice spacings. On the other hand, for $\delta J \neq 0$ and $T \approx T_K$ the string density is low, and the correlation lengths become very large, especially in the direction parallel to the strings. For $T \to T_K$ an expansion of $n_{\text{string}}$ [Eq. 49] shows that the correlation lengths diverge as $\zeta_\perp \sim (T - T_K)^{-\nu_\perp}$ and $\zeta_\parallel \sim (T - T_K)^{-\nu_\parallel}$, with $\nu_\perp = 1/2$ and $\nu_\parallel = 1$.

For $T < T_K$ the spin structure factor clearly does not follow the functional form of Eq. 50. Instead it is constant in real space, since the stripe state admits no fluctuations, and has Bragg peaks in reciprocal space. On cooling through $T = T_K$ the pair of algebraically-diverging peaks in $S(\mathbf{q})$ coalesce to form a single Bragg peak at $\mathbf{q} = \mathbf{q}_{\text{stripe}} = (0, 2\pi/\sqrt{3})$ (see Fig. 18).

## C.5 Mapping to 1D quantum model

A slightly different perspective on the nearest-neighbour TLIAF is achieved by making a mapping onto a 1D quantum model of spinless fermions. The idea is that the strings can be viewed as the worldlines of spinless fermions, and the spatial direction parallel to the strings interpreted as imaginary time. The non-crossing constraint of the strings corresponds to the Pauli exclusion principle, and periodic boundary conditions in the 2D classical model enforce peri-

odicity in imaginary time in the 1D quantum model. This type of mapping has been frequently used for related 2D classical models with non-crossing domain walls [39–44].

We show in Appendix G that $\mathcal{H}_{ABB}$ [Eq. 32] maps exactly onto the 1D quantum model,

$$\mathcal{H}_{1D} = \sum_i \left[ -\mu c_i^\dagger c_i + t \left( c_i^\dagger c_{i+1} + c_{i+1}^\dagger c_i \right) \right], \qquad t = z^2, \quad \mu = 2z^2 - 1. \qquad (51)$$

The mapping demonstrates that the Grassmann variables, $a_i$ and $b_i$, describe coherent states of fermions/strings (for details see Appendix G).

$\mathcal{H}_{1D}$ [Eq. 51] is simply diagonalised by Fourier transform, giving,

$$\mathcal{H}_{1D} = \sum_k \omega_k c_k^\dagger c_k, \quad \omega_k = 2t \cos k - \mu, \qquad (52)$$

and the phase diagram of the fermion model can be matched to that of the nearest-neighbour TLIAF. For $\mu/2t < -1$ there are no fermions in the system and this is analagous to the stripe phase in which there are no strings. At $\mu/2t = -1$ there is a phase transition due to the minima of the fermion band touching zero, and for $\mu/2t > -1$ the fermion density, $n_f$, is given by, $n_f = 1 - \arccos[\mu/2t]/\pi$, which can be seen to be exactly equal to $n_{string}$ [Eq. 49]. According to the mapping given in Eq. 51, $\mu$ and $t$ are not independent parameters, and the maximum value of their ratio is given by $\mu/2t = 1/2$. This corresponds to $z = 1$ (equivalently $T \to \infty$) and at this point $n_f = n_{string} = 2/3$ as expected. Other physical quantities, such as the heat capacity or the spin-spin structure factor can be calculated within the 1D fermion picture, and in some cases this simplifies the procedure.

It should be noted that if the 2D classical model has periodic boundary conditions, then the number of strings in the system is constrained to be even. The 1D quantum model is therefore restricted to the even-parity fermion subsector. If the 2D model is instead defined on a cylinder with the periodic direction parallel to the strings, then this restriction is lifted.

One of the utilities of the 2D classical to 1D quantum mapping is that for more complicated 2D models with longer range interactions it provides a good starting point for phenomenologial theories.

# D   $J_{1A}$-$J_{1B}$ model with an unconstrained manifold

The next model we consider is the TLIAF with anisotropic nearest-neighbour interactions but now with defect triangles allowed (i.e. in the unconstrained manifold). The point is to better understand the crossover between the spin liquid and the weakly-correlated paramagnet and the crossover between Ising and Pokrovsky-Talapov criticality, which are both also features of the dipolar TLIAF. The Hamiltonian $\mathcal{H}_{ABB}$ [Eq. 32] is the same as in Appendix C, except for the important difference that the manifold of Ising configurations is unconstrained, meaning defect triangles are allowed.

## D.1   Dimer mapping

$\mathcal{H}_{ABB}$ [Eq. 32] with an unconstrained manifold can be mapped onto a dimer model on the honeycomb lattice, but there is no longer a hardcore constraint, since vertices at the centre of defect triangles are covered by 3 dimers. Since the Grassmann path integral approach to determining the partition function requires the dimers to obey a hardcore constraint, it is necessary to instead consider the mapping onto a dimer model on the extended honeycomb lattice (described in Appendix B.1 and Fig. 13). This type of mapping was suggested in a more general context in [56], and makes possible an exact evaluation of the partition function.

The dimers can be categorised as those covering A, B or C bonds of the original triangular lattice (see Fig. 1 for bond labelling) or "extra" dimers, covering the bonds introduced in the act of extending the honeycomb lattice. The total number of dimers is fixed and given by,

$$N_{\text{dim}}^{\text{A}} + N_{\text{dim}}^{\text{B}} + N_{\text{dim}}^{\text{C}} + N_{\text{dim}}^{\text{ext}} = N_{\text{bond}}, \tag{53}$$

where $N_{\text{bond}} = 3N$ refers to the number of bonds of the original triangular lattice and $N_{\text{dim}}^{\text{ext}}$ is the number of dimers on "extra" bonds. The energy of a given configuration relative to that of the ground state can be written as,

$$\Delta E_{\text{ABB}} = \frac{4}{3} J_{1\text{A}} N_{\text{bond}} + 2\delta J (N_{\text{dim}}^{\text{B}} + N_{\text{dim}}^{\text{C}}) - 2 J_{1\text{A}} N_{\text{dim}}^{\text{ext}}, \tag{54}$$

and therefore,

$$\mathcal{Z}_{\text{ABB}} \propto \mathcal{Z}_{\text{exhon}} = z_{\text{A}}^{2N} \sum_{\text{dimer cov}} z^{N_{\text{dim}}^{\text{B}} + N_{\text{dim}}^{\text{C}}} z_{\text{A}}^{-N_{\text{dim}}^{\text{ext}}}, \tag{55}$$

where the sum is over all dimer coverings of the extended honeycomb lattice,

$$z_{\text{A}} = e^{-\frac{2J_{1\text{A}}}{T}}, \quad z_{\text{B}} = e^{-\frac{2J_{1\text{B}}}{T}}, \quad z = \frac{z_{\text{B}}}{z_{\text{A}}}, \tag{56}$$

and the factor $z_{\text{A}}^{2N}$ ensures that $\mathcal{Z}_{\text{exhon}}$ is equal to $\mathcal{Z}_{\text{hon}}$ [Eq. 36] in the limit where $z_{\text{A}} \to 0$ and $z_{\text{B}} \to 0$ with $z$ finite (i.e. the condition for being in the constrained manifold of Ising configurations).

## D.2 Evaluation of the partition function

The evaluation of the partition function proceeds as in Appendix C, with the main difference being that there are 6 rather than 2 lattice sites in the unit cell (see also Ref. [60] for slightly different way of evaluating the partition function).

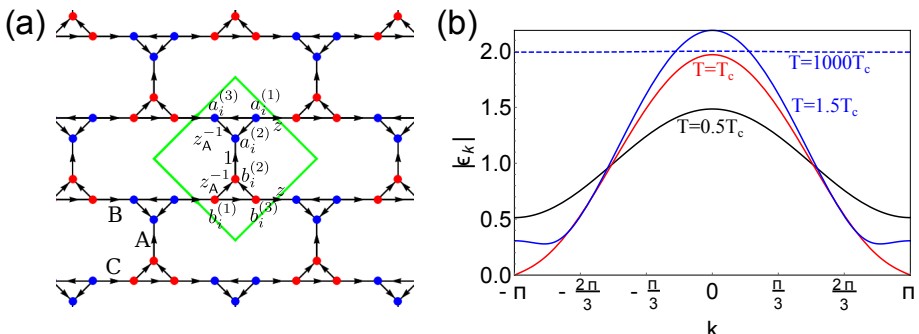

Figure 19: The extended brick lattice and spectrum $|\epsilon_{\mathbf{k}}|$ [Eq. 60] used to calculate the partition function of $\mathcal{H}_{\text{ABB}}$ [Eq. 32] in the unconstrained manifold. (a) Bond directions (black arrows) are chosen so as to respect Kasteleyn's theorem [46], and bond weights are chosen to be $z$ on B and C bonds, 1 on A bonds and $z_{\text{A}}^{-1}$ on "extra" bonds, in accordance with $\mathcal{Z}_{\text{exhon}}$ [Eq. 55]. The $i$th unit cell (green) contains 6 sites with associated Grassmann variables $a_i^{(1)}$, $a_i^{(2)}$ and $a_i^{(3)}$ (blue) and $b_i^{(1)}$, $b_i^{(2)}$ and $b_i^{(3)}$ (red). The translation vectors of the unit cell are $\hat{e}_x$ and $\hat{e}_y$ (see Fig. 16), and these are taken to be unit length. (b) The spectrum $|\epsilon_{\mathbf{k}}|$ [Eq. 60] along the path $\mathbf{k} = (k, k + \pi)$ for $J_{1\text{A}} = 1$ and $J_{1\text{B}} = 1.5$. For $T < T_c$ (black) the spectrum is gapped at all $\mathbf{k}$, and this corresponds to the stripe-ordered phase. At $T = T_c$ (red) the gap closes at $\mathbf{k} = (\pi, 0)$ and an Ising transition occurs. For $T > T_c$ (blue) the gap reopens. For $T_c < T < T_{\text{ls}}$ the minimum of $|\epsilon_{\mathbf{k}}|$ is at $\mathbf{k} = (\pi, 0)$. For $T > T_{\text{ls}}$ the minimum migrates away from $\mathbf{k} = (\pi, 0)$. In the limit $T \to \infty$ the spectrum becomes flat.

Each site of the extended honeycomb lattice is assigned a real Grassmann variable, as shown in Fig. 19, and these are labelled $a_i^l$ and $b_i^l$ where $i$ labels the unit cell, and $l \in \{1, 2, 3\}$

labels sites within the unit cell. Bond weights are $z$ on B and C bonds, 1 on A bonds and $z_A^{-1}$ on "extra" bonds, in accordance with $\mathcal{Z}_{\text{exhon}}$ [Eq. 55]. The partition function is given by,

$$\mathcal{Z}_{\text{exhon}} = z_A^{2N} \int \prod_{i,l} da_i^{(l)} db_i^{(l)} e^{S_2}, \tag{57}$$

where,

$$\mathcal{S}_2 = \sum_i \Bigg[ b_i^{(2)} a_i^{(2)} + z \left( b_i^{(3)} a_{i+\hat{e}_x}^{(3)} + a_i^{(1)} b_{i+\hat{e}_y}^{(1)} \right) \\ + z_A^{-1} \left( b_i^{(1)} b_i^{(3)} + b_i^{(1)} b_i^{(2)} + b_i^{(3)} b_i^{(2)} + a_i^{(1)} a_i^{(3)} + a_i^{(1)} a_i^{(2)} + a_i^{(3)} a_i^{(2)} \right) \Bigg]. \tag{58}$$

This can be diagonalised by taking the Fourier Transform of the Grassmann variables (see Eq. 40), resulting in,

$$\mathcal{S}_2 = \sum_{\mathbf{k}} \Bigg[ b_{-\mathbf{k}}^{(2)} a_{\mathbf{k}}^{(2)} e^{-i\frac{k_x - k_y}{4}} + z \left( b_{-\mathbf{k}}^{(3)} a_{\mathbf{k}}^{(3)} + a_{-\mathbf{k}}^{(1)} b_{\mathbf{k}}^{(1)} \right) e^{i\frac{k_x + k_y}{4}} \\ + z_A^{-1} \left( b_{-\mathbf{k}}^{(1)} b_{\mathbf{k}}^{(3)} e^{i\frac{k_x + k_y}{4}} + b_{-\mathbf{k}}^{(1)} b_{\mathbf{k}}^{(2)} e^{i\frac{k_y}{4}} + b_{-\mathbf{k}}^{(3)} b_{\mathbf{k}}^{(2)} e^{-i\frac{k_x}{4}} \\ + a_{\mathbf{k}}^{(1)} a_{-\mathbf{k}}^{(3)} e^{i\frac{k_x + k_y}{4}} + a_{\mathbf{k}}^{(1)} a_{-\mathbf{k}}^{(2)} e^{i\frac{k_y}{4}} + a_{\mathbf{k}}^{(3)} a_{-\mathbf{k}}^{(2)} e^{-i\frac{k_x}{4}} \right) \Bigg]. \tag{59}$$

After rewriting the action as a matrix equation, taking the Pfaffian of the matrix and absorbing the $z_A^{2N}$ factor, one can show that,

$$\mathcal{Z}_{\text{exhon}} = \prod_{\mathbf{k}} |\epsilon_{\mathbf{k}}|, \quad |\epsilon_{\mathbf{k}}| = \Big\{ 1 + 2z(\cos k_x - \cos k_y) + 2z^2(1 - \cos[k_x + k_y])$$

$$+ 2z_B^2 \cos[k_x + k_y] - 2z_B^2 z(\cos k_x - \cos k_y) + z_B^4 \Big\}^{\frac{1}{2}}. \tag{60}$$

It can be seen that this reduces to the $|\epsilon_{\mathbf{k}}|$ of the constrained manifold [Eq. 42] when the limit $z_A \to 0$ and $z_B \to 0$ is taken such that $z$ remains finite (equivalently $J_{1A} \to \infty$ and $J_{1B} \to \infty$ while $\delta J$ remains finite). It can also easily be checked that in the $T \to \infty$ limit the entropy per site is $S/N = \log 2$ as expected.

## D.3 Physical properties

The spectrum $|\epsilon_{\mathbf{k}}|$ [Eq. 60], which is is shown in Fig. 19, determines the physical properties of $\mathcal{H}_{\text{ABB}}$ [Eq. 32]. As in the case of the constrained manifold there is a phase transition between an ordered and disordered phase. However, we label the transition temperature $T_c$ rather than $T_K$, since it is not technically a Kasteleyn transition, as will be explained below.

For $T < T_c$ the spectrum is gapped at all $\mathbf{k}$, and this corresponds to the stripe-ordered phase. The main difference from the case of the constrained manifold is that local fluctuations involving the creation of pairs of defect triangles are possible, though, depending on the value of $T_c$, they can be highly suppressed.

There is a phase transition at $T = T_c$ associated with the closing of the gap in $|\epsilon_{\mathbf{k}}|$, and this occurs at $\mathbf{k} = (\pi, 0)$. It can be seen from Eq. 60 that this requires,

$$1 - 2z(T_c) - z_B(T_c)^2 = 0, \tag{61}$$

and the solution of this equation gives the critical temperature.

For $T > T_c$ a gap reopens in $|\epsilon_{\mathbf{k}}|$, and this signifies that correlations are exponential in the paramagnetic state [47].

In order to investigate the nature of the phase transition, it is natural to define a second temperature, $T_{\text{ls}}$, such that in the temperature range $T_{\text{c}} < T < T_{\text{ls}}$ the minimum of $|\epsilon_{\mathbf{k}}|$ is at $\mathbf{k} = (\pi, 0)$, while for $T > T_{\text{ls}}$ the minimum of $|\epsilon_{\mathbf{k}}|$ is at a temperature-dependent incommensurate wavevector. $T_{\text{ls}}$ can be determined from the equation,

$$(1-2z)^2 + 4\frac{z_{\text{B}}^2}{z} - 10z_{\text{B}}^2 + 4z_{\text{B}}^2 z + 4\frac{z_{\text{B}}^4}{z^2} - 4\frac{z_{\text{B}}^4}{z} + z_{\text{B}}^4 \Bigg|_{T \to T_{\text{ls}}} = 0, \qquad (62)$$

and it can be seen from Eq. 60 that for $T_{\text{c}} < T < T_{\text{ls}}$ the gap is given by, $\min|\epsilon_{\mathbf{k}}| = |1-2z-z_{\text{B}}^2|$.

After setting $T = T_{\text{c}} + \delta T$, with $\delta T \ll T_{\text{c}}$ and $\delta T < T_{\text{ls}} - T_{\text{c}}$, one can show that the gap goes as $\min|\epsilon_{\mathbf{k}}| \propto \delta T$. Taking the correlation length to be inversely proportional to the gap, $\xi \propto 1/\min|\epsilon_{\mathbf{k}}|$, results in $\xi \propto \delta T^{-\nu}$ with $\nu = 1$, and this is typical of a 2D Ising transition [43]. Therefore Ising critical exponents are realised in the temperature window $T_{\text{c}} < T < T_{\text{ls}}$. However, the caveat to this is that the Ising temperature window can be exponentially small, and this is the case for $\delta J \ll J_{1\text{A}}$ where, $T_{\text{ls}} - T_{\text{c}} \propto \exp[-J_{1\text{A}}/\delta J]$.

For $T > T_{\text{ls}}$ the minimum of $|\epsilon_{\mathbf{k}}|$ moves away from $\mathbf{k} = (\pi, 0)$ and the critical behaviour crosses over to that of the Pokrovsky-Talapov universality class for $\delta T \gg T_{\text{ls}} - T_{\text{c}}$. Thus in the situation where $\delta J \ll J_{1\text{A}}$ the transition is technically an Ising transition, but all practical measurements, whether in experiment or simulation, will show the features of a Kasteleyn transition. The values of $T_{\text{c}}$ and $T_{\text{ls}}$ are shown as a function of $\delta J/J_{1\text{A}}$ in Fig. 22, and it can be seen that the Ising temperature window only starts to be significant for $\delta J/J_{1\text{A}} \gtrsim 0.3$.

Further increases in $T$ increase the size of the gap and in the limit $T \to \infty$ the spectrum, $|\epsilon_{\mathbf{k}}|$, becomes completely flat, corresponding to an uncorrelated paramagnet where all configurations are equally likely.

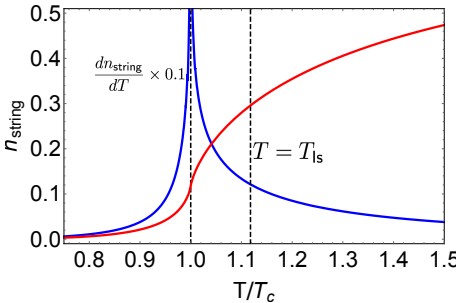

**Figure 20:** The density of strings, $n_{\text{string}}$ [Eq. 45] calculated from $|\epsilon_{\mathbf{k}}|$ [Eq. 60] in the unconstrained manifold (red). The parameters used are $J_{1\text{A}} = 1$ and $J_{1\text{B}} = 1.5$. Also shown are the value of $T_{\text{ls}}$ (black, dashed line) and $dn_{\text{string}}/dT$ (blue), which shows a logarithmic divergence at $T = T_{\text{c}}$.

The density of strings, $n_{\text{string}}$, can be calculated using Eq. 45 and the result is shown in Fig. 20. In the stripe-ordered phase $n_{\text{string}}$ is low, but not fixed to zero, as it is possible to create bound pairs of defect triangles, connected by a pair of strings. Its value increases rapidly at $T = T_{\text{c}}$, since the defect triangles unbind, and therefore strings can wind the system. On further increasing $T$ the density of strings passes through $n_{\text{string}} = 2/3$ (the value realised in the constrained manifold) before saturating at $n_{\text{string}} = 3/4$.

Since $n_{\text{string}}$ is not zero in the stripe phase, it is not, strictly speaking, an order parameter. However, it remains a useful indicator of where the transition occurs, since the derivative $dn_{\text{string}}/dT$ diverges logarithmically, as can be seen in Fig. 20.

## D.4 Ising to Pokrovsky-Talapov crossover

The nearest-neighbour TLIAF provides a good setting in which to study the crossover from Ising to Pokrovsky-Talapov critical behaviour, since physical quantities can be calculated directly in

the thermodynamic limit.

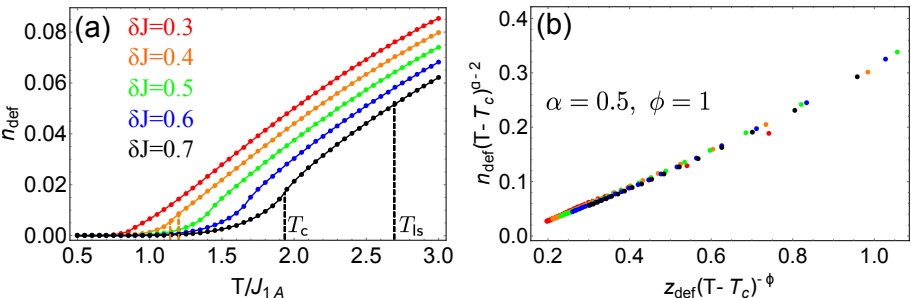

Figure 21: Determination of the Ising to Pokrovsky-Talapov crossover exponent, $\phi$, via scaling of the defect triangle density, $n_{\text{def}}$. (a) $n_{\text{def}}$ calculated in the thermodynamic limit for $\delta J = 0.3$ (red) to $\delta J = 0.7$ (black). $T_{\text{c}}$ and $T_{\text{ls}}$ are shown for $\delta J = 0.7$ (black) and $\delta J = 0.4$ (orange). (b) Scaling of $n_{\text{def}}$ using Eq. 63 gives good data collapse for $\alpha = 1/2$ and $\phi = 1$.

For $T_{\text{c}} < T < T_{\text{ls}}$ the system shows Ising critical exponents, while for $T - T_{\text{c}} \gg T_{\text{ls}} - T_{\text{c}}$ it shows Pokrovsky-Talapov criticality. The crossover between these two limiting cases can be understood by studying the density of defect triangles, $n_{\text{def}}$ (see Ref. [51] for a similar analysis in terms of monopoles in spin ice), and we postulate a scaling ansatz,

$$n_{\text{def}}(T, z_{\text{def}}) = |T - T_{\text{c}}|^{2-\alpha} g_\phi\left(\frac{z_{\text{def}}}{|T - T_{\text{c}}|^\phi}\right), \tag{63}$$

where $z_{\text{def}} = \exp[-E_{\text{def}}/T]$, $E_{\text{def}} = 2J_{1B} = 2(J_{1A} + \delta J)$ and $g_\phi$ is an unknown function. The exponent $\alpha$ is the usual heat capacity exponent, and is expected to take the value $\alpha = 1/2$ [48], while $\phi$ is the crossover exponent. By calculating $n_{\text{def}}$ in the thermodynamic limit and performing scaling according to Eq. 63 we find a convincing data collapse for $\phi = 1$, as shown in Fig. 21.

## D.5 Phase diagram and correlations

The phase diagram for $\mathcal{H}_{\text{ABB}}$ [Eq. 32] in the unconstrained manifold can be calculated exactly, and is shown in Fig. 22. The nature of the correlations can be explored via the spin structure factor [Eq. 4], and this is shown in the same figure for a representative set of parameters.

The phase diagram shows three regions, a stripe-ordered phase, a strongly-correlated spin-liquid region and a weakly correlated paramagnet. The stripe-ordered phase is separated from the disordered region by a phase transition at $T_{\text{c}}$ [Eq. 61], while we take the crossover between the spin-liquid and paramagnetic regions to occur when the density of defect triangles, $n_{\text{def}}$ reaches 10% of its saturation value (i.e. $n_{\text{def}} = 0.025$). As $\delta J$ is increased, the transition temperature $T_{\text{c}}$ increases faster than the crossover temperature, and therefore the spin-liquid region shrinks.

The nature of the correlations in the disordered phase changes significantly with varying $T$ and $\delta J$, and this can be seen from studying the structure factor [Eq. 4]. $S(\mathbf{r})$ can be calculated in the thermodynamic limit via the Grassmann path integral approach (see Appendix F), and some examples are shown in Fig. 22. Also shown is $S(\mathbf{q})$, which for simplicity is calculated using Monte Carlo simulation.

We make the ansatz that in the disordered regions $S(\mathbf{r})$ takes the long-distance asymptotic form [5] (see Appendix F),

$$S(\mathbf{r}) \propto \frac{\cos \mathbf{q} \cdot \mathbf{r} \, e^{-\frac{r_x}{\xi_\perp}} e^{-\frac{r_y}{\xi_\|}}}{\sqrt{|\mathbf{r}|}}, \tag{64}$$

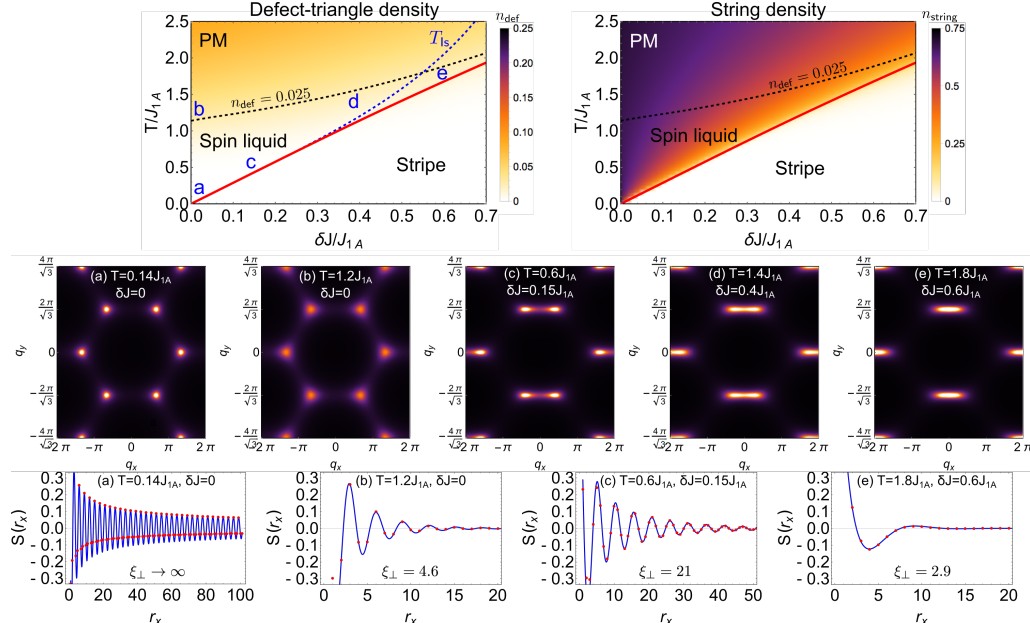

**Figure 22:** The phase diagram and structure factor of $\mathcal{H}_{\mathrm{ABB}}$ [Eq. 32] in the unconstrained manifold. (Top) Phase diagram showing stripe-ordered, spin-liquid and paramagnetic (PM) regions, with lines showing the critical temperature $T_c$ [Eq. 61, red] as well as $T_{ls}$ [Eq. 62, blue dashed]. Colour overlays show the density of defect triangles, $n_{\mathrm{def}}$, and string density, $n_{\mathrm{string}}$, calculated from the Grassmann path integral approach. (Middle) The structure factor $S(\mathbf{q})$ calculated on an $L = 72$ hexagonal cluster using Monte Carlo simulation. (Bottom) The real space structure factor $S(r_x)$ [Eq. 4] in the direction perpendicular to the strings (i.e. parallel to A bonds), calculated in the thermodynamic limit using the Grassmann path integral approach. The blue line shows a fit to the long-distance asymptotic form given in Eq. 64, with the fitting parameters $\xi_\perp$, $q_x$ and a multiplicative prefactor. (a) At $\delta J = 0$ and $T \ll J_{1A}$ the structure factor is indistinguishable from $T = 0$, with sharp essentially algebraic peaks at $\mathbf{q} = (\pm 2\pi/3, 2\pi/\sqrt{3})$ and a correlation length $\xi_\perp \to \infty$. (b) At $\delta J = 0$ and $T \sim J_{1A}$ the peaks remain at $\mathbf{q} = (\pm 2\pi/3, 2\pi/\sqrt{3})$ but broaden and the correlation length is only a few times larger than the lattice spacing. (c) For $\delta J \ll J_{1A}$ and for temperatures deep in the spin-liquid regime there are a pair of peaks whose positions approximately track the string density according to $\mathbf{q}_{\mathrm{string}}(T) = (\pm \pi n_{\mathrm{string}}, 2\pi/\sqrt{3})$. The correlation length is typically many times the lattice spacing. (d) At $\delta J \ll J_{1A} = 0.4$ the spin-liquid region is narrow, but weight at $\mathbf{q}_{\mathrm{string}}(T)$ remains more significant than that at $\mathbf{q}_{\mathrm{stripe}} = (0, 2\pi/\sqrt{3})$. (e) At $\delta J \ll J_{1A} = 0.6$ the spin-liquid regime has disappeared and above the transition the structure factor is dominated by correlations at the ordering vector, $\mathbf{q}_{\mathrm{stripe}} = (0, 2\pi/\sqrt{3})$.

where in the case of $\delta J \neq 0$ the correlation length perpendicular to the strings, $\xi_\perp$, can be different from that parallel to the strings, $\xi_\parallel$. This is found to give good fits to the calculated values of $S(\mathbf{r})$ after taking into account the definition of long distance given in Appendix C.4.

In the spin-liquid region the correlation length is considerably larger than the lattice spacing, and the system approximately realises the algebraically decaying correlation function studied in Appendix C.4 for the constrained manifold. In particular for $\delta J = 0$ and $T \ll J_{1A}$ the correlation length diverges as $\xi_\perp = \xi_\parallel \propto \exp[2J_{1A}/T]$ [53]. At the crossover to the paramagnetic region, the correlation length is approximately $\xi_\perp \sim 5$, with $\xi_\parallel \geq \xi_\perp$. Since the density of defect triangles is by definition low within the spin-liquid region, most of the strings wind the system, and therefore the relationship $\mathbf{q} \approx \pm \mathbf{q}_{\mathrm{string}}(T) = (\pm \pi n_{\mathrm{string}}(T), 2\pi/\sqrt{3})$ holds to a good approximation.

In the paramagnetic region the correlation lengths become comparable with the lattice spacing, and the structure factor has a very different form to the algebraic decay found for the constrained manifold. In this region the strings mostly form short closed loops, and therefore the relationship between $\mathbf{q}$ and $n_{\mathrm{string}}$ breaks down. For $\delta J \gtrsim 0.5$ there is a direct transition from the stripe-ordered phase to the paramagnet. In the Ising critical region close to the

transition the correlation function shows the usual 2D Ising scaling and is peaked in reciprocal space at $\mathbf{q}_{\text{stripe}}$.

In the stripe-ordered phase the asympotic form of $S(\mathbf{r})$ given in Eq. 64 is no longer relevant, and Bragg peaks form in $S(\mathbf{q})$ at the ordering vector $\mathbf{q}_{\text{stripe}} = (0, 2\pi/\sqrt{3})$. Fluctuations around the ground state are not strictly forbidden, but are rare unless $T \sim J_{1A}$, which is only possible for large anisotropies.

### D.6   Mapping to 1D quantum model

$\mathcal{H}_{\text{ABB}}$ [Eq. 32] in the unconstrained manifold can be exactly mapped onto a 1D quantum model of spinless fermions, as was found to be the case for the constrained manifold in Appendix C.5. The main difference is that in the unconstrained manifold defect triangles act as sources and sinks of pairs of strings. In consequence, pair creation and annihilation terms appear in the 1D quantum model.

Following a similar logic to that of Appendix G, there is an exact mapping of $\mathcal{H}_{\text{ABB}}$ [Eq. 32] onto

$$\mathcal{H}_{1\text{D}} = \sum_i \left[ -\mu c_i^\dagger c_i + t \left( c_i^\dagger c_{i+1} + c_{i+1}^\dagger c_i \right) + \Delta \left( c_i^\dagger c_{i+1}^\dagger + c_{i+1} c_i \right) \right], \tag{65}$$

where

$$t = z^2 + z_B^2, \quad \mu = 2z^2 - (1 + z_B^4), \quad \Delta = 2z_B z, \tag{66}$$

and the evolution of these parameters with the temperature of the classical model is shown in Fig. 23.

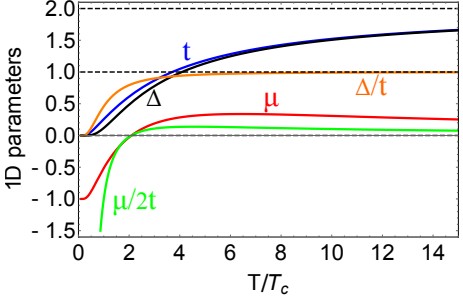

**Figure 23:** Mapping between the 2D classical model $\mathcal{H}_{\text{ABB}}$ [Eq. 32] and the 1D quantum model $\mathcal{H}_{1\text{D}}$ [Eq. 65]. The parameters of the 1D quantum model depend on those of the classical model according to Eq. 66, and the relationship is shown for $J_{1B}/J_{1A} = 1.5$. As $T \to 0$ then $t \to 0$, $\mu \to -1$, $\Delta \to 0$, $\mu/2t \to -\infty$ and $\Delta/t \to 0$. The phase transition occurs when $\mu/2t = -1$. In the limit $T \to \infty$ then $t \to 2$, $\mu' \to 0$, $\Delta \to 2$, $\mu/2t \to 0$ and $\Delta/t \to 1$.

$\mathcal{H}_{1\text{D}}$ [Eq. 65] can be diagonalised by Fourier and Bogoliubov transformations, resulting in,

$$\mathcal{H}_{1\text{D}} = \sum_k \omega_k a_k^\dagger a_k + \frac{1}{2} \sum_k (A_k - \omega_k), \tag{67}$$

where,

$$A_k = 2t \cos k - \mu, \quad B_k = 2\Delta \sin k, \quad \omega_k = \sqrt{A_k^2 + B_k^2}. \tag{68}$$

Physical properties of the classical TLIAF can be calculated directly from the quantum model. For example the classical quantity $n_{\text{string}}$ [Eq. 45] is equal to the fermion density [43].

# E $J_{1A}$-$J_{1B}$-$J_2$ model with a constrained manifold

The last simplified model we study consists of a TLIAF with first and second-neighbour interactions and a constraint forbidding defect triangles. The motivation is that this is the simplest form of further-neighbour interactions, and can be used to study a number of features of the more physically relevant TLIAF in a simplified setting. In particular we will consider the crossover of the phase transition into the stripe-ordered phase from first to second order via a Pokrovsky-Talapov tricritical point, the string Luttinger liquid (as opposed to the free-fermion spin liquid studied in Appendix C) and its crossover into a domain-wall network state. Since the further-neighbour interactions destroy the mapping onto a free-fermion model, we rely on a combination of Monte Carlo and perturbation theory.

The Hamiltonian is given by,

$$\mathcal{H}_{\text{ABB2}} = J_{1\text{A}} \sum_{\langle ij \rangle_{\text{A}}} \sigma_i \sigma_j + J_{1\text{B}} \sum_{\langle ij \rangle_{\text{B,C}}} \sigma_i \sigma_j + J_2 \sum_{\langle ij \rangle_2} \sigma_i \sigma_j, \tag{69}$$

where the second-neighbour bonds are labelled $\langle ij \rangle_2$ and we consider the constrained manifold of Ising configurations (i.e. no defect triangles).

## E.1 General considerations

Before turning to detailed calculations, it is worth considering some of the qualitative features of $\mathcal{H}_{\text{ABB2}}$ [Eq. 69], both in terms of the nature of the phase transition and of the correlations in the spin-liquid phase (there is no paramagnetic region due to being in the constrained manifold).

The second-neighbour interaction, $J_2$, and the nearest-neighbour anisotropy, $\delta J$, act in concert with one another, in the sense that they both favour a stripe-ordered ground state. However, they act in opposition in the sense that $J_2$ favours a first-order phase transition, while $\delta J$ favours a second-order transition.

This can be seen by comparing the $\delta J = 0$ case to that with $\delta J \gg J_2$. At $\delta J = 0$ the $J_2$ interaction selects a 6-fold degenerate, stripe-ordered ground state from the manifold of constrained Ising configurations [10]. The $J_1$-$J_2$ TLIAF has been extensively studied, both analytically and by Monte Carlo simulation, and it is known that there is a first-order phase transition into the stripe phase [7–11]. In the limit of $J_1 \rightarrow \infty$ the transition occurs at $T_1 = 6.39 J_2$ [11]. Therefore we expect that, in the region where $J_2 \gg \delta J$, the transition between the paramagnet and stripe-ordered state will be first order.

In contrast, the first-neighbour anisotropy, $\delta J$, favours a 2-fold degenerate stripe-ordered ground state, with stripes running parallel to A bonds (see Fig. 1 for the definition of bond directions). For $\delta J \gg J_2$ the $J_2$ interaction is irrelevant, and to a good approximation the analysis of Appendix C applies, indicating that the transition is second order. One focus here will be to study the crossover between the first and second-order phase transitions, which occurs when $J_2$ and $\delta J$ are comparable in magnitude.

When the transition is second order it is driven by the creation of isolated strings that wind the system (in [10] this is discussed in terms of the closely related concept of double domain walls). In order for a second-order transition to occur it is necessary that there is a repulsive interaction between these strings, and this repulsion is entropically driven and associated with the no-crossing constraint [39, 40]. We show below that further-neighbour interactions result in an energetically-driven attraction between the strings, and that the second to first order crossover occurs when this balances the entropically-driven repulsion.

The free energy of an isolated string can be calculated exactly, and this can be used to find the exact transition temperature in the case of a second-order transition. Relative to the

ground-state energy, strings cost an energy per unit length of $E_{\text{string}} = 2\delta J + 4J_2$ and corners, at which the string changes direction, have an energy cost $E_c = 2J_2$ [10, 11]. It follows that the free energy per unit length of an isolated string relative to the stripe ground state is given by [10],

$$f_{\text{string}}(T) = E_{\text{string}} - T \log\left[1 + e^{-\frac{E_c}{T}}\right]. \tag{70}$$

The second order transition temperature, $T_K$, can be calculated from solving the equation $f_{\text{string}}(T_K) = 0$, and in the case of $J_2 = 0$ it can be seen that this reduces to Eq. 44.

The behaviour in the spin-liquid state should be closely related to the nature of the phase transition, since it is also sensitive to whether the interaction between strings is attractive or repulsive. In the introduction it was argued that the associated fermionic model can be weakly or strongly coupled, and it makes intuitive sense that weakly-coupled fermions correspond to repulsive string-string interactions, while strongly-coupled fermions correspond to attractive string-string interactions. In the weak-coupling case it can be expected that the spin liquid realises a 2D classical equivalent of a Luttinger liquid. In the strong coupling case it is less clear what to expect *a priori*. The crossover between weak and strong coupling is controlled by the ratio $J_2/T$, with weak coupling for $T \gg J_2$.

### E.2 Diagrammatic perturbation theory

The first approximate method we use to better understand $\mathcal{H}_{\text{ABB2}}$ [Eq. 69] in the constrained manifold is that of perturbation theory around the high-temperature limit. This approach cannot hope to compete with Monte Carlo simulations in terms of quantitative measures of, for example, the transition temperature, but does provide useful physical insights that are not apparent in Monte Carlo. While the approach is well motivated in the "weak-coupling" regime, we find that it also gives some clues as to how the system crosses over to the "strong-coupling" regime and to the appearance of a first-order phase transition.

The starting point of the perturbation expansion is the exact solution of $\mathcal{H}_{\text{ABB}}$ [Eq. 32], which is summarised in Appendix C. This captures the behaviour of $\mathcal{H}_{\text{ABB2}}$ [Eq. 69] in the limit $J_2/T \to 0$. The perturbation expansion involves introducing the effect of the $J_2$ interactions order by order in the small parameter $|z_2 - 1|$, where,

$$z_2 = e^{-\frac{2J_2}{T}}, \tag{71}$$

and this can be done using a Grassmann path integral approach, following in spirit Ref. [61].

The first step is to map the Ising model, $\mathcal{H}_{\text{ABB2}}$ [Eq. 69], onto a dimer model on the dual honeycomb lattice. For the nearest-neighbour interactions the mapping is the same as in Appendix C. The second-neighbour coupling maps onto dimer-dimer interactions, where dimers on the same hexagon interact if they are separated by one unfilled bond (see Fig. 24). It follows that the partition function can be written as,

$$\mathcal{Z}_{\text{ABB2}} \propto \mathcal{Z}_{\text{hon2}} = \sum_{\text{dimer cov}} z^{N_{\text{dim}}^{\text{B}} + N_{\text{dim}}^{\text{C}}} z_2^{N_2}, \tag{72}$$

where $N_2$ is the number of dimer-dimer interactions (see Fig. 24).

The mapping of $\mathcal{Z}_{\text{hon2}}$ [Eq. 72] onto a Grassmann path integral does not result in a purely quadratic action, and therefore it is not exactly solvable by this method. Instead the mapping results in an action including terms with $2, 4, 6 \ldots 2N$ Grassmann variables, and one can write,

$$\mathcal{Z}_{\text{hon2}} = \int \prod_i da_i db_i \, e^{\mathcal{S}_2[a,b] + \mathcal{S}_4[a,b] + \mathcal{S}_6[a,b] + \cdots + \mathcal{S}_{2N}[a,b]}, \tag{73}$$

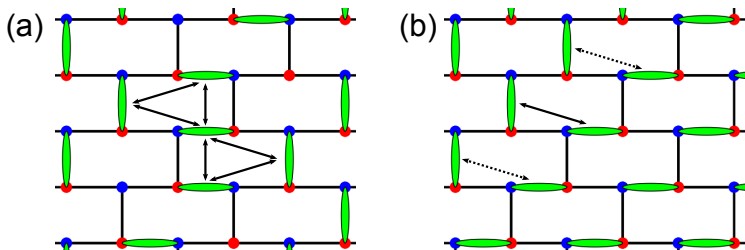

**Figure 24:** Dimer-dimer interactions on the brick (honeycomb) lattice. Dimers interact if they are on the same hexagonal plaquette and are separated by a single unfilled bond, and interactions are shown by black arrows. Each interaction carries a weight $z_2$ in the partition function $\mathcal{Z}_{\text{hon2}}$ [Eq. 72]. The system can be split into connected clusters of mutually interacting dimers, and these are only assigned the correct weight when the expansion of the action reaches that of the cluster size (see Eq. 73). (a) The largest connected cluster has 5 dimers, and it is therefore necessary to consider terms up to $\mathcal{S}_{10}[a, b]$. (b) The largest connected cluster has only 2 dimers, and the weight is correctly assigned by considering terms up to $\mathcal{S}_4[a, b]$.

where the quadratic term $\mathcal{S}_2[a, b]$ is given in Eq. 39 and contains products of 2 Grassmann variables, the quartic term, $\mathcal{S}_4[a, b]$, contains products of 4 Grassmann variables and similarly for higher order terms, with $2N$ the number of honeycomb lattice sites.

For a particular dimer configuration, one can ask which terms in the expansion of the action are required to correctly assign the weight. The answer depends on the size of the largest cluster of dimers connected by pairwise interactions (see Fig. 24). If the largest cluster contains $n$ dimers, then it is necessary to consider the terms $\mathcal{S}_{2m}[a, b]$ with $m \leq n$. Since clusters that include a sizeable fraction of all the dimers are common, many dimer configurations require one to consider terms up to $n \sim N$.

For an infinite lattice it is necessary to truncate the expansion of the action in order to be able to perform calculations. This can be done systematically by considering $z_2 - 1$ to be a small parameter, which is valid for $T \gg 2J_2$. The reason that this is a useful expansion parameter is due to the fact that $\mathcal{S}_{2n}[a, b]$ has a lowest order contribution proportional to $(z_2 - 1)^{n-1}$. Thus for a chosen value of $n$, it is only necessary to consider terms in the action up to $\mathcal{S}_{2n}[a, b]$. A simple worked example on a finite-size lattice is given in Appendix H to show how this type of expansion works in detail. Here we will consider $n = 2$, and therefore only retain the $\mathcal{S}_2[a, b]$ and $\mathcal{S}_4[a, b]$ terms in the action, thus working at first order in the small parameter $|z_2 - 1|$.

The quartic term in the action can be determined by observing that for a 2-site unit cell there are 6 terms containing 4 Grassmann variables, and these are shown schematically in Fig. 25. Thus one finds

$$\mathcal{S}_4[a, b] = (z_2 - 1) \tag{74}$$

$$\sum_i \Big[ z \big( b_i a_i b_{i+\hat{e}_y} a_{i+\hat{e}_x+\hat{e}_y} + b_i a_i a_{i+\hat{e}_x} b_{i+\hat{e}_x+\hat{e}_y} + b_i a_i b_{i-\hat{e}_x-\hat{e}_y} a_{i-\hat{e}_y} + b_i a_i a_{i-\hat{e}_x-\hat{e}_y} b_{i-\hat{e}_x} \big)$$

$$+ z^2 \big( b_i a_{i+\hat{e}_x} a_i b_{i+\hat{e}_y} + a_{i-\hat{e}_y} b_i b_{i-\hat{e}_x} a_i \big) \Big], \tag{75}$$

and taking the Fourier transform using Eq. 40 results in,

$$\mathcal{S}_4[a, b] = \frac{z_2 - 1}{N} \sum_{\mathbf{k}_1, \mathbf{k}_2, \mathbf{k}_3, \mathbf{k}_4} \delta_{\mathbf{k}_1 + \mathbf{k}_2 + \mathbf{k}_3 + \mathbf{k}_4, 0} V_4^{\text{sym}}(\mathbf{k}_1, \mathbf{k}_2, \mathbf{k}_3, \mathbf{k}_4) \, a_{\mathbf{k}_1} b_{\mathbf{k}_2} a_{\mathbf{k}_3} b_{\mathbf{k}_4}, \tag{76}$$

where

$$V_4^{\text{sym}}(\mathbf{k}_1, \mathbf{k}_2, \mathbf{k}_3, \mathbf{k}_4) = \frac{1}{4} \Big[ V_4(\mathbf{k}_1, \mathbf{k}_2, \mathbf{k}_3, \mathbf{k}_4) - V_4(\mathbf{k}_3, \mathbf{k}_2, \mathbf{k}_1, \mathbf{k}_4)$$

$$- V_4(\mathbf{k}_1, \mathbf{k}_4, \mathbf{k}_3, \mathbf{k}_2) + V_4(\mathbf{k}_3, \mathbf{k}_4, \mathbf{k}_1, \mathbf{k}_2) \Big] \tag{77}$$

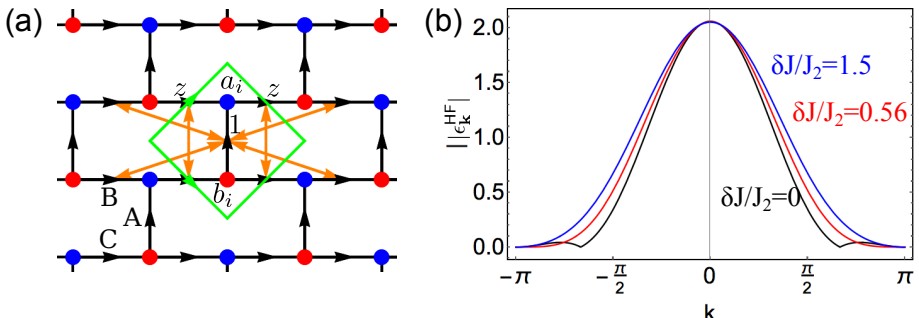

**Figure 25:** The interacting brick lattice and Hartree-Fock spectrum $\epsilon_{\mathbf{k}}^{\mathsf{HF}}$ [Eq. 85] used to perturbatively calculate $\mathcal{Z}_{\mathsf{hon2}}$ [Eq. 72]. (a) At first order in the perturbation expansion interactions occur between pairs of bonds that are on the same hexagon and separated by a single unfilled bond (shown in orange), resulting in a quartic interaction, $\mathcal{S}_4[a,b]$ [Eq. 76]. Higher-order terms in the action are associated with connected pairwise interactions, and thus involve 3 or more bonds (b) The spectrum $\epsilon_{\mathbf{k}}^{\mathsf{HF}}$ [Eq. 85] at the temperature for which $\epsilon_{(\pi,0)}^{\mathsf{HF}}(T) = 0$ along the path $\mathbf{k} = (k, k + \pi)$. For $\delta J/J_2 = 1.5$ (blue) the minimum of the dispersion occurs at $\mathbf{k} = (\pi, 0)$, indicating a second-order phase transition. For $\delta J/J_2 = 0$ (black) there is an additional zero at $\mathbf{k} \neq (\pi, 0)$, indicating a break-down of the perturbation theory and a first-order phase transition. The crossover between these two types of behaviour occurs at $\delta J/J_2 = 0.56$ (red), which corresponds to a Pokrovsky-Talapov tricritical point.

is the interaction vertex symmetrised over the pairs $\{\mathbf{k}_1, \mathbf{k}_3\}$ and $\{\mathbf{k}_2, \mathbf{k}_4\}$ and,

$$
\begin{aligned}
V_4(\mathbf{k}_1, \mathbf{k}_2, \mathbf{k}_3, \mathbf{k}_4) = z\, e^{-i\frac{k_{1x} - k_{1y}}{2}} &\left[ z\left( e^{i\frac{k_{3x} + k_{3y}}{2}} e^{ik_{4y}} + e^{-i\frac{k_{3x} + k_{3y}}{2}} e^{-ik_{4x}} \right) \right. \\
&\left. + e^{i\frac{k_{3x} + 3k_{3y}}{2}} e^{ik_{4y}} - e^{-i\frac{3k_{3x} + k_{3y}}{2}} e^{-ik_{4x}} - e^{i\frac{k_{3x} + k_{3y}}{2}} e^{i(k_{4x} + k_{4y})} + e^{-i\frac{k_{3x} + k_{3y}}{2}} e^{-i(k_{4x} + k_{4y})} \right].
\end{aligned} \tag{78}
$$

The truncated action, which is given by the sum of $\mathcal{S}_2[a,b]$ [Eq. 41] and $\mathcal{S}_4[a,b]$ [Eq. 76], has a quartic interaction term, and therefore it is not possible to perform the path integral exactly. Instead a perturbative diagrammatic approach can be used, as is standard in quantum field theory [61]. It is important to note that the expansion order of the perturbation theory is set by the truncation of the action, and only diagrams consistent with this order should be considered.

The first step in the construction of a diagrammatic perturbation theory is the calculation of the free Green's function, and this is given by,

$$
\langle a_{\mathbf{k}_1} b_{\mathbf{k}_2} \rangle_0 = \frac{1}{\mathcal{Z}_{\mathsf{hon}}} \int \prod_{\mathbf{k}} da_{\mathbf{k}} db_{-\mathbf{k}} a_{\mathbf{k}_1} b_{\mathbf{k}_2} e^{\mathcal{S}_2[a,b]} = \frac{\delta_{\mathbf{k}_1 + \mathbf{k}_2, 0}}{\epsilon_{\mathbf{k}_1}}. \tag{79}
$$

This can be used to perturbatively construct the interacting Green's function, which is given by,

$$
\langle a_{\mathbf{k}_1} b_{\mathbf{k}_2} \rangle = \frac{1}{\mathcal{Z}_{\mathsf{hon}}} \int \prod_{\mathbf{k}} da_{\mathbf{k}} db_{-\mathbf{k}} a_{\mathbf{k}_1} b_{\mathbf{k}_2} e^{\mathcal{S}_2[a,b] + \cdots + \mathcal{S}_{2N}[a,b]} = \frac{\delta_{\mathbf{k}_1 + \mathbf{k}_2, 0}}{\tilde{\epsilon}_{\mathbf{k}_1}}, \tag{80}
$$

where $\tilde{\epsilon}_{\mathbf{k}} = \epsilon_{\mathbf{k}} + \Sigma_{\mathbf{k}}$ and $\Sigma_{\mathbf{k}}$ is the self energy.

In the case we are considering, the anomalous Green's functions $\langle a_{\mathbf{k}_1} a_{\mathbf{k}_2} \rangle$ and $\langle b_{\mathbf{k}_1} b_{\mathbf{k}_2} \rangle$ vanish at all orders of perturbation theory, and this is related to the absence of defect triangles. In consequence the effective quadratic action takes the simple form,

$$
\mathcal{S}_{2,\mathsf{eff}}[a,b] = \sum_{\mathbf{k}} \tilde{\epsilon}_{\mathbf{k}}\, a_{\mathbf{k}} b_{-\mathbf{k}}, \tag{81}
$$

and it follows that the partition function can be written as,

$$\mathcal{Z}_{\text{hon2}} = \prod_{\mathbf{k}} \tilde{\epsilon}_{\mathbf{k}}. \tag{82}$$

In order to be consistent with the expansion of the action to first order in the small parameter $|z_2 - 1|$, we consider the Hartree-Fock diagrams, and therefore approximate the self energy as,

$$\Sigma_{\mathbf{k}} \approx \Sigma_{\mathbf{k}}^{\text{HF}} = (z_2 - 1)\Omega_{\mathbf{k}}, \tag{83}$$

where

$$\Omega_{\mathbf{k}} = \frac{1}{N} \sum_{\mathbf{k}'} \frac{2}{\epsilon_{\mathbf{k}'}} \left[ V_4^{\text{sym}}(\mathbf{k}, -\mathbf{k}, \mathbf{k}', -\mathbf{k}') - V_4^{\text{sym}}(\mathbf{k}, -\mathbf{k}', \mathbf{k}', -\mathbf{k}) \right]. \tag{84}$$

At this level of approximation the partition function is given by

$$\mathcal{Z}_{\text{hon2}} \approx \prod_{\mathbf{k}} \epsilon_{\mathbf{k}}^{\text{HF}}, \qquad \epsilon_{\mathbf{k}}^{\text{HF}} = \epsilon_{\mathbf{k}} + \Sigma_{\mathbf{k}}^{\text{HF}}. \tag{85}$$

The effective action $\mathcal{S}_{2,\text{eff}}[a,b]$ [Eq. 81] can be used to study the physical properties of the system, as in Appendix C. In particular we focus on the crossover between a second and first-order phase transition, which corresponds to the crossover from the weak to the strong coupling regimes (in fermionic language). Information about the nature of the phase transition can be extracted from the spectrum, $\epsilon_{\mathbf{k}}^{\text{HF}}$ [Eq. 85], and it can be seen in Fig. 25 that this undergoes a change of structure at $\delta J/J_2 = 0.56$.

For $\delta J/J_2 > 0.56$ the spectrum, $\epsilon_{\mathbf{k}}^{\text{HF}}$ [Eq. 85], shows the characteristic features of a second-order transition. In the disordered phase it has a gapless point at a temperature-dependent and incommensurate wavevector. As the temperature is lowered towards the critical point the gapless point migrates towards the wavevector $\mathbf{k} = (\pi, 0)$, and the critical temperature can be found from solving the equation $\epsilon_{(\pi,0)}^{\text{HF}}(T) = 0$. Below the transition the spectrum is gapped at all wavevectors, and the minimum is at $\mathbf{k} = (\pi, 0)$.

The second-order transition temperature is known exactly from Eq. 70, and Fig. 26 shows a comparison between the exact value and the estimate from first-order perturbation theory. First-order perturbation theory seems to work well even approaching the tricritical point, where $T_{\text{tri}} \approx 9J_2$ (the tricritical temperature will be determined more accurately by Monte Carlo simulations in the next section). At this temperature the small parameter is $1 - z_2(T_{\text{tri}}) \approx 0.2$, and so the perturbation expansion is reasonably well controlled. The discrepancy in the critical temperature between zeroth and first-order perturbation theory can be seen from expanding the exact second-order transition temperature as,

$$\frac{T_K}{\delta J} = \frac{2}{\log 2} + \frac{5}{\log 2} \frac{J_2}{\delta J} + \mathcal{O}\left( \frac{J_2^2}{\delta J^2} \right), \tag{86}$$

where it can be seen that for $\delta J \approx J_2$ the $J_2/\delta J$ term is larger than the leading term. In fact further expansion of the transition temperature reveals that at $\delta J \approx J_2$ higher order terms are not small, but do cancel one another. However, it is important to remember that $J_2/\delta J$ is not the expansion parameter.

Exactly at the critical temperature the spectrum has qualitatively the same behaviour as the $J_2 = 0$ case (see Appendix C) close to the gapless point. Along the path $\mathbf{k} = (k, k + \pi)$ the spectrum grows as $(k-\pi)^2$. This behaviour is typical of a Pokrovsky-Talapov transition [39,40].

At $\delta J/J_2 = 0.56$ the spectrum shows a change of character. The coefficient in front of the quadratic term goes to zero, and the spectrum grows as $(k - \pi)^4$ around the gapless point.

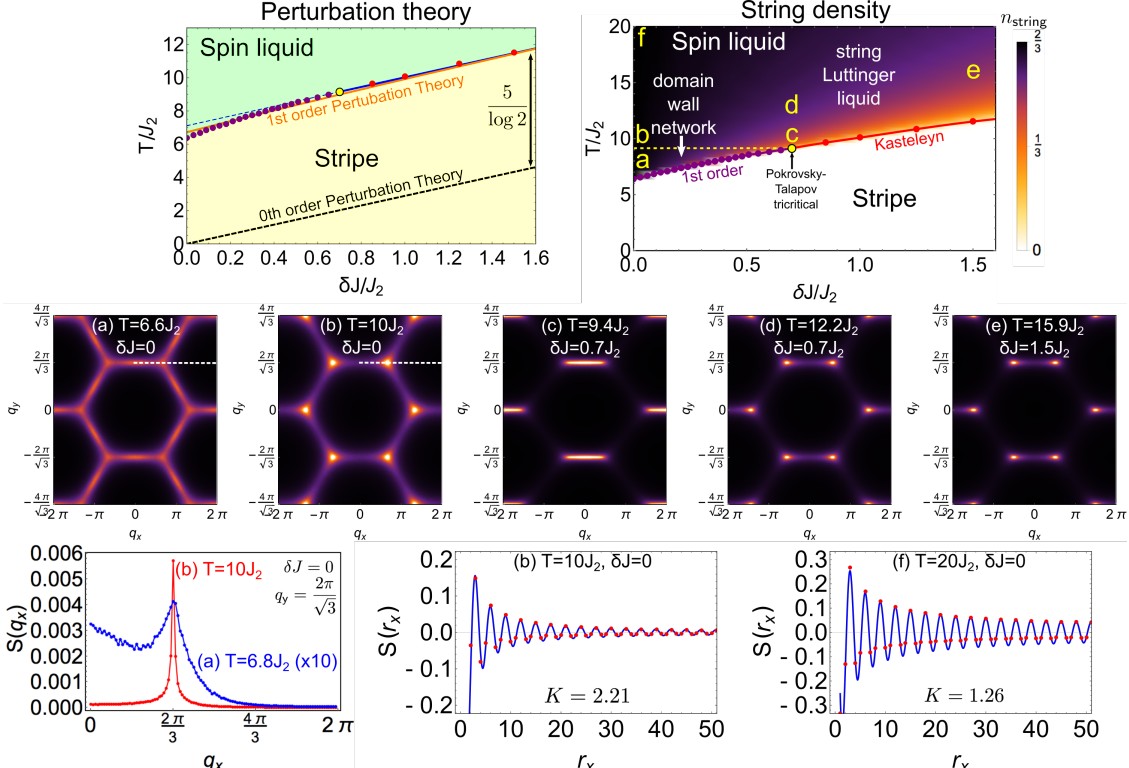

**Figure 26:** The phase diagram and structure factor of $\mathcal{H}_{\text{ABB2}}$ [Eq. 69] in the constrained manifold. (Top) Phase diagram showing the stripe-ordered and spin-liquid phases. First (purple), second (red) and Pokrovsky-Talapov tricritical (large yellow dot) transitions are determined from finite-size scaling analysis of Monte Carlo simulations [Eq. 87], and in the case of the second-order transition compare well to the exact value (blue solid line). Zeroth (black-dashed) and first-order (orange) perturbation theory calculations of the critical temperature (orange) are shown on the left hand plot (i.e. the temperature at which $\epsilon^{\text{HF}}_{(\pi,0)} = 0$ [Eq. 81]). On the right-hand plot the spin-liquid is split into a string Luttinger liquid for $T > T_{\text{tri}} = 9.14J_2$ and a domain-wall network for $T < T_{\text{tri}}$ (separated by dashed yellow line). (Middle) Structure factor $S(\mathbf{q})$ calculated by Monte Carlo simulation of an $L = 72$ hexagonal cluster (letters correspond to those on the phase diagram). (Bottom) Cuts through both $S(\mathbf{q})$ and $S(\mathbf{r})$. (a) For $\delta J = 0$ and close to the first-order phase transition $S(\mathbf{q})$ has significant spectral weight around the perimeter of the triangular-lattice Brillouin zone, as is typical of a domain-wall network. (b) On increasing the temperature spectral weight rapidly accumulates at $\mathbf{q} = (\pm 2\pi/3, 2\pi/\sqrt{3})$, as is typical for a string Luttinger liquid. In the whole string Luttinger liquid region the asymptotic form of $S(\mathbf{r})$ follows Eq. 90 with a parameter-dependent Luttinger parameter, $K \geq 1$. (c) At temperatures just above the Pokrovsky-Talapov tricritical point, there is a near-degeneracy between string sectors and the structure factor therefore shows extended spectral weight in the $q_x$ direction. (d) Further increasing the temperature breaks this quasi-degeneracy and sharp peaks form at $\mathbf{q}_{\text{string}}(T) = (\pm \pi n_{\text{string}}, 2\pi/\sqrt{3})$. (e) At temperatures just above the second-order transition the structure factor is sharply peaked at $\mathbf{q}_{\text{string}}(T)$. (f) For $T \gg J_2$ the behaviour of the nearest-neighbour TLIAF in the constrained manifold is recovered, with $K = 1$.

We refer to this point as a Pokrovsky-Talapov tricritical point, as the critical exponents are different from the standard ones of the Kasteleyn transition. This change of behaviour is not just an artifact of first-order perturbation theory, since its effects can be observed in Monte Carlo simulation (albeit at $\delta J/J_2 = 0.7$ – see Appendix E.3).

For $\delta J/J_2 < 0.56$ the perturbative approach breaks down, but can be used to find some clues as to the true situation. In the paramagnet there is a gapless point at an incommensurate wavevector, as shown for the case of $\delta J = 0$ in Fig. 27. As the temperature is reduced this migrates towards $\mathbf{k} = (\pi, 0)$, as is the case for a second-order transition. However, before the gapless point reaches $\mathbf{k} = (\pi, 0)$ the gap at $\mathbf{k} = (\pi, 0)$ closes, resulting in a pair of gapless points. This situation is not physical, and does not obviously correspond to the expected first-

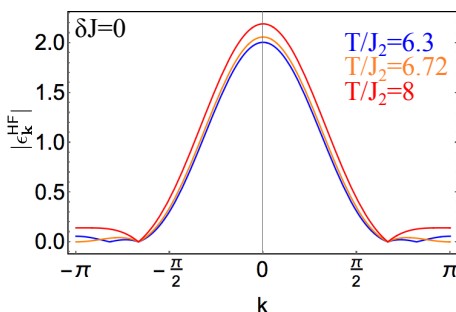

**Figure 27:** The spectrum $\epsilon_{\mathbf{k}}^{\mathsf{HF}}$ [Eq. 85] at $\delta J = 0$ and for varying temperature. The path through 2D reciprocal space is parametrised by $\mathbf{k} = (k, k + \pi)$. In the paramagnet (red) there is a single gapless point in the region $k > 0$ and this occurs at an incommensurate wavevector. At $T/J_2 = 6.72$ (orange) the gap at $\mathbf{k} = (\pi, 0)$ closes, but there remains a gapless point at an incommensurate wavevector. At lower $T$ (blue) the gapless points approach one another, and the spectrum is very flat in their vicinity. While it is clear that the perturbative approach has broken down at such a small value of $\delta J/J_2$, the results are suggestive that lines of zeros appear in the spectrum, and this would be consistent with a first-order phase transition.

order phase transition. However, it can be seen that the spectrum is very flat between the two gapless points. We suggest that in reality gapless lines should develop in this region, and this would correspond to a first-order phase transition. This type of behaviour can never be exactly recovered using a perturbative approach, since a gapless line relies on the correct relationship between all coefficents in the expansion of the free energy.

### E.3  Phase diagram determined from Monte Carlo simulations

As a complement to the perturbation theory approach, we also study $\mathcal{H}_{\mathsf{ABB2}}$ [Eq. 69] using Monte Carlo simulation.

The simulations are carried out using a worm algorithm very similar to that presented in Ref. [11]. This works in the dimer representation (see Appendix B.1), and creates loops of alternating dimer-filled and empty bonds, which are then flipped, resulting in the reversal of all the Ising spins contained within the loop. The loop creation is carefully controlled such that detailed balance is maintained, and the absence of rejection results in an efficient algorithm. Hexagonal shaped clusters with periodic boundary conditions are used, containing $N = 3L^2$ Ising spins, where $L$ measures the length of one side. Simulations are performed using system sizes from $L = 24$ up to $L = 192$.

The phase diagram of $\mathcal{H}_{\mathsf{ABB2}}$ [Eq. 69], as determined by Monte Carlo simulation, is shown in Fig. 26. The phase transitions can be located either from measuring the triangular average of the winding number and associated susceptibility, defined in Eq. 3, or by measuring the heat capacity, and the results are consistent.

In the region where the phase transition is second order, the critical temperature is found from finite-size scaling analysis. We use the standard relation for a Kasteleyn transition [62],

$$T_{\mathsf{K}}(L) = T_{\mathsf{K}}(\infty) - cL^{-1/\nu_{\parallel}}, \tag{87}$$

where $c$ is a constant, $L$ is the linear dimension of the system and $\nu_{\parallel} = 1$ is the critical exponent of the correlation length in the direction parallel to the double domain walls, below which the algebraic scaling of spin correlations breaks down [40, 47, 48]. We consider the parallel correlation length, $\nu_{\parallel}$, rather than the perpendicular correlation length, $\nu_{\perp}$, due to the anisotropy of the system which results in $\nu_{\perp} = 1/2 \neq \nu_{\parallel}$. Since the clusters used in the simulations are hexagonal in shape, and therefore isotropic, the growth of correlations parallel to the strings dominates the finite size effects. The exact second-order transition temperature is known from

solving Eq. 70, and it can be seen in Fig. 26 that the finite-size-scaled Monte Carlo results are in good agreement with this.

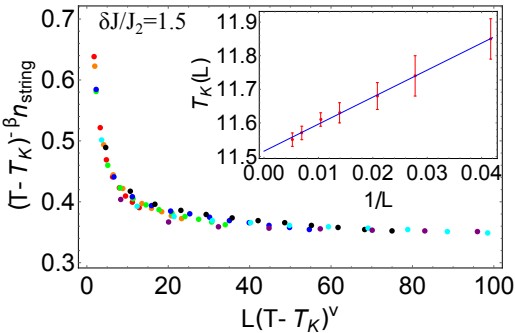

Figure 28: Data collapse demonstrating a second-order Pokrovsky-Talapov phase transition. The model in question is $\mathcal{H}_{\text{ABB2}}$ [Eq. 69] in the constrained manifold and $\delta J/J_2 = 1.5$. Simulations are run on hexagonal clusters with $N = 3L^2$ and $L = 24$ (red), $L = 36$ (orange), $L = 48$ (green), $L = 72$ (blue), $L = 96$ (black), $L = 144$ (cyan), $L = 192$ (purple). The data are plotted according to the scaling hypothesis given in Eq. 9, and the best collapse is found for $\beta = 0.47 \pm 0.04$ and $\nu_\parallel = 1.05 \pm 0.09$. This is consistent with $\beta = 1/2$ and $\nu_\parallel = 1$, which are the expected values for a second-order Pokrovsky-Talapov transition. The inset shows the scaling of the critical temperature, which follows Eq. 87.

As an example of such data collapse one can consider $\mathcal{H}_{\text{ABB2}}$ [Eq. 69] in the constrained manifold. We set $\delta J/J_2 = 1.5$, since this is far enough from the tricritical point that deviations from the Pokrovsky-Talapov universality class are expected to be negligible. The results are shown in Fig. 28, and a convincing data collapse is found for $\beta = 0.47 \pm 0.04$ and $\nu_\parallel = 1.05 \pm 0.09$, which is consistent with the expected $\beta = 1/2$ and $\nu_\parallel = 1$.

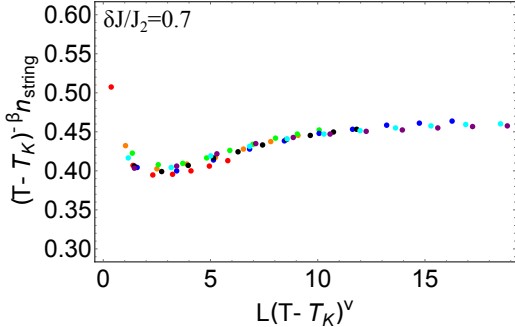

Figure 29: Data collapse demonstrating a Pokrovsky-Talapov tricritical point. The model in question is $\mathcal{H}_{\text{ABB2}}$ [Eq. 69] in the constrained manifold and $\delta J/J_2 = 0.7$. Simulations are run on hexagonal clusters with $N = 3L^2$ and $L = 24$ (red), $L = 36$ (orange), $L = 48$ (green), $L = 72$ (blue), $L = 96$ (black), $L = 144$ (cyan), $L = 192$ (purple). The data are plotted according to the scaling hypothesis given in Eq. 88, and the best collapse is found for $\beta = 0.21 \pm 0.04$ and $\nu_\parallel = 0.91 \pm 0.25$. This is consistent with $\beta = 1/4$, which is the expected value at a Pokrovsky-Talapov tricritical point.

The line of second order transitions ends at a Pokrovsky-Talapov tricritical point, which is found to be at $\delta J/J_2 = 0.7$ and $T = T_{\text{tri}} = 9.14 J_2$.

In order to test for the presence of a Pokrovsky-Talapov tricritical point in Monte Carlo simulations one can use the scaling hypothesis,

$$n_{\text{string}}(T, L) = (T - T_{\text{K}})^\beta g_{\text{tri}}\left(\frac{L}{\zeta_\parallel}\right). \tag{88}$$

If the data for different system sizes can be collapsed using $\beta = 1/4$, then this provides good evidence of the presence of a Pokrovsky-Talapov tricritical point. We apply this scaling hypoth-

esis to $\mathcal{H}_{\mathrm{ABB2}}$ [Eq. 69] in the constrained manifold in Fig. 29, and find that for $\delta J = 0.7$ the data can be convincingly collapsed using $\beta = 0.21 \pm 0.04$ and $\nu_{\parallel} = 0.91 \pm 0.25$.

The findings from Monte Carlo can be seen to be in reasonable agreement with first order perturbation theory (see Appendix E.2), where a Pokrovsky-Talapov tricritical point was found at $\delta J / J_2 = 0.56$.

For $\delta J / J_2 < 0.7$ the transition is first order, and it is typically possible to simulate large-enough systems that the finite-size effects are small. In consequence the transition temperatures plotted in Fig. 26 are taken from the largest simulated systems. For $0.5 < \delta J / J_2 < 0.7$ this is $L = 192$, while for $\delta J / J_2 < 0.5$ it is sufficient to consider $L = 48$.

### E.4 Monte Carlo simulations in the 2-string sector

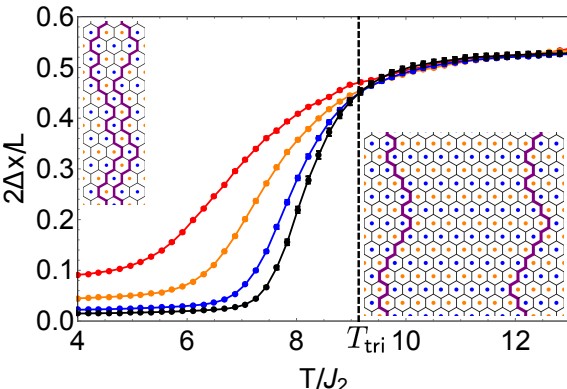

**Figure 30:** The average separation of a pair of strings, $2\Delta x / L$ [Eq. 89]. Monte Carlo simulations are carried out in a reduced manifold of Ising configurations, constrained to have exactly two strings. The simulations use a square cluster with linear sizes $L = 24$ (red), $L = 48$ (orange), $L = 96$ (blue) and $L = 144$ (black). At low temperature the strings bind together (see left-hand inset), while at higher temperatures the strings repel one another (see right-hand inset). The dashed line shows the temperature of the Pokrovsky-Talapov tricritical point, $T_{\mathrm{tri}}$, as measured by Monte Carlo simulation (see Fig. 26).

In order to gain physical insight into the crossover between a second and first-order phase transition, which in the spin-liquid region corresponds to the crossover between weak and strong coupling, we perform Monte Carlo simulations in a reduced manifold of states. The number of strings is fixed to be two, and the idea is to study the interaction between a pair of strings.

Monte Carlo simulations are performed on a square cluster with periodic boundary conditions, linear dimension $L$ and total number of sites $L^2$. In the 2-string manifold, allowed Ising configurations are distinguished by their $J_2$ energy, but all have the same energy in terms of $\delta J$, since there are a fixed number of dimers occupying B and C bonds. In consequence it is not necessary to vary $\delta J / J_2$, but only $T / J_2$. This shows that the string-string interactions are independent of $\delta J$, and therefore the temperature at which weak coupling crosses over to strong coupling is also $\delta J$ independent.

For each considered temperature we measure the average separation between the strings, taking into account the periodic boundary conditions, and this is given by

$$\Delta x = \langle \min[x_2 - x_1, L - (x_2 - x_1)] \rangle, \tag{89}$$

where $x_1$ and $x_2$ are the positions of the strings along the $x$ axis at a given height.

It can be seen in Fig. 30 that as the temperature is reduced there is a change in $\Delta x$ starting at about $T = T_{\mathrm{tri}}$. For $T > T_{\mathrm{tri}}$ the strings repel one another, and $\Delta x / (L/2) \approx 1/2$. This

repulsion is entropically driven, and is due to the no-crossing constraint obeyed by the strings, which reduces the available fluctuations of a string if it is in close proximity to another string. This type of pairwise repulsion is crucial for the existence of a second-order phase transition out of the stripe phase, since it limits the number of strings that are condensed into the system when the free energy of an isolated, $f_{\text{string}}$ [Eq. 70], goes to zero.

For $T < T_{\text{tri}}$ the pair of strings start to approach one another, showing that the energetically-driven attractive interaction starts to dominate over the repulsive interaction. The strings gain some binding free energy by being, on average, proximate to one another, and this is consistent with the crossover from a second to a first-order phase transition at $T = T_{\text{tri}}$ seen in the full Monte-Carlo simulations (see Fig. 26). The lower the temperature the more tightly the strings bind, suggesting that the transition should become more first-order as the temperature is decreased, and this is also consistent with the full simulations.

Since the string-string interactions are independent of $\delta J$, the spin liquid region should have attractive string-string interactions in the temperature window $T_1 < T < T_{\text{tri}}$, and we will discuss the implications of this in the next section.

### E.5 Mapping to 1D quantum model and correlations

The nature of the correlations in $\mathcal{H}_{\text{ABB2}}$ [Eq. 69] can be used to understand the behaviour of the spin liquid, and can be determined by combining Monte Carlo simulation of the spin structure factor with insights from fermionic mappings.

It is useful to first consider at a qualitative level how the mapping to a 1D quantum model of spinless fermions is altered by the further-neighbour interactions (see Appendix C.5 and Appendix G for the nearest-neighbour case). At the level of an isolated string the $J_2$ interaction both increases the internal energy, and adds an energy penalty to "corners" where the string changes direction [10,11]. In the fermion model this alters the values of $\mu$ and $t$ and adds a history dependence to the motion of the fermion, such that the passage from the imaginary timestep $\tau$ to $\tau + \Delta\tau$ depends not only on the fermion configuration at $\tau$ but also on the configuration at $\tau - \Delta\tau$.

A second effect of the $J_2$ coupling is to drive an attractive interaction between strings. When strings neighbour one another their $J_2$ energy is reduced, and therefore the fermionic model also has an attractive interaction of the form $V(z_2)c_i^\dagger c_i c_{i+1}^\dagger c_{i+1}$. In the string picture this attractive interaction is energetically-driven and competes with the entropically-driven repulsive interaction arising from the string non-crossing contraint. In the fermionic language the entropic repulsion maps onto the Pauli exclusion principle, which is a property of free fermions, and therefore the fermionic model is always attractive.

One advantage of mapping onto a fermion model is that it is known that fermions with weak attractive interactions form a Luttinger liquid, with Luttinger parameter $K > 1$ ($K = 1$ for free fermions) [52]. We therefore make the ansatz that the spin structure factor in the 2D classical model takes the asymptotic form [48,52],

$$S(\mathbf{r}) \propto \frac{\cos[\mathbf{q}_{\text{string}} \cdot \mathbf{r}]}{|\mathbf{r}|^{\frac{K}{2}}}, \tag{90}$$

where $K > 1$. This corresponds to a reciprocal space structure factor with algebraically sharp peaks at $\mathbf{q} = \mathbf{q}_{\text{string}}$. The asymptotic form given in Eq. 90 can be tested against Monte Carlo simulations, and we find that it gives a good fit to the simulations for $T \gtrsim T_{\text{tri}}$, and some examples are shown in Fig. 26. The value of $K$ can be extracted from the fits to the simulations, and the result of doing this for $T > T_{\text{tri}}$ and $\delta J = 0$ is shown in Fig. 31. It can be seen that close to $T = T_{\text{tri}}$ the Luttinger parameter, $K$, becomes significantly different from the free fermion case of $K = 1$, while in the limit $T/J_2 \to \infty$ the free fermion case is recovered, corresponding

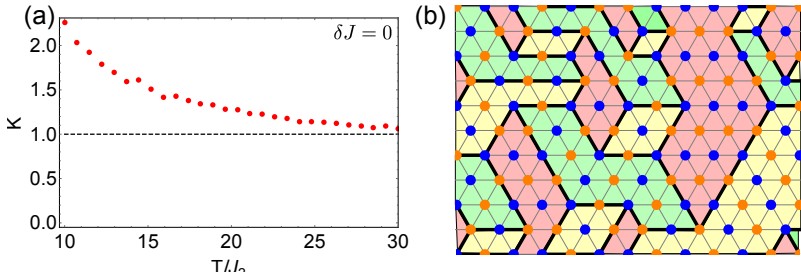

Figure 31: String Luttinger liquid and domain-wall network states. (a) The String Luttinger liquid is characterised by the parameter, $K$, and this is determined from Monte Carlo simulations of an $L = 72$ hexagonal cluster by fitting $S(\mathbf{r})$ with the asymptotic form given in Eq. 90. This type of fitting breaks down at $T \approx T_{\text{tri}} = 9.14 J_2$ (b) Snapshot of a domain-wall network configuration, taken from a Monte Carlo simulation at $\delta J = 0$ and $T = 6.5 J_2$. Domains have stripes parallel to A (red), B (green) or C (yellow) bonds. A domains correspond to an absence of strings while B and C domains to parallel neighbouring strings, and this type of configuration is driven by string-string attraction.

to $1/\sqrt{|\mathbf{r}|}$ spin correlations (see Eq. 50). As a result of these findings we label the region of the spin liquid with $T > T_{\text{tri}}$ as a string Luttinger liquid.

At $T = T_{\text{tri}}$ the entropic repulsion and energetic attraction between strings becomes comparable (see Fig. 30) and the strings start to bind together. At this temperature the distribution of spectral weight in the structure factor starts to rearrange itself such that $S(\mathbf{q})$ is no longer dominated by a single $\mathbf{q}$ value, and Eq. 90 is inapplicable. Instead the weight is distributed around the perimeter of the triangular-lattice Brillouin zone (see Fig. 26), and this is typical of a domain-wall network (see supplementary material of Ref. [11]). Neighbouring parallel strings form domains in which Ising stripes are parallel to either B or C bonds, while domains with stripes parallel to A bonds correspond to an absence of strings, and an example of this is shown in Fig. 31. We find that the spin-liquid region of $\mathcal{H}_{\text{ABB2}}$ [Eq. 69] is best described as a domain-wall network in the region $T_1 < T < T_{\text{tri}}$, as shown in Fig. 26. The domain-wall network state can be thought of as being a fluctuating, phase-separated state, with a loose analogy to the clustering of holes in superconductors [54, 55].

The more the energetically-driven attraction between strings dominates over the entropic repulsion, the more tightly bound the strings and the larger the average domain size. In the case of $\mathcal{H}_{\text{ABB2}}$ [Eq. 69] and for $\delta J = 0$ a first-order phase transition into the stripe phase occurs while the average domain size is relatively small. The addition of a third-neighbour interaction with $0 < J_3 < J_2/2$ suppresses the transition temperature, and therefore allows the average domain size to become larger since the attractive interaction becomes more important at low temperature [10, 11].

Increasing $\delta J$ causes domains with stripes parallel to A bonds to grow, which corresponds to decreasing the string density, $n_{\text{string}}$. At the tricritical point the A-domains coalesce and cover the whole system and there is a continous transition into the stripe phase.

## F The spin-spin correlation function for the nearest-neighbour TLIAF

Here we show how to calculate the real-space, spin-spin correlation function, $S(\mathbf{r})$ [Eq. 4], for the nearest-neighbour TLIAF, working in both the constrained manifold (i.e. without defect triangles) and the full, unconstrained manifold. Integral expressions for the correlation function can be derived in the thermodynamic limit, and numerical evaluation results in exact results up to numerical error. In the isotropic case these calculations just show how to derive the long-established results of Ref. [4, 5] within the Grassmann variable approach [59]. The point of showing the calculations here is that the Grassmann approach makes it simple to extend

the old results to the case of anisotropic interactions, for which it is necessary to separately consider correlations parallel and perpendicular to the string direction.

In this Appendix we show the mechanical steps used to calculate the correlation functions, while a physical discussion of the results is given in Appendix C, Appendix D and the main text. We consider separation vectors, $\mathbf{r} = \mathbf{r}_j - \mathbf{r}_i$, that are either perpendicular or parallel to the average direction of the strings, and label the corresponding correlation functions as $S_\perp(\mathbf{r})$ and $S_\parallel(\mathbf{r})$ (corresponding to the $\hat{e}_x$ and $\hat{e}_y$ direction in Fig. 32).

## F.1 Spin-spin correlations in the constrained manifold

First we consider the nearest-neighbour TLIAF with a constrained manifold. The calculations are slightly simplified by using a unit cell that contains 2 triangular lattice sites and 4 honeycomb/brick lattice sites, as shown in Fig. 32 (as opposed to the minimal unit cell with 1 triangular and 2 honeycomb/brick sites used in Appendix C). The two spins contained within the $i$th unit cell are labelled $\sigma_{1,i}$ and $\sigma_{2,i}$ and the perpendicular and parallel spin-spin correlation functions are,

$$S_\perp(r\hat{e}_x) = \langle \sigma_{1,i}\sigma_{1,i+\hat{e}_x}\rangle, \quad S_\parallel(r\hat{e}_y) = \langle \sigma_{2,i}\sigma_{2,i+\hat{e}_y}\rangle. \tag{91}$$

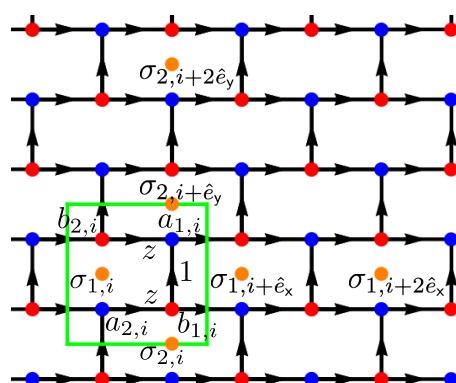

**Figure 32:** Brick lattice used for calculation of the nearest-neighbour TLIAF spin-spin correlation function in the constrained manifold. The (non-minimal) unit cell contains two spins, $\sigma_{1,i}$ and $\sigma_{2,i}$, as well as four Grassmann variables, labelled $a_{1,i}$, $b_{1,i}$, $a_{2,i}$ and $b_{2,i}$. Correlations between spins can be determined by studying expectation values of pairs of Grassmann variables associated with the intermediate bonds.

The unit cell also contains 4 Grassmann variables, labelled $a_{1,i}$, $b_{1,i}$, $a_{2,i}$ and $b_{2,i}$. These can be used to determine the partition function as in Appendix C, resulting in

$$\mathcal{Z}_{\text{hon}} = \int \prod_i da_{1,i}\,db_{1,i}\,da_{2,i}\,db_{2,i}\ e^{\mathcal{S}_2[a_1,b_1,a_2,b_2]}, \tag{92}$$

where the action is

$$\mathcal{S}_2[a_1,b_1,a_2,b_2] = \sum_i \Big[ b_{1,i}a_{1,i} + b_{2,i}a_{2,i+e_y} + z\big(a_{2,i}b_{1,i} + b_{2,i}a_{1,i} + b_{1,i}a_{2,i+e_x} + a_{1,i}b_{2,i+e_x}\big)\Big]. \tag{93}$$

Fourier transforming the Grassmann variables results in

$$\mathcal{S}_2[a_1,b_1,a_2,b_2] = \sum_{\mathbf{k}} \big(b_{1,-\mathbf{k}}, b_{2,-\mathbf{k}}\big)\begin{pmatrix} e^{ik_y/2} & 2iz\sin\frac{k_x}{2} \\ 2iz\sin\frac{k_x}{2} & e^{ik_y/2} \end{pmatrix}\begin{pmatrix} a_{1,\mathbf{k}} \\ a_{2,\mathbf{k}} \end{pmatrix}, \tag{94}$$

and this is diagonalised to give

$$\mathcal{Z}_{\text{hon}} = \prod_{\mathbf{k}} \epsilon_{\mathbf{k}}^{(4)}, \qquad \epsilon_{\mathbf{k}}^{(4)} = e^{ik_y} + 2z^2(1 - \cos k_x). \tag{95}$$

### F.1.1   Correlations perpendicular to the strings

In order to calculate the spin-spin correlation function, it is necessary to express products of spins in terms of Grassmann variables. Before considering the general case, it is useful to first consider a pair of spins, $\sigma_{1,i}$ and $\sigma_{1,i+\hat{e}_x}$, separated by a single honeycomb/brick lattice bond (see Fig. 32). If this bond is covered by a dimer then the spins are equivalent and $\sigma_{1,i}\sigma_{1,i+\hat{e}_x} = 1$, while if it is not dimer-covered $\sigma_{1,i}\sigma_{1,i+\hat{e}_x} = -1$. The expectation value is therefore given by

$$\langle \sigma_{1,i}\sigma_{1,i+\hat{e}_x} \rangle = P_{b_{1,i};a_{1,i}}^{\text{dim}} - \left(1 - P_{b_{1,i};a_{1,i}}^{\text{dim}}\right) = 2P_{b_{1,i};a_{1,i}}^{\text{dim}} - 1 \tag{96}$$

where $P_{b_{1,i};a_{1,i}}^{\text{dim}}$ is the probability of finding a dimer on the bond connecting the Grassmann variables $b_{1,i}$ and $a_{1,i}$. In order to determine $P_{b_{1,i};a_{1,i}}^{\text{dim}}$ one can calculate a reduced partition function in which the sites $b_{1,i}$ and $a_{1,i}$ are excluded. Exclusion of these sites effectively fixes a dimer on the bond between them, and therefore $P_{b_{1,i};a_{1,i}}^{\text{dim}}$ is given by the ratio of the reduced partition function to the original partition function, $\mathcal{Z}_{\text{hon}}$ [Eq. 95]. In order to exclude the two sites, it is simply necessary to place $b_{1,i}$ and $a_{1,i}$ inside the partition function integral, using the properties of Grassmann variables ($a^2 = 0$). In consequence one finds,

$$P_{b_{1,i};a_{1,i}}^{\text{dim}} = \frac{1}{\mathcal{Z}_{\text{hon}}} \int \prod_j da_{1,j}db_{1,j}da_{2,j}db_{2,j} \; b_{1,i}a_{1,i} \; e^{\mathcal{S}_2[a_1,b_1,a_2,b_2]} = \langle b_{1,i}a_{1,i} \rangle, \tag{97}$$

and it is clear that $P_{b_{1,i};a_{1,i}}^{\text{dim}}$ is just the thermodynamic average of $b_{1,i}a_{1,i}$. In consequence,

$$\langle \sigma_{1,i}\sigma_{1,i+\hat{e}_x} \rangle = \langle 2b_{1,i}a_{1,i} - 1 \rangle. \tag{98}$$

The thermodynamic average of two Grassmann variables can be calculated using

$$\langle b_{1,i}a_{1,j} \rangle = \frac{2}{N} \sum_{\mathbf{k}} \langle b_{1,-\mathbf{k}}a_{1,\mathbf{k}} \rangle e^{i\mathbf{k}\cdot(\mathbf{r}_j - \mathbf{r}_i)} e^{ik_y/2}, \quad \langle b_{1,-\mathbf{k}}a_{1,\mathbf{k}} \rangle = \frac{e^{ik_y/2}}{\epsilon_{\mathbf{k}}^{(4)}}. \tag{99}$$

In the isotropic case ($z = 1$) integration yields $P_{b_{1,i};a_{1,i}}^{\text{dim}} = 1/3$, as expected, and therefore $\langle \sigma_{1,i}\sigma_{1,i+\hat{e}_x} \rangle = -1/3$.

More generally, the correlation between a pair of spins with a separation vector parallel to $\hat{e}_x$ is given by

$$S_\perp(r\hat{e}_x) = \left\langle \prod_{l=0}^{r-1} (2b_{1,i+l\hat{e}_x}a_{1,i+l\hat{e}_x} - 1) \right\rangle. \tag{100}$$

This can be expanded using Wick's theorem, and rewritten as the deteminant of an $r \times r$-dimensional Toeplitz matrix, resulting in

$$S_\perp(r\hat{e}_x) = \det \mathbf{M}_\perp, \tag{101}$$

with components,

$$(\mathbf{M}_\perp)_{mn} = 2\langle b_{1,i}a_{1,i+(n-m)\hat{e}_x} \rangle - \delta_{mn}. \tag{102}$$

In the thermodynamic limit the sum can be converted into an integral giving

$$(\mathbf{M}_\perp)_{mn} = \frac{1}{2\pi^2} \int_{-\pi}^{\pi} dk_x e^{i(n-m)k_x} \int_{-\pi}^{\pi} dk_y \frac{e^{ik_y}}{e^{ik_y} + u} - \delta_{mn}, \tag{103}$$

where $u = 2z^2(1 - \cos k_x)$. The integral over $k_y$ is given by

$$\frac{1}{2\pi} \int_{-\pi}^{\pi} dk_y \frac{e^{ik_y}}{e^{ik_y} + u} = \left\{ \begin{array}{ll} 1 & |u| < 1 \\ 0 & |u| > 1 \end{array} \right. . \tag{104}$$

It follows that,

$$(\mathbf{M}_\perp)_{mn} = \frac{2 \sin[k_F(n-m)]}{\pi(n-m)} - \delta_{mn}, \tag{105}$$

where

$$k_F = \arccos\left(1 - \frac{1}{2z^2}\right), \tag{106}$$

is the Fermi wavevector of the quantum model $\mathcal{H}_{1D}$ [Eq. 51]. It can be seen that $(\mathbf{M}_\perp)_{mn} = (\mathbf{M}_\perp)_{nm}$ and thus the Toeplitz matrix is symmetric. It is also worth noting that the matrix elements could have been calculated by making use of the exact mapping onto the 1D quantum model given in Appendix C.5.

### F.1.2 Correlations parallel to the strings

The correlation between spins parallel to $\hat{e}_y$ can be calculated by an analagous method. The difference is that a pair of spins are separated by not one but two dimers (see Fig. 32). As such the correlation function is given by

$$S_\parallel(r\hat{e}_y) = \left\langle \prod_{l=0}^{r-1} (2z a_{2,i+l\hat{e}_y} b_{1,i+l\hat{e}_y} - 1)(2z b_{2,i+l\hat{e}_y} a_{1,i+l\hat{e}_y} - 1) \right\rangle. \tag{107}$$

where the $z$'s take into account the weights of the excluded dimers. Wick's theorem allows this to be rewritten as the determinant of a $2r \times 2r$-dimensional Toeplitz matrix,

$$S_\perp(r\hat{e}_x) = \det \mathbf{M}_\parallel, \tag{108}$$

with components

$$\begin{aligned} (\mathbf{M}_\parallel)_{2m-1,2n-1} &= 2z\langle a_{2,i} b_{1,i+(n-m)\hat{e}_y} \rangle - \delta_{mn} \\ (\mathbf{M}_\parallel)_{2m,2n} &= 2z\langle b_{2,i} a_{1,i+(n-m)\hat{e}_y} \rangle - \delta_{mn} \\ (\mathbf{M}_\parallel)_{2m-1,2n} &= 2z\langle a_{2,i} b_{2,i+(n-m)\hat{e}_y} \rangle \\ (\mathbf{M}_\parallel)_{2m,2n-1} &= 2z\langle b_{1,i} a_{1,i+(n-m)\hat{e}_y} \rangle, \end{aligned} \tag{109}$$

where $m, n \in \{1 \ldots r\}$. The matrix elements can be calculated from

$$\begin{aligned} \langle b_{1,i} a_{2,j} \rangle &= \frac{2}{N} \sum_{\mathbf{k}} \langle b_{1,-\mathbf{k}} a_{2,\mathbf{k}} \rangle e^{i\mathbf{k}\cdot(\mathbf{r}_j - \mathbf{r}_i)} e^{-ik_x/2} \\ \langle b_{2,i} a_{1,j} \rangle &= \frac{2}{N} \sum_{\mathbf{k}} \langle b_{2,-\mathbf{k}} a_{1,\mathbf{k}} \rangle e^{i\mathbf{k}\cdot(\mathbf{r}_j - \mathbf{r}_i)} e^{ik_x/2} \\ \langle b_{2,i} a_{2,j} \rangle &= \frac{2}{N} \sum_{\mathbf{k}} \langle b_{2,-\mathbf{k}} a_{2,\mathbf{k}} \rangle e^{i\mathbf{k}\cdot(\mathbf{r}_j - \mathbf{r}_i)} e^{-ik_y/2} \\ \langle b_{1,i} a_{1,j} \rangle &= \frac{2}{N} \sum_{\mathbf{k}} \langle b_{1,-\mathbf{k}} a_{1,\mathbf{k}} \rangle e^{i\mathbf{k}\cdot(\mathbf{r}_j - \mathbf{r}_i)} e^{ik_y/2}, \end{aligned} \tag{110}$$

where

$$\langle b_{1,-\mathbf{k}} a_{2,\mathbf{k}} \rangle = \langle b_{2,-\mathbf{k}} a_{1,\mathbf{k}} \rangle = -\frac{2iz \sin \frac{k_x}{2}}{\epsilon_{\mathbf{k}}^{(4)}}$$

$$\langle b_{1,-\mathbf{k}} a_{1,\mathbf{k}} \rangle = \langle b_{2,-\mathbf{k}} a_{2,\mathbf{k}} \rangle = \frac{e^{ik_y/2}}{\epsilon_{\mathbf{k}}^{(4)}}. \tag{111}$$

## F.2 Spin-spin correlations in the unconstrained manifold

Calculation of the spin-spin correlation function in the nearest-neighbour TLIAF with an unconstrained manifold follows a very similar pattern to that of the constrained manifold. However it is complicated by having to work with 6 or 12 Grassmann variables in the unit cell, as well as the fact that the extended brick lattice is not bipartite.

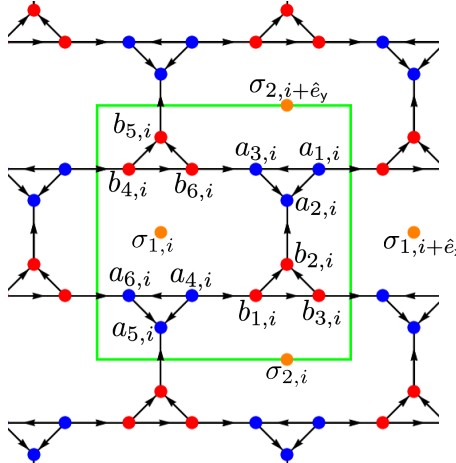

**Figure 33:** Extended brick lattice used for calculation of the nearest-neighbour TLIAF spin-spin correlation function in the unconstrained manifold. The (non-minimal) unit cell contains two spins, $\sigma_{1,i}$ and $\sigma_{2,i}$, as well as twelve Grassmann variables, labelled $a_1...a_6$ and $b_1...b_6$. Correlations between spins can be determined by studying expectation values of pairs of Grassmann variables associated with the intermediate bonds.

The two-spin unit cell is shown in Fig. 33 and contains 12 sites of the extended brick lattice, and therefore 12 Grassmann variables, which are labelled $a_1...a_6$ and $b_1...b_6$. The partition function can be calculated as in the constrained case, and this results in

$$\mathcal{Z}_{\text{exhon}} = \prod_{k_x>0,k_y} \epsilon_{\mathbf{k}}^{(12)}, \tag{112}$$

with

$$\epsilon_{\mathbf{k}}^{(12)} = (1-z_B^4)^2 + 4(z^2+z_B^2)^2 + 4\cos k_x (z_B^2(1+z_B^4)-2z^4)$$
$$+ 4\cos^2 k_x (z^2-z_B^2)^2 + 4z^2(1-z_B^2)^2 \cos k_y (1-\cos k_x). \tag{113}$$

The spin-spin correlation function in the direction perpendicular to the strings (parallel to $\hat{e}_x$) is given by

$$S_\perp(r\hat{e}_x) = \left\langle \prod_{l=0}^{r-1} (2b_{2,i+l\hat{e}_x} a_{2,i+l\hat{e}_x} - 1) \right\rangle, \tag{114}$$

and as in the constrained manifold case the correlation function can be written as the determinant of an $r \times r$-dimensional Toeplitz matrix, $\mathbf{M}_\perp$, with matrix elements

$$(\mathbf{M}_\perp)_{mn} = 2\langle b_{2,i} a_{2,i+(n-m)\hat{e}_x}\rangle - \delta_{mn}. \tag{115}$$

The correlation function in the direction parallel to the strings (parallel to $\hat{e}_y$) is given by

$$S_\parallel(r\hat{e}_y) = \left\langle \prod_{l=0}^{r-1} (2z a_{4,i+l\hat{e}_y} b_{1,i+l\hat{e}_y} - 1)(2z b_{6,i+l\hat{e}_y} a_{3,i+l\hat{e}_y} - 1) \right\rangle, \tag{116}$$

and this can be rewritten as the determinant of a $2r \times 2r$-dimensional Toeplitz matrix, $\mathbf{M}_\parallel$, with matrix elements,

$$(\mathbf{M}_\parallel)_{2m-1,2n-1} = 2z\langle a_{4,i} b_{1,i+(n-m)\hat{e}_y}\rangle - \delta_{mn}$$
$$(\mathbf{M}_\parallel)_{2m,2n} = 2z\langle b_{6,i} a_{3,i+(n-m)\hat{e}_y}\rangle - \delta_{mn}$$
$$(\mathbf{M}_\parallel)_{2m-1,2n} = 2z\langle a_{4,i} b_{6,i+(n-m)\hat{e}_y}\rangle$$
$$(\mathbf{M}_\parallel)_{2m,2n-1} = 2z\langle b_{1,i} a_{3,i+(n-m)\hat{e}_y}\rangle. \tag{117}$$

The matrix elements of interest can be determined from

$$\langle b_{2,i} a_{2,j}\rangle = \frac{2}{N} \sum_{\mathbf{k}} \langle b_{2,-\mathbf{k}} a_{2,\mathbf{k}}\rangle e^{i\mathbf{k}\cdot(\mathbf{r}_j - \mathbf{r}_i)} e^{ik_y/4}$$

$$\langle b_{1,i} a_{4,j}\rangle = \frac{2}{N} \sum_{\mathbf{k}} \langle b_{1,-\mathbf{k}} a_{4,\mathbf{k}}\rangle e^{i\mathbf{k}\cdot(\mathbf{r}_j - \mathbf{r}_i)} e^{-ik_x/4}$$

$$\langle b_{6,i} a_{3,j}\rangle = \frac{2}{N} \sum_{\mathbf{k}} \langle b_{6,-\mathbf{k}} a_{3,\mathbf{k}}\rangle e^{i\mathbf{k}\cdot(\mathbf{r}_j - \mathbf{r}_i)} e^{ik_x/4}$$

$$\langle b_{6,i} a_{4,j}\rangle = \frac{2}{N} \sum_{\mathbf{k}} \langle b_{6,-\mathbf{k}} a_{4,\mathbf{k}}\rangle e^{i\mathbf{k}\cdot(\mathbf{r}_j - \mathbf{r}_i)} e^{-ik_y/2}$$

$$\langle b_{1,i} a_{3,j}\rangle = \frac{2}{N} \sum_{\mathbf{k}} \langle b_{1,-\mathbf{k}} a_{3,\mathbf{k}}\rangle e^{i\mathbf{k}\cdot(\mathbf{r}_j - \mathbf{r}_i)} e^{ik_y/2}, \tag{118}$$

where $\langle b_{2,-\mathbf{k}} a_{2,\mathbf{k}}\rangle$ etc. are relatively simple to calculate within the Grassmann variable approach, but result in very length expressions.

Finally, we note that when calculating the determinant of the Toeplitz matrices, there is typically a finely-tuned cancellation between different terms. In the calculations presented above, this is unproblematic, since the matrix elements can be computed exactly (at least up to numerical accuracy). However, this limits the utility of the Toeplitz matrix approach for models with further-neighbour interactions where only a perturbative solution of the Grassmann variable spectrum is available. Small, unavoidable errors in the calculation of the matrix elements quickly have a significant effect on the determinant, resulting in unphysical results. For this reason we do not use this method for the $J_{1A}$-$J_{1B}$-$J_2$ model considered in Appendix E, but instead rely on finite-size Monte Carlo simulations of the correlation function.

# G  Mapping between the 2D Kasteleyn partition function and a 1D fermionic coherent-state path integral

Here we demonstrate the correspondence between the 2D, classical, nearest-neighbour TLIAF, $\mathcal{H}_{ABB}$ [Eq. 32], and a 1D quantum model of spinless fermions, $\mathcal{H}_{1D}$ [Eq. 51]. For simplicity,

we work in the constrained manifold of Ising configurations, but an analagous calculation can be carried out in the unconstrained manifold. In particular, we show that the Kasteleyn formulation of the partition function naturally maps onto a quantum path integral written in terms of fermionic coherent states. This makes clear that the Grassmann variables introduced in the Kasteleyn method describe coherent states of strings.

The standard way to map between a $d$ dimensional classical theory and a $d-1$ dimensional quantum theory is by equating the transfer matrix and the Hamiltonian according to $\mathcal{T} \equiv e^{-\delta\tau\mathcal{H}}$, where $\delta\tau$ is a small step in imaginary time. The partition function can then be viewed either as a matrix product of transfer matrices, or as a quantum path integral. Detailed examples of how to carry out this procedure in the cases of spin-ice and the cubic dimer model are presented in Ref. [63, 64].

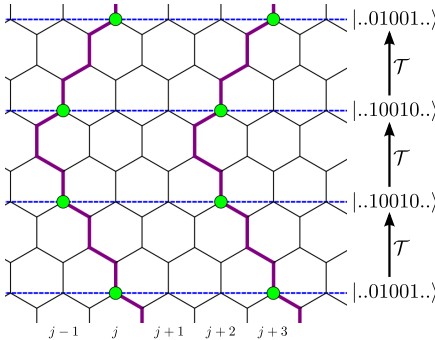

Figure 34: The action of the transfer matrix for the nearest-neighbour TLIAF in a constrained manifold. The transfer matrix translates strings (shown in purple) by four honeycomb bonds in the $y$ direction (from one blue dashed line to the next). At each translation the string can follow one of four possible routes, with the result that its $x$ coordinate either remains the same or gets translated one step to the left or right. The string state on one of the blue dashed lines can be specified by the presence (1) or absence (0) of a string at each site on the line. This can be re-interpreted as the presence or absence of a spinless fermion at a particular imaginary timestep in a 1D quantum model.

In the case of the TLIAF, the transfer matrix acts on states of strings, as shown in Fig. 34, and a translation across 4 bonds is necessary before the lattice structure repeats. These string states can be reinterpreted as the fermionic state of a 1D quantum model at a given imaginary time coordinate, and the classical partition function sum is therefore equivalent to the fermionic path integral.

While it is possible to solve the nearest-neighbour TLIAF using a transfer matrix approach [2, 3], the solution is considerably more compact using the Kasteleyn formulation expressed as a multiple integral over Grassmann variables (see Appendix C and Appendix D). Furthermore, the Kasteleyn appoach provides a good starting point for perturbative studies of more complicated models (see Appendix E.2). As such it would be useful to know how to link the Kasteleyn action to that of the fermionic path integral.

We demonstrate below that the Kasteleyn action naturally maps onto the quantum action when written in terms of fermionic coherent states. To do this we first re-examine the classical partition function using a non-minimal, 4-site unit cell, motivated by the fact that the string states shown in Fig. 34 involve a translation across 4 sites. We then examine the coherent-state fermionic path integral, and show that the action can be brought to the same form as the Kasteleyn action by introducing and summing over extra degrees of freedom that take into account the intermediate sites present in the honeycomb/brick lattice (see Fig. 34). Finally we link the Kasteleyn spectrum to that of the quantum model.

It should be noted that the mapping could just as well have been performed in the other direction, by starting from the Kasteleyn action and performing a Gaussian integral over half of the Grassmann variables to arrive at the fermionic coherent-state path integral. While this

alternative method is probably slightly more direct, we feel that the method we present makes clearer the physical link between the two.

## G.1 Kasteleyn action in 4-site basis

The procedure for determining the Kasteleyn action in terms of Grassmann variables was developed in [6] and is reviewed in Appendix C. To ease the comparison with the 1D quantum model, $\mathcal{H}_{1D}$ [Eq. 51], we here re-determine the Kasteleyn action of the nearest-neighbour TLIAF in the constrained manifold, using a 4-site unit cell and a different set of bond orientations compared to the main text.

The reason for using a 4-site cell is that it is natural to identify a string traversing 4 sites of the honeycomb/brick lattice with a single imaginary timestep in the quantum model (see Fig. 34). The bond orientations are shown in Fig. 35 and the reason they are different from those in the main text is just to simplify the mapping. They are of course chosen in accordance with Kasteleyn's theorem [46] and therefore there is no effect on the physical properties.

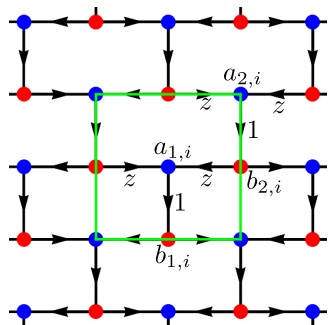

Figure 35: The set-up of the brick lattice used to map between the 2D classical model $\mathcal{H}_{\text{ABB}}$ [Eq. 32] and the 1D quantum model $\mathcal{H}_{1D}$ [Eq. 51]. To simplify the mapping a non-minimal, 4-site (6-bond) unit cell is chosen and the bond directions are different from in the main text. The four Grassmann variables contained within a unit cell are labelled $a_1$, $b_1$, $a_2$ and $b_2$.

Referring to Fig. 35, the partition function $\mathcal{Z}_{\text{hon}}$ [Eq. 38] can be rewritten as

$$\mathcal{Z}_{\text{hon}} = \int \prod_i da_{1,i} db_{1,i} da_{2,i} db_{2,i} \, e^{\mathcal{S}_2[a_1,b_1,a_2,b_2]}, \tag{119}$$

with

$$
\begin{aligned}
\mathcal{S}_2[a_1,b_1,a_2,b_2] = \\
\sum_i -b_{1,i} a_{1,i} - b_{2,i} a_{2,i} + z(b_{1,i+e_y} a_{2,i} + b_{1,i+e_x+e_y} a_{2,i} + b_{2,i} a_{1,i} + b_{2,i-e_x} a_{1,i}),
\end{aligned} \tag{120}
$$

where $e_x$ and $e_y$ are the translation vectors of the unit cell.

The action can be block diagonalised by Fourier transform, resulting in

$$\mathcal{S}_2[a_1,b_1,a_2,b_2] = \sum_{\mathbf{p}} \left( a_{1,\mathbf{p}}, a_{2,\mathbf{p}} \right) \begin{pmatrix} e^{ip_y/2} & -2z\cos\frac{p_x}{2} \\ -2z\cos\frac{p_x}{2} & e^{ip_y/2} \end{pmatrix} \begin{pmatrix} b_{1,-\mathbf{p}} \\ b_{2,-\mathbf{p}} \end{pmatrix}. \tag{121}$$

The partition function can therefore be written as

$$\mathcal{Z}_{\text{hon}} = \prod_{\mathbf{p}} \epsilon_{\mathbf{p}}^{K} = \prod_{\mathbf{p}} \sqrt{\epsilon_{\mathbf{p}}^{K} \epsilon_{-\mathbf{p}}^{K}} = \prod_{\mathbf{p}} |\epsilon_{\mathbf{p}}^{K}| \tag{122}$$

where the determinant of the $2 \times 2$ matrix is

$$\epsilon_{\mathbf{p}}^{\mathsf{K}} = e^{ip_y} - 4z^2 \cos^2 \frac{p_x}{2}, \tag{123}$$

with modulus

$$|\epsilon_{\mathbf{p}}^{\mathsf{K}}| = \sqrt{1 - 8z^2 \cos^2 \frac{p_x}{2} \cos p_y + 16z^4 \cos^4 \frac{p_x}{2}}. \tag{124}$$

## G.2 Fermionic coherent-state path integral

The Kasteleyn action can be mapped onto the fermionic, coherent-state path integral. To make the discussion self-contained, we briefly review how to construct such a path integral, following [65].

The 1D quantum model under consideration is given by

$$\mathcal{H}_{1D} = \frac{1}{2} \sum_l \left[ -(2z^2 - 1)c_l^\dagger c_l + z^2 \left( c_{l+1}^\dagger c_l + c_{l-1}^\dagger c_l \right) \right], \tag{125}$$

where the coefficients have been chosen in anticipation of the final result. The associated partition function is

$$\mathcal{Z}_{1D} = \sum_{\{n\}} \langle n|e^{-\beta \mathcal{H}_{1D}}|n\rangle, \tag{126}$$

where $\{n\}$ is a complete set of states in any Hilbert-state basis, and it should be remembered that the quantum inverse temperature, $\beta$, is related to the periodicity of the classical model in the $y$ direction and not to the temperature of the TLIAF.

The idea is to replace the Hilbert-state basis with that of fermionic-coherent states. These are eigenvectors of the annihilation operator, $c_l$, and therefore obey the eigenvalue equation,

$$c_l|\eta\rangle = \eta_l|\eta\rangle, \tag{127}$$

where $\eta_l$ is a Grassmann variable. It follows that the coherent states are described by

$$|\eta\rangle = \exp\left[ -\sum_l \eta_l c_l^\dagger \right]|0\rangle, \quad \langle \eta| = \langle 0| \exp\left[ \sum_l \bar{\eta}_l c_l \right], \tag{128}$$

where $|0\rangle$ is the fermionic vacuum. The action of creation and annihilation operators is given by

$$c_l|\eta\rangle = \eta_l|\eta\rangle, \quad \langle \eta|c_l^\dagger = \langle \eta|\bar{\eta}_l, \tag{129}$$

where $\eta$ and $\bar{\eta}$ are independent variables. The coherent states form an overcomplete basis, with overlap,

$$\langle \theta|\eta\rangle = \exp\left[ \sum_l \bar{\theta}_l \eta_l \right]|0\rangle, \tag{130}$$

and the completeness relation,

$$\int \prod_l d\bar{\eta}_l d\eta_l \, e^{-\sum_l \bar{\eta}_l \eta_l} |\eta\rangle\langle \eta| = \mathbb{1}. \tag{131}$$

Insertion of the completeness relation into $\mathcal{Z}_{1D}$ [Eq. 126] results in

$$\mathcal{Z}_{1D} = \int d(\bar{\eta}_0, \eta_0) e^{-\sum_l \bar{\eta}_{l,0} \eta_{l,0}} \langle -\eta_0 | e^{-\beta \mathcal{H}} | \eta_0 \rangle, \tag{132}$$

where

$$\langle -\eta | = \langle 0 | \exp \left[ -\sum_l \bar{\eta}_l c_l \right], \tag{133}$$

and

$$\int d(\bar{\eta}, \eta)_m = \int \prod_l d\bar{\eta}_{l,m} d\eta_{l,m}. \tag{134}$$

The path integral is then formed by the usual time slicing procedure to give

$$
\begin{aligned}
\mathcal{Z}_{1D} = \int & \left[ \prod_m d(\bar{\eta}, \eta)_m \right] \langle -\eta_0 | e^{-\delta\tau \mathcal{H}_{1D}} | \eta_{L-1} \rangle e^{-\sum_l \bar{\eta}_{l,L-1} \eta_{l,L-1}} \\
& \times \dots \\
& \times \langle \eta_2 | e^{-\delta\tau \mathcal{H}_{1D}} | \eta_1 \rangle e^{-\sum_l \bar{\eta}_{l,1} \eta_{l,1}} \\
& \times \langle \eta_1 | e^{-\delta\tau \mathcal{H}_{1D}} | \eta_0 \rangle e^{-\sum_l \bar{\eta}_{l,0} \eta_{l,0}} \\
= \int & \left[ \prod_m d(\bar{\eta}, \eta)_m \right] e^{\mathcal{S}_{1D}[\eta, \bar{\eta}]},
\end{aligned}
\tag{135}
$$

where $\delta\tau = \beta/L$ and $\eta_{l,m}$ is labelled by a spatial index $l$ and an imaginary time index $m$. Since $\mathcal{H}_{1D}$ [Eq. 125] is normal ordered, its matrix elements are simply calculated using Eq. 129, and

$$\mathcal{S}_{1D}[\eta, \bar{\eta}] = \sum_m \left[ -\delta\tau \mathcal{H}_{1D}(\bar{\eta}_{m+1}, \eta_m) - \sum_l \bar{\eta}_{l,m+1}(\eta_{l,m+1} - \eta_{l,m}) \right]. \tag{136}$$

### G.3 Matching the fermionic and Kasteleyn actions

The action, $\mathcal{S}_{1D}[\eta, \bar{\eta}]$, exactly reproduces the partition function of the nearest-neighbour TLIAF, but it does this by averaging over some of the degrees of freedom of the Kasteleyn action. In order to make the mapping explicit, it is necessary to introduce these extra degrees of freedom into the quantum path integral.

The microscopic relation between the quantum and classical partition functions requires a correspondence between an imaginary timestep and a translation of the classical system across 4 bonds in the $y$ direction (see Fig. 36). In the classical set-up the string can hop by a maximum of one 1D lattice site per imaginary timestep, and the expansion of the quantum time-translation operator can therefore be truncated to first order without approximation, giving

$$e^{-\delta\tau \mathcal{H}_{1D}} \to \mathbb{1} - \delta\tau \mathcal{H}_{1D}. \tag{137}$$

However, it can be seen in Fig. 36 that, if the string configuration is only known every 4 bonds (i.e. on the blue dashed lines in Fig. 36), there is an ambiguity, since a string can take two possible routes that leave its $x$ coordinate invariant. In terms of the original Ising spins, these two possible routes describe different configurations. In order to explicitly describe all

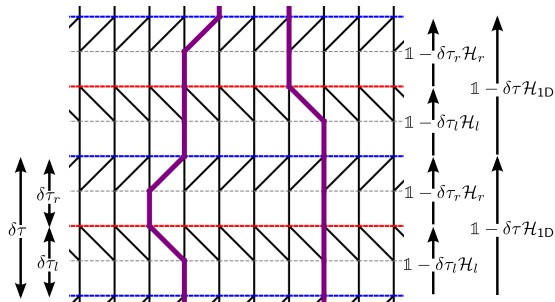

**Figure 36:** A redrawing of the honeycomb/brick lattice that clarifies the mapping between the nearest-neighbour TLIAF in the constrained manifold and $\mathcal{H}_{1D}$ [Eq. 125]. $\mathcal{H}_{1D}$ translates the strings/fermions (purple lines) by four bonds, from dashed blue line to dashed blue line (the four possible paths are shown). Knowing the string/fermion configuration only on the blue dashed lines does not completely specify the Ising configuration, since the string can take two possible routes that result in no change in its $x$ position. As a result, it is necessary to introduce into the coherent-state path integral extra states that take into account the string/fermion configuration on the red dashed lines (see Eq. 140). In order to do this the Hamiltonian is split into $\mathcal{H}_l$ and $\mathcal{H}_r$, describing hopping to the left and right [Eq. 138].

the classical degrees of freedom, it is necessary to consider the translation of a string by only 2 bonds at a time, and therefore the quantum Hamiltonian is split into

$$\mathcal{H}_l = \sum_l \left[ -(z-1)c_l^\dagger c_l - z c_{l-1}^\dagger c_l \right], \quad \mathcal{H}_r = \sum_l \left[ -(z-1)c_l^\dagger c_l - z c_{l+1}^\dagger c_l \right], \tag{138}$$

where the effect of the non-Hermitian operator $\mathcal{H}_l$ is to translate from the blue to red dashed lines in Fig. 36, while $\mathcal{H}_r$ translates from red to blue dashed lines. It is clear from Fig. 36 that $\mathcal{H}_l$ only includes left-hopping while $\mathcal{H}_r$ only includes right-hopping. The coefficients of $\mathcal{H}_l$ and $\mathcal{H}_r$ have been chosen such that the matrix elements obey the relationship

$$\langle n_{m+1}|(\mathbb{1} - \delta\tau\mathcal{H}_{1D})|n_m\rangle = \sum_{\{u\}} \langle n_{m+1}|(\mathbb{1} - \delta\tau_r\mathcal{H}_r)|u\rangle \langle u|(\mathbb{1} - \delta\tau_l\mathcal{H}_l)|n_m\rangle, \tag{139}$$

with $\{u\}$ a complete set of states in any Hilbert-space basis and $\delta\tau = \delta\tau_l + \delta\tau_r$. Furthermore we have set $\delta\tau_l = \delta\tau_r = 1$ to correspond to the lattice spacing of the classical model.

The splitting of the Hamiltonian can be built into the coherent-state path integral by introducing sets of intermediate coherent states that are associated with the red dashed lines in Fig. 36. These are labelled by $\theta$, and the resulting path integral is

$$\begin{aligned}
\mathcal{Z} &= \int \left[ \prod_m d(\bar{\eta}, \eta)_m d(\bar{\theta}, \theta)_m \right] \langle \eta_0 | e^{-\mathcal{H}_r} | \theta_{L-1} \rangle \, e^{-\sum_l \bar{\theta}_{l,L-1} \theta_{l,L-1}} \\
&\qquad \times \langle \theta_{L-1} | e^{-\mathcal{H}_l} | \eta_{L-1} \rangle \, e^{-\sum_l \bar{\eta}_{l,L-1} \eta_{l,L-1}} \\
&\qquad \times \ldots \\
&\qquad \times \langle \eta_2 | e^{-\mathcal{H}_r} | \theta_1 \rangle \, e^{-\sum_i \bar{\theta}_{i,1} \theta_{i,1}} \langle \theta_1 | e^{-\mathcal{H}_l} | \eta_1 \rangle \, e^{-\sum_i \bar{\eta}_{i,1} \eta_{i,1}} \\
&\qquad \times \langle \eta_1 | e^{-\mathcal{H}_r} | \theta_0 \rangle \, e^{-\sum_i \bar{\theta}_{i,0} \theta_{i,0}} \langle \theta_0 | e^{-\mathcal{H}_l} | \eta_0 \rangle \, e^{-\sum_i \bar{\eta}_{i,0} \eta_{i,0}} \\
&= \int \left[ \prod_m d(\bar{\eta}, \eta)_m d(\bar{\theta}, \theta)_m \right] e^{\mathcal{S}_{1D}[\eta, \bar{\eta}, \theta, \bar{\theta}]},
\end{aligned} \tag{140}$$

where

$$\mathcal{S}_{1D}[\eta, \bar{\eta}, \theta, \bar{\theta}] =$$
$$\sum_{l,m} \left[ -\bar{\eta}_{l,m}\eta_{i,m} - \bar{\theta}_{l,m}\theta_{l,m} + z(\bar{\eta}_{l,m+1}\theta_{l,m} + \bar{\eta}_{l+1,m+1}\theta_{l,m} + \bar{\theta}_{l,m}\eta_{l,m} + \bar{\theta}_{l-1,m}\eta_{l,m}) \right]. \tag{141}$$

The fermionic action is now in a form that can be directly compared with the Kasteleyn action $\mathcal{S}_2[a_1, b_1, a_2, b_2]$ [Eq. 120]. It can be seen that these can be brought to the same form simply by identifying

$$\eta_{l,m} \to a_{1,i}, \ \bar{\eta}_{l,m} \to b_{1,i}, \ \theta_{l,m} \to a_{2,i}, \ \bar{\theta}_{l,m} \to b_{2,i}, \tag{142}$$

where there is an equivalence between $i$, which labels the unit cells in the 2D lattice, and $(l, m)$, which labels the sites and timeslices in the 1D quantum problem. This justifies the mapping between the quantum and classical coefficients given in Eq. 51.

### G.4 Matching the classical and quantum spectrums

As well as showing how the Kasteleyn and fermionic coherent state actions can be brought to the same form, it is also useful to show the link between the spectrums. Since the fermions/strings are free, the partition functions can be simply evaluated by Fourier transform, and the spectrums compared.

The Kasteleyn spectrum is given by $|\epsilon_{\mathbf{p}}^{\mathsf{K}}|$ [Eq. 124], and the partition function, $\mathcal{Z}_{\text{hon}}$ [Eq. 122] is the product of this spectrum.

The fermionic spectrum is (see Appendix C.5)

$$\omega_{p_x} = 2t \cos p_x - \mu = 2z^2(\cos p_x - 1) + 1, \tag{143}$$

and this appears in the Fourier transform of $\mathcal{S}_{1D}[\eta, \bar{\eta}]$ [Eq. 136],

$$\mathcal{S}_{1D}[\eta, \bar{\eta}] = \sum_{\mathbf{p}} \epsilon_{\mathbf{p}}^{1D} \ \bar{\eta}_{\mathbf{p}} \eta_{\mathbf{p}}, \tag{144}$$

where $\delta\tau = 2$ has been used,

$$\epsilon_{\mathbf{p}}^{1D} = -\omega_{p_x} e^{ip_y} - 1 + e^{ip_y}, \tag{145}$$

and

$$\bar{\eta}_{\mathbf{p}} = \frac{1}{L^2} \sum_{l,m} \bar{\eta}_{l,m} e^{i(lp_x + mp_y)}, \quad \eta_{\mathbf{p}} = \frac{1}{L^2} \sum_{l,m} \eta_{l,m} e^{-i(lp_x + mp_y)}. \tag{146}$$

Since the action is diagonal, the partition function is just given by

$$\mathcal{Z}_{1D} = \prod_{\mathbf{p}} \epsilon_{\mathbf{p}}^{1D}. \tag{147}$$

The equivalence between $\mathcal{Z}_{\text{hon}}$ [Eq. 122] and $\mathcal{Z}_{1D}$ is now clear since the modes can be matched according to

$$\epsilon_{\mathbf{p}}^{\mathsf{K}} = -e^{ip_y} \epsilon_{p_x + \pi, -p_y - \pi}^{1D} = -\omega_{p_x + \pi} + e^{ip_y} + 1. \tag{148}$$

In conclusion, there is an exact mapping between the Kasteleyn action and that of the fermionic, coherent-state path integral with Hamiltonian $\mathcal{H}_{1D}$ [Eq. 125]. This mapping also makes it clear that the Grassmann variables introduced in the Kasteleyn formulation of the classical partition function describe coherent states of strings, and therefore demonstrates the link between the Kasteleyn and transfer matrix approach to solving the nearest-neighbour TLIAF.

# H    Perturbative expansion of the action: a simple example

The Grassmann path integral representation of interacting dimer problems, as used in Appendix E.2, is not unknown [61,66] but has not been widely explored in the literature. As an aid to the interested reader, we here consider $\mathcal{Z}_{hon2}$ [Eq. 72], and show a worked example of how to evaluate this partition function via Grassmann path integration on the simplest, nontrivial lattice: the hexagonal plaquette. This provides useful insights into the construction of a perturbation theory for the infinite lattice, as presented in Appendix E.2.

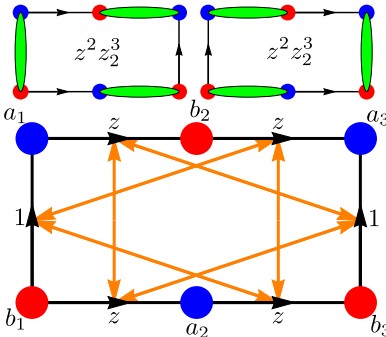

Figure 37:   Dimer covering and Grassmann path integral representation of the partition function for the hexagonal plaquette. The interacting dimer model, $\mathcal{Z}_{hon2}$ [Eq. 72], has dimer weight 1 on A bonds and $z$ on B and C bonds, and there is a weight $z_2$ associated with interactions between dimers separated by a single unfilled bond. (Top) The two dimer coverings of the hexagonal plaquette are shown, along with the associated weights. (Bottom) The partition function can be recast as a path integral over Grassmann variables associated with vertices of the plaquette (red and blue disks). The action consists of 2, 4 and 6 body interactions, and the allowed 4-body interactions are represented as orange arrows connecting pairs of bonds.

We consider the dimer covering of a hexagonal plaquette with a dimer weight of 1 on A bonds, $z$ on B and C bonds and a dimer interaction with weight $z_2$ between dimers separated by one unfilled bond (see Fig. 37). For a single plaquette there are only two possible dimer coverings, shown in Fig. 37, and each of these has a weight $z^2 z_2^3$. The partition function, $\mathcal{Z}_{hon2}$ [Eq. 72], is therefore given by

$$\mathcal{Z}_{hon2} = 2z^2 z_2^3 = 2z^2 \left[ (z_2-1)^3 + 3(z_2-1)^2 + 3(z_2-1) + 1 \right], \tag{149}$$

where the second equality is an exact rewriting that will prove useful below.

While in such a simple case the partition function can be calculated exactly just by inspection, it is instructive to perform the calculation via the Grassmann path integral representation. On a finite lattice the highest order term in the action is $\mathcal{S}_{2N}[a, b]$, where $2N$ is the number of honeycomb lattice sites, and for the 6-site plaquette the partition function can therefore be rewritten as

$$\mathcal{Z}_{hon2} = \int \prod_i da_i db_i \ e^{\mathcal{S}_2[a,b]+\mathcal{S}_4[a,b]+\mathcal{S}_6[a,b]}, \tag{150}$$

where $i = \{1, 2, 3\}$.

The quadratic term in the action does not take into account the $z_2$ interaction, and is given by

$$\mathcal{S}_2 = b_1 a_1 + b_3 a_3 + z(a_1 b_2 + b_1 a_2 + b_2 a_3 + a_2 b_3). \tag{151}$$

If the action is truncated at quadratic order, then the usual rules of Grassmann integration can be used to find

$$\int \prod_i da_i db_i \; e^{\mathcal{S}_2[a,b]} = 2z^2. \tag{152}$$

Comparison with the exact value of the partition function [Eq. 149] shows that this quadratic approximation becomes exact in the limit $z_2 - 1 \to 0$.

The quartic term in the action takes into account pairwise interactions of the dimers in isolation from other pairwise interactions, and is given by

$$\begin{aligned}
\mathcal{S}_4 =&\, z(z_2 - 1)(b_1 a_1 b_2 a_3 + b_1 a_1 a_2 b_3 + b_3 a_3 a_1 b_2 + b_3 a_3 b_1 a_2) \\
&+ z^2(z_2 - 1)(b_1 a_2 a_1 b_2 + a_2 b_3 b_2 a_3).
\end{aligned} \tag{153}$$

The factor $z_2 - 1$ is chosen such that if there were a configuration with only 1 dimer-dimer interaction, the -1 would remove the contribution from the purely quadratic action, while the $z_2$ would replace this with a contribution that takes the interaction into account. In the case of the hexagonal plaquette the allowed dimer configurations contain 3 mutually interacting dimers, and this mutual interaction is not fully taken into account by the quartic term. Direct evaluation results in

$$\int \prod_i da_i db_i \; e^{\mathcal{S}_2[a,b]+\mathcal{S}_4[a,b]} = 2z^2 [3(z_2 - 1) + 1], \tag{154}$$

reproducing the exact partition function [Eq. 149] to first order in $z_2 - 1$.

Finally, the hexatic term takes into account the fact that the dimers are not interacting in isolation, but are all mutually interacting, and is given by

$$\begin{aligned}
\mathcal{S}_6 &= z^2 \left[ z_2^3 - 3(z_2 - 1) - 1 \right] (b_1 a_1 b_2 a_3 a_2 b_3 + b_1 a_2 a_1 b_2 a_3 b_3) \\
&= z^2 \left[ (z_2 - 1)^3 + 3(z_2 - 1)^2 \right] (b_1 a_1 b_2 a_3 a_2 b_3 + b_1 a_2 a_1 b_2 a_3 b_3),
\end{aligned} \tag{155}$$

where in the first line the $-1$ removes the contribution from the quadratic action, the $3(z_2 - 1)$ removes the contribution from the quartic action and the $z_2^3$ replaces these with a contribution that correctly reproduces the weight of three mutually interacting dimers. Direct calculation including quadratic, quartic and hexatic terms correctly reproduces Eq. 149 for the partition function,

$$\begin{aligned}
&\int \prod_i da_i db_i \; e^{\mathcal{S}_2[a,b]+\mathcal{S}_4[a,b]+\mathcal{S}_6[a,b]} \\
&= 2z^2 \left[ (z_2 - 1)^3 + 3(z_2 - 1)^2 + 3(z_2 - 1) + 1 \right] = 2z^2 z_2^3.
\end{aligned} \tag{156}$$

As the lattice size is increased, it rapidly becomes impossible to determine the partition function by inspection. A full expansion of the partition function in terms of Grassmann variables also becomes complicated due to the increase in the number of terms in the action. However, the advantage of this method is that it provides a way of systematically carrying out perturbation theory around the non-interacting limit $|z_2 - 1| \to 0$. For large or infinite lattices direct evaluation of Grassmann actions with quartic and higher order interacting terms is not possible, but approximate diagrammatic methods can be used, and the order of expansion matched to that of the truncation of the action.

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
