# Peer review of "Spin-liquid behaviour and the interplay between Pokrovsky-Talapov and Ising criticality in the distorted, triangular-lattice, dipolar Ising antiferromagnet"

_SciPost Physics, doi:SciPost Phys. 5, 030 (2018)_

## Round 1 · Referee Report · Anonymous (Referee 1) · 2017-8-8

Strengths

1-Addresses a fundamental model of frustrated magnets with long-range interactions, and provides some new and interesting results.
2-Careful and systematic study of a set of related toy models and continuum theories.

Weaknesses

see report

Report

The authors study the triangular-lattice Ising antiferromagnet (TLIAF) with dipolar interactions, using a combination of analytical arguments and Monte Carlo methods. Their approach is to identify the general features of the phase diagram and then introduce a set of "toy models" and continuum theories to provide approximate descriptions of certain regions of the phase diagram. The insights from these effective models are then applied to the full dipolar TLIAF.

The paper is interesting and it provides some original insights into a fundamental model of frustrated magnets with long-range interactions. The approach used is sensible, and the final results for the phase diagram and critical properties certainly seem to be valid. I have several comments that the authors should address, as well as some more minor suggested changes listed below.

  1. I would like the authors to comment more explicitly in the manuscript about the peculiar behavior of the stripe order across the phase transition. In the limit where defects are forbidden, the stripe order apparently disappears discontinuously across the transition, as in Figure 8(a), even for a continuous transition. Does this feature survive in the full model? It seems to be possible only because the staggered magnetization is a nonlocal quantity when written in terms of the strings.

  2. I have some concerns about the interpretation of the results of perturbation theory on p. 33. First of all, is it not possible to continue the expansion in Eq. (74) beyond first-order, and see whether higher-order terms are indeed small? Separately, is there any reason to believe that first-order perturbation theory will be accurate for other quantities, besides the critical temperature? The latter is special, in that it depends only on single-string properties, and one might worry that other quantities will not be so accurate at this order.

  3. Can Eq. (81) be extended to include the effect of defect triangles considered in Section 5.2? This would allow the crossover between PT and Ising critical behaviors to be quantified. I believe it is also possible to write down scaling forms for this crossover, analogous to Eq. (83), including the Boltzmann weight of defect triangles as an additional relevant scaling variable [see PRB 87, 064414 (2013)].

  4. In Section 5.4, a free-fermion model is constructed that has a flat dispersion, which reproduces the behavior of the string density across the first-order transition. It is not completely clear how this should be interpreted. I agree that this is the only way to get a discontinuity in the string density from noninteracting fermions, but does it actually provide a useful description of the transition? In particular, does it make any nontrivial predictions for properties near the first-order transition?

Requested changes

1-If the points in Figure 1(b) are from MC, this should be stated in the caption.
2-Do the nano-magnets referred to on page 4 have their moments oriented perpendicular to the plane? This should be clarified—the physics is presumably quite different if the moments are in the plane.
3-In Section 3.1, is it possible to give some typical acceptance ratios for the MC algorithm?
4-In Section 3.3, it would be worth showing an example energy or winding-number histogram to illustrate the change to a continuous transition.
5-Toeplitz is spelled incorrectly on p. 18.
6-After Eq. (94), I think the value quoted for ζ is actually ν.
7-In Figure 25, error bars should be added to help judge the quality of the data collapse.

---

## Round 1 · Referee Report · Anonymous (Referee 2) · 2017-8-25

Strengths

  1. The paper gives a very thorough treatment of the problem studied.
  2. Reliable evidence is provided for the main results (e.g. the phase diagram in Fig 1).
  3. Clear physical insights into the problem studied are provided by discussions of subsidiary problems.

Weaknesses

  1. The paper seems very long in relation to the interest of the results. It is more in the style of a PhD thesis than a journal publication.
  2. This referee would prefer a height model treatment of the problem to the ones given.

Report

The paper gives a detailed treatment of the phase diagram of the triangular lattice Ising model with dipolar couplings, as a function of temperature and of lattice distortion (from equilateral to isosceles triangles). The motivation is partly theoretical: the nearest-neighbor Ising antiferromagnet is strongly frustrated, being disordered at all non-zero temperature and critical at zero temperature, and so it is interesting to know the perturbing effect of further neighbor interactions. The motivation is also potentially experimental, since it may be possible to realise the model as an artificial array of nanomagnets.

Behavior of the model is determined using Monte Carlo simulations, and elucidated by examining a sequence of simplified versions of the original system with the help of various mappings. To my taste, it is disappointing that the height model [H. W. J. Bloete and H. J. Hilhorst, J. Phys. A 15, L631 (1982)] is not one of the mappings that the authors use, since I think that it provides the clearest representation of the nearest neighbor model, which stands at the center of the discussion.

The paper is clearly written, although this is partly at the expense of being overlong. While the various toy models analyzed in Section 4 help understanding, I suspect that some potential readers will be discouraged by the 74-page length of the article.

Requested changes

A substantially shorter version of this paper might attract more readers. Beyond this I have only minor suggestions for changes:

[1] Eq 9 omits entropic effects, which are important at low temperature and become clear if one views the defects as vortices in a height model. This should be acknowledged and ideally corrected.

[2] It seems surprising if the content in Sec 4.2.2 is not already available in the literature. Have the authors checked this carefully?

[3] Page 25: the authors write "for T<<J_A then to all practical purposes xi-> infinity". It would be better to indicate the functional form of the divergence of xi with 1/T.

[4] End of page 25 and top of page 26: the issues referred to here were well-known long before Ref 51 - see e.g Schultz, Mattis and Lieb, Rev Mod Phys 36, 856 (1964). Similar comments apply to the last paragraph of Sec 7.

[5] End of page 33: an extensive discussion is given of an approximate dispersion relation, but I am unclear what the significance of the dispersion relation is, beyond the Hartree Fock approximation.

[6] Page 37: the discussion here would be much better if the authors had used a mapping to a height model. Using strings as they do, three symmetry-related phases appear quite different. In a height model, they would be tilted phases with different tilt directions.

[7] Eq 94. The argument of the function g_{tri} appears to me to have a misprint in it.

[8] Page 51: I disagree with the statement in paragraph 2 of Sec 7: "... the most intuitive way to understand such models is by considering the strong degrees of freedom ...". I think a height representation would be better.

[9] Eq 110 misses the relevant physics, at least for the nearest neighbor model, because it omits the dependence of degeneracy on the number and location of defects.

[10] What is the relation of appendix B to the work of Stephenson [Refs 4 and 5]? Is the appendix a rederivation?

---

## Round 2 · Referee Report · Anonymous (Referee 1) · 2018-7-12

Report

The resubmission amounts to a substantial reworking of the original manuscript, with significant reorganization of the text and some new results. I agree that the new structure, with most of the results for the toy models moved to appendices, helps to clarify the main thrust of the work. I am also happy with the authors' responses to my comments and those of the other referee, and I believe that the paper is now ready for publication.

---

## Round 2 · Referee Report · Anonymous (Referee 2) · 2018-9-9

Report

This paper has been very extensively revised in Version 2. In particular, the main results are summarised reasonably concisely in the body of the paper, with much additional material moved to appendices. I think this reorganisation will make the paper accessible to a broader readership.

The authors have also responded appropriately to suggestions and requests by both referees, in each instance either making changes or explaining why their original formulation is justified.

I recommend this version for publication as a SciPost paper.

---

## Round 2 · Author Response

List of changes

The manuscript has been substantially reorganised, as set out in the response to the referees.

---

## Round 2 · List of Changes

The manuscript has been substantially reorganised, as set out in the response to the referees.

---

## Editorial Decision

published